# Sample-Efficient Learning of Correlated Equilibria in Extensive-Form Games

**Ziang Song**
Stanford University
ziangs@stanford.edu

**Song Mei**
UC Berkeley
songmei@berkeley.edu

**Yu Bai**
Salesforce Research
yu.bai@salesforce.com

## Abstract

Imperfect-Information Extensive-Form Games (IIEFGs) is a prevalent model for real-world games involving imperfect information and sequential plays. The Extensive-Form Correlated Equilibrium (EFCE) has been proposed as a natural solution concept for multi-player general-sum IIEFGs. However, existing algorithms for finding an EFCE require full feedback from the game, and it remains open how to efficiently learn the EFCE in the more challenging bandit feedback setting where the game can only be learned by observations from repeated playing.

This paper presents the first sample-efficient algorithm for learning the EFCE from bandit feedback. We begin by proposing $K$-EFCE—a generalized definition that allows players to observe and deviate from the recommended actions for $K$ times. The $K$-EFCE includes the EFCE as a special case at $K = 1$, and is an increasingly stricter notion of equilibrium as $K$ increases. We then design an uncoupled no-regret algorithm that finds an $\varepsilon$-approximate $K$-EFCE within $\widetilde{\mathcal{O}}(\max_i X_i A_i^K / \varepsilon^2)$ iterations in the full feedback setting, where $X_i$ and $A_i$ are the number of information sets and actions for the $i$-th player. Our algorithm works by minimizing a wide-range regret at each information set that takes into account all possible recommendation histories. Finally, we design a sample-based variant of our algorithm that learns an $\varepsilon$-approximate $K$-EFCE within $\widetilde{\mathcal{O}}(\max_i X_i A_i^{K+1} / \varepsilon^2)$ episodes of play in the bandit feedback setting. When specialized to $K = 1$, this gives the first sample-efficient algorithm for learning EFCE from bandit feedback.

## 1 Introduction

This paper is concerned with the problem of learning equilibria in Imperfect-Information Extensive-Form Games (IIEFGs) [29]. IIEFGs is a general formulation for multi-player games with both imperfect information (such as private information) and sequential play, and has been used for modeling and solving real-world games such as Poker [23, 32, 7, 8], Bridge [39], Scotland Yard [37], and so on. In a two-player zero-sum IIEFG, the standard solution concept is the celebrated notion of Nash Equilibrium (NE) [35], that is, a pair of independent policies for both players such that no player can gain by deviating. However, in multi-player general-sum IIEFGs, computing an (approximate) NE is PPAD-hard and unlikely to admit efficient algorithms [12]. A more amenable class of solution concepts is the notion of *correlated equilibria* [4], that is, a correlated policy for all players such that no player can gain by deviating from the correlated play using certain types of deviations.

The notion of Extensive-Form Correlated Equilibria (EFCE) proposed by Von Stengel and Forges [40] is a natural definition of correlated equilibria in multi-player general-sum IIEFGs. An EFCE is a correlated policy that can be thought of as a "mediator" of the game who recommends actions to each player privately and sequentially (at visited information sets), in a way that disincentivizes any player to deviate from the recommended actions. Polynomial-time algorithms for computing EFCEs have been established, by formulating as a linear program and using the ellipsoid method [24, 36, 26],

36th Conference on Neural Information Processing Systems (NeurIPS 2022).

min-max optimization [16], or uncoupled no-regret dynamics using variants of counterfactual regret minimization [11, 20, 34, 2].

However, all the above algorithms require that the full game is known (or *full feedback*). In the more challenging *bandit feedback* setting where the game can only be learned by observations from repeated playing, it remains open how to learn EFCEs sample-efficiently. This is in contrast to other types of equilibria such as NE in two-player zero-sum IIEFGs, where sample-efficient learning algorithms under bandit feedback are known [30, 19, 28, 5]. A related question is about the structure of the EFCE definition: An EFCE only allows players to deviate *once* from the observed recommendations, upon which further recommendations are no longer revealed and the player needs to make decisions on her own instead. This may be too restrictive to model situations where players can still observe the recommendations after deviating [34]. It is of interest how we can extend the EFCE definition in a structured fashion to disincentivize such stronger deviations while still allowing efficient algorithms.

This paper makes steps towards answering both questions above, by proposing stronger and more generalized definition of EFCEs, and designing efficient learning algorithms under both full-feedback and bandit-feedback settings. We consider IIEFGs with $m$ players, $H$ steps, where each player $i$ has $X_i$ information sets and $A_i$ actions. Our contributions can be summarized as follows.

- We propose $K$-EFCE, a natural generalization of EFCE, at which players have no gains when allowed to observe and deviate from the recommended actions for $K$ times (Section 3). At $K = 1$, the $K$-EFCE is equivalent to the existing definition of EFCE based on trigger policies. For $K \geq 1$, the $K$-EFCE are increasingly stricter notions of equilibria as $K$ increases.

- We design an algorithm $K$ Extensive-Form Regret Minimization ($K$-EFR) which finds an $\varepsilon$-approximate $K$-EFCE within $\widetilde{\mathcal{O}}(\max_i X_i A_i^{K \wedge H}/\varepsilon^2)$ iterations under full feedback (Section 4). At $K = 1$, our linear in $\max_i X_i$ dependence improves over the best known $\widetilde{\mathcal{O}}(\max_i X_i^2)$ dependence for computing $\varepsilon$-approximate EFCE. At $K > 1$, this gives a sharp result for efficiently computing (the stricter) $K$-EFCE, improving over the best known $\widetilde{\mathcal{O}}(\max_i X_i^2 A_i^{K \wedge H}/\varepsilon^2)$ iteration complexity of Morrill et al. [34].

- We further design Balanced $K$-EFR—a sample-based variant of $K$-EFR—for the more challenging bandit-feedback setting (Section 5). Balanced $K$-EFR learns an $\varepsilon$-approximate $K$-EFCE within $\widetilde{\mathcal{O}}(\max_i X_i A_i^{K \wedge H+1}/\varepsilon^2)$ episodes of play. This is the first line of results for learning EFCE and $K$-EFCE from bandit feedback, and the linear in $X_i$ dependence matches information-theoretic lower bounds. Technically, our bandit-feedback result builds on a novel stochastic wide range regret minimization algorithm SWRHEDGE, as well as sample-based estimators of counterfactual loss functions using newly designed sampling policies, which may be of independent interest.

## 1.1 Related work

**Computing Correlated Equilibria from full feedback**    The notion of Extensive-Form Correlated Equilibria (EFCE) in IIEFGs is introduced in Von Stengel and Forges [40]. Huang and von Stengel [24] design the first polynomial time algorithm for computing EFCEs in multi-player IIEFGs from full feedback, using a variation of the *Ellipsoid against hope* algorithm [36, 26]. Farina et al. [16] later propose a min-max optimization formulation of EFCEs which can be solved by first-order methods.

Celli et al. [11] and its extended version [20] design the first uncoupled no-regret algorithm for computing EFCEs. Their algorithms are based on minimizing the trigger regret (first considered in Dudik and Gordon [13], Gordon et al. [22]) via counterfactual regret decomposition [45]. Morrill et al. [34] propose the stronger definition of "Behavioral Correlated Equilibria" (BCE) using general "behavioral deviations", and design the Extensive-Form Regret Minimization (EFR) algorithm to compute a BCE by using a generalized version of counterfactual regret decomposition. They also propose intermediate notions such as a "Twice Informed Partial Sequence" (TIPS) (and its $K$-shot generalization) as an interpolation between the EFCE and BCE. Our definition of $K$-EFCE offers a new interpolation between the EFCE and BCE that is different from theirs, as the deviating player in $K$-EFCE does not observe and does not follow recommended actions after $K$ deviations has happened, whereas the deviator in $K$-shot Informed Partial Sequence resumes to following after $K$ deviations has happened. The iteration complexity for computing an $\varepsilon$-approximate correlated equilibrium in both [20, 34] scales quadratically in $\max_{i \in [m]} X_i$. Our $K$-EFR algorithm for the

full feedback setting builds upon the EFR algorithm, but specializes to the notion of $K$-EFCE, and achieve an improved linear in $\max_{i \in [m]} X_i$ iteration complexity.

Apart from the EFCE, there are other notions of (coarse) correlated equilibria in IIEFGs such as Normal-Form Coarse-Correlated Equilibria (NFCCE) [45, 10, 9, 18, 44], Extensive-Form Coarse-Correlated Equilibria (EFCCE) [17], and Agent-Form (Coarse-)Correlated Equilibria (AF(C)CE) [38, 40]; see [33, 34] for a detailed comparison. All above notions are either weaker than or incomparable with the EFCE, and thus results there do not imply results for computing EFCE.

**Learning Equilibria from bandit feedback**    A line of work considers learning Nash Equilibria (NE) in two-player zero-sum IIEFGs and NFCCE in multi-player general-sum IIEFGs from bandit feedback [30, 19, 15, 21, 43, 41, 28, 5]; Note that the NFCCE is weaker than (and does not imply results for learning) EFCE. Dudík and Gordon [14] consider sample-based learning of EFCE in succinct extensive-form games; however, their algorithm relies on an approximate Markov-Chain Monte-Carlo (MCMC) sampling subroutine that does not lead to an end-to-end sample complexity guarantee. To our best knowledge, our results are the first for learning the EFCE (and $K$-EFCE) under bandit feedback.

## 2    Preliminaries

We formulate IIEFGs as partially-observable Markov games (POMGs) with tree structure and perfect recall, following [28, 5]. For a positive integer $i$, we denote by $[i]$ the set $\{1, 2, \cdots, i\}$. For a finite set $\mathcal{A}$, we let $\Delta(\mathcal{A})$ denote the probability simplex over $\mathcal{A}$. Let $\binom{n}{m}$ denote the binomial coefficient (i.e. number of combinations) of choosing $m$ elements from $n$ different elements, with the convention that $\binom{n}{m} = 0$ if $m > n$. We use $m \wedge n$ to denote $\min\{m, n\}$.

**Partially observable Markov game**    We consider an episodic, tabular, $m$-player, general-sum, partially observable Markov game

$$\text{POMG}(m, \mathcal{S}, \{\mathcal{X}_i\}_{i \in [m]}, \{\mathcal{A}_i\}_{i \in [m]}, H, p_0, \{p_h\}_{h \in [H]}, \{r_{i,h}\}_{i \in [m], h \in [H]}),$$

where $\mathcal{S}$ is the state space of size $|\mathcal{S}| = S$; $\mathcal{X}_i$ is the space of information sets (henceforth *infosets*) for the $i^{\text{th}}$ player, which is a partition of $\mathcal{S}$ (i.e., $x_i \subseteq \mathcal{S}$ for all $x_i \in \mathcal{X}_i$, and $\mathcal{S} = \sqcup_{x_i \in \mathcal{X}_i} x_i$ where $\sqcup$ stands for disjoint union) with size $|\mathcal{X}_i| = X_i$, and we also use $x_i : \mathcal{S} \to \mathcal{X}_i$ to denote the $i^{\text{th}}$ player's emission (observation) function; $\mathcal{A}_i$ is the action space for the $i^{\text{th}}$ player with size $|\mathcal{A}_i| = A_i$, and we let $\mathcal{A} = \mathcal{A}_1 \times \cdots \mathcal{A}_m$ denote the space of joint actions $\mathbf{a} = (a_1, \ldots, a_m)$; $H \in \mathbb{Z}_{\geq 1}$ is the time horizon; $p_0 \in \Delta(\mathcal{S})$ is the distribution of the initial state $s_1$; $p_h : \mathcal{S} \times \mathcal{A} \to \Delta(\mathcal{S})$ are transition probabilities where $p_h(s_{h+1}|s_h, \mathbf{a}_h)$ is the probability of transiting to the next state $s_{h+1}$ from state-action $(s_h, \mathbf{a}_h) \in \mathcal{S} \times \mathcal{A}$; and $r_{i,h} : \mathcal{S} \times \mathcal{A} \to [0, 1]$ is the deterministic[1] reward function for the $i^{\text{th}}$ player at step $h$.

**Tree structure and perfect recall assumption**    We use a POMG with tree structure and perfect recall to formulate imperfect information games[2][3], following [28, 5]. We assume that the game has a tree structure: for any state $s \in \mathcal{S}$, there is a unique step $h$ and history $(s_1, \mathbf{a}_1, \ldots, s_{h-1}, \mathbf{a}_{h-1}, s_h = s)$ to reach $s$. Precisely, for any policy of the players, for any realization of the game (i.e., trajectory) $(s'_k, \mathbf{a}'_k)_{k \in [H]}$, conditionally to $s'_l = s$, it almost surely holds that $l = h$ and $(s'_1, \ldots, s'_h) = (s_1, \ldots, s_h)$. We also assume perfect recall, which means that each player remembers its past information sets and actions. In particular, for each information set (infoset) $x_i \in \mathcal{X}_i$ for the $i^{\text{th}}$ player, there is a unique history of past infosets and actions $(x_{i,1}, a_{i,1}, \ldots, x_{i,h-1}, a_{i,h-1}, x_{i,h} = x_i)$ leading to $x_i$. This requires that $\mathcal{X}_i$ can be partitioned to $H$ subsets $(\mathcal{X}_{i,h})_{h \in [H]}$ such that $x_{i,h} \in \mathcal{X}_{i,h}$ is reachable only at time step $h$. We define $X_{i,h} := |\mathcal{X}_{i,h}|$. Similarly, the state set $\mathcal{S}$ can be also partitioned into $H$ subsets $(S_h)_{h \in [H]}$. As we mostly focus on the $i^{\text{th}}$ player, we use $x_h$ to also denote $x_{i,h}$ and use them interchangeably.

---

[1]Our results can be directly extended to the case of stochastic rewards.

[2]We remark that POMGs with tree structure and perfect recall is a specific subclass of POMGs; General POMGs could possess different challenges and are beyond the scope of this paper.

[3]Our definition of POMGs with tree structure and perfect recall can express any IIEFG satisfying an additional mild condition called *timeability* [25]. Further, our algorithms and guarantees can be generalized directly to any general IIEFG that is not necessarily timeable; see Appendix H.3 for discussions.

For $s \in \mathcal{S}$ and $x_i \in \mathcal{X}_i$, we write $s \in x_i$ if infoset $x_i$ contains the state $s$. With an abuse of notation, for $s \in \mathcal{S}$, we let $x_i(s)$ denote the $i^{\text{th}}$ player's infoset that $s$ belongs to. For any $h < h'$, $x_{i,h} \in \mathcal{X}_{i,h}, x_{i,h'} \in \mathcal{X}_{i,h'}$, we write $x_{i,h} \prec x_{i,h'}$ if the information set $x_{i,h'}$ can be reached from information set $x_{i,h}$ by $i^{\text{th}}$ player's actions; we write $(x_{i,h}, a_{i,h}) \prec x_{i,h'}$ if the infoset $x_{i,h'}$ can be reached from infoset $x_{i,h}$ by $i^{\text{th}}$ player's action $a_{i,h}$. For any $h < h'$ and $x_{i,h} \in \mathcal{X}_{i,h}$, we let $\mathcal{C}_{h'}(x_{i,h}, a_{i,h}) := \{x \in \mathcal{X}_{i,h'} : (x_{i,h}, a_{i,h}) \prec x\}$ and $\mathcal{C}_{h'}(x_{i,h}) := \{x \in \mathcal{X}_{i,h'} : x_{i,h} \prec x\} = \cup_{a_{i,h} \in \mathcal{A}_i} \mathcal{C}_{h'}(x_{i,h}, a_{i,h})$ denote the infosets within the $h'$-th step that are reachable from (i.e. children of) $x_{i,h}$ or $(x_{i,h}, a_{i,h})$, respectively. For shorthand, let $\mathcal{C}(x_{i,h}, a_{i,h}) := \mathcal{C}_{h+1}(x_{i,h}, a_{i,h})$ and $\mathcal{C}(x_{i,h}) := \mathcal{C}_{h+1}(x_{i,h})$ denote the set of immediate children.

**Policies**  We use $\pi_i = \{\pi_{i,h}(\cdot|x_{i,h})\}_{h \in [H], x_{i,h} \in \mathcal{X}_{i,h}}$ to denote a policy of the $i^{\text{th}}$ player, where each $\pi_{i,h}(\cdot|x_{i,h}) \in \Delta(\mathcal{A}_i)$ is the action distribution at infoset $x_{i,h}$. We say $\pi_i$ is a pure policy if $\pi_{i,h}(\cdot|x_{i,h})$ takes some single action deterministically for any $(h, x_{i,h})$; in this case we let $\pi_i(x_{i,h}) = \pi_{i,h}(x_{i,h})$ denote the action taken at infoset $x_{i,h}$ for shorthand. We use $\pi = \{\pi_i\}_{i \in [m]}$ to denote a product policy for all players, and let $\pi_{-i} = \{\pi_j\}_{j \in [m], j \neq i}$ denote policies of all players other than the $i^{\text{th}}$ player. We call $\pi$ a pure product policy if $\pi_i$ is a pure policy for all $i \in [m]$. Let $\Pi_i$ denote the set of all possible policies for the $i^{\text{th}}$ player and $\Pi = \prod_{i \in [m]} \Pi_i$ denote the set of all possible product policies. Any probability measure $\overline{\pi}$ on $\Pi$ induces a *correlated policy*, which executes as first sampling a product policy $\pi = \{\pi_i\}_{i \in [m]} \in \Pi$ from probability measure $\overline{\pi}$ and then playing the product policy $\pi$. We also use $\overline{\pi}$ to denote this policy. A correlated policy $\overline{\pi}$ can be viewed as a *mediator* of the game which samples $\pi \sim \overline{\pi}$ before the game starts, and privately *recommends* action sampled from $\pi_i(\cdot|x_i)$ to the $i^{\text{th}}$ player when infoset $x_i \in \mathcal{X}_i$ is visited during the game.

**Reaching probability**  With the tree structure assumption, for any state $s_h \in \mathcal{S}_h$ and actions $\mathbf{a} \in \mathcal{A}$, there exists a unique history $(s_1, \mathbf{a}_1, \ldots, s_h = s, \mathbf{a}_h = \mathbf{a})$ ending with $(s_h = s, \mathbf{a}_h = \mathbf{a})$. Given any product policy $\pi$, the probability of reaching $(s_h, \mathbf{a}_h)$ at step $h$ can be decomposed as

$$p_h^\pi(s_h, \mathbf{a}) = p_{1:h}(s_h) \prod_{i \in [m]} \pi_{i,1:h}(s_h, a_{i,h}), \tag{1}$$

where we define the *sequence-form transitions* $p_{1:h}$ and *sequence-form policies* $\pi_{i,1:h}$ as

$$p_{1:h}(s_h) := p_0(s_1) \prod_{h'=1}^{h-1} p_{h'}(s_{h'+1}|s_{h'}, \mathbf{a}_{h'}), \tag{2}$$

$$\pi_{i,1:h}(s_h, a_{i,h}) := \pi_{i,1:h}(x_{i,h}, a_{i,h}) := \prod_{h'=1}^{h} \pi_{i,h'}(a_{i,h'}|x_{i,h'}), \tag{3}$$

where $(s_{h'}, \mathbf{a}_{h'})_{h' \leq h-1}$ is the unique history of states and actions that leads to $s_h$ by the tree structure; $x_{i,h} = x_i(s_h)$ is the $i^{\text{th}}$ player's infoset at the $h$-th step, and $(x_{i,h'}, a_{i,h'})_{h' \leq h-1}$ is the unique history of infosets and actions that leads to $x_{i,h}$ by perfect recall. We also define $\pi_{i,h:h'}(x_{i,h'}, a_{i,h'}) := \prod_{h''=h}^{h'} \pi_{i,h''}(a_{i,h''}|x_{i,h''})$ for any $1 \leq h \leq h' \leq H$.

**Value functions and counterfactual loss functions**  Let $V_i^\pi := \mathbb{E}_\pi[\sum_{h=1}^{H} r_{i,h}]$ denote the value function (i.e. expected cumulative reward) for the $i^{\text{th}}$ player under policy $\pi$. By the product form of the reaching probability in (1), the value function $V_i^\pi$ admits a multi-linear structure over the sequence-form policies. Concretely, fixing any sequence of product policies $\{\pi^t\}_{t=1}^{T}$ where each $\pi^t = \{\pi_i^t\}_{i \in [m]}$, we have

$$V_i^{\pi^t} = \sum_{h=1}^{H} \sum_{(s_h, \mathbf{a}_h = (a_{j,h})_{j \in [m]}) \in \mathcal{S}_h \times \mathcal{A}} p_{1:h}(s_h) \prod_{j=1}^{m} \pi_{j,1:h}^t(x_j(s_h), a_{j,h}) r_{i,h}(s_h, \mathbf{a}_h).$$

For any sequence of policies $\{\pi^t\}_{t=1}^{T}$, we also define the *counterfactual loss functions* [45] $\{L_{i,h}^t(x_{i,h}, a_{i,h})\}_{i,h,x_{i,h},a_{i,h}}$ as:

$$\ell_{i,h}^t(x_{i,h}, a_{i,h}) := \sum_{\substack{s_h \in x_{i,h}, \\ \mathbf{a}_{-i,h} \in \mathcal{A}_{-i}}} p_{1:h}(s_h) \prod_{j \neq i} \pi_{j,1:h}^t(x_j(s_h), a_{j,h})[1 - r_{i,h}(s_h, \mathbf{a}_h)], \tag{4}$$

$$L_{i,h}^t(x_{i,h}, a_{i,h}) := \ell_{i,h}^t(x_{i,h}, a_{i,h}) + \sum_{h'=h+1}^{H} \sum_{\substack{x_{h'} \in \mathcal{C}_{h'}(x_{i,h}, a_{i,h}), \\ a_{h'} \in \mathcal{A}_i}} \pi_{i,(h+1):h'}^t(x_{h'}, a_{h'}) \ell_{i,h'}^t(x_{h'}, a_{h'}). \tag{5}$$

---

**Algorithm 1** Executing modified policy $\phi \diamond \pi_i$

---

**Input:** $K$-EFCE strategy modification $\phi \in \Phi_i^K$ ($0 \leq K \leq \infty$), policy $\pi_i \in \Pi_i$ for the $i^{\text{th}}$ player.
1: Initialize recommendation history $\mathbf{b} = \emptyset$.
2: **for** $h = 1, \ldots, H$ **do**
3:     Receive infoset $x_{i,h} \in \mathcal{X}_{i,h}$.
4:     **if** $\mathbf{b} \in \Omega_i^{(\text{I}),K}(x_{i,h})$ **then**
5:         Observe recommendation $b_h \sim \pi_{i,h}(\cdot|x_{i,h})$.
6:         Take swapped action $a_h = \phi(x_{i,h}, \mathbf{b}, b_h)$.
7:         Update recommendation history $\mathbf{b} \leftarrow (\mathbf{b}, b_h) \in \mathcal{A}_i^h$.
8:     **else**
9:         // Must have $\mathbf{b} \in \Omega_i^{(\text{II}),K}(x_{i,h})$, do not observe recommendation from $\pi_i$
10:        Take action $a_h = \phi(x_{i,h}, \mathbf{b})$.

---

Intuitively, $L_{i,h}^t(x_{i,h}, a_{i,h})$ measures the $i^{\text{th}}$ player's expected cumulative loss (one minus reward) conditioned on reaching $(x_{i,h}, a_{i,h})$, weighted by the (environment) transitions and all other players' policies $\pi_{-i}^t$ at all time steps, and the $i^{\text{th}}$ player's own policy $\pi_i^t$ from step $h+1$ onward. We will omit the $i$ subscript and use $L_h^t$ to denote the above when clear from the context.

**Feedback protocol** We consider two standard feedback protocols for our algorithms: *full feedback*, and *bandit feedback*. In the full feedback case, the algorithm can query a product policy $\pi^t = \{\pi_i^t\}_{i \in [m]}$ in each iteration and observe the counterfactual loss functions $\{L_{i,h}^t(x_{i,h}, a_{i,h})\}_{i,h,x_{i,h},a_{i,h}}$ *exactly*[4]. In the bandit feedback case, the players can only play repeated episodes with some policies and observe the trajectory of their own infosets and rewards from the environment.

# 3 $K$ Extensive-Form Correlated Equilibria

We now introduce the definition of $K$ Extensive-Form Correlated Equilibria ($K$-EFCE) and establish its relationship with existing notions of correlated equilibria in IIEFGs.

## 3.1 Definition of $K$-EFCE

Intuitively, a $K$-EFCE is a correlated policy in which no player can gain if allowed to deviate from the observed recommended actions $K$ times, and forced to choose her own actions without observing further recommendations afterwards. To state its definition formally, letting $\mathcal{A}_i^h := \{(b_1, \ldots, b_h)|b_{h'} \in \mathcal{A}_i, \forall h' \leq h\}$, we categorize all possible *recommendation histories* (henceforth *rechistories*) at each infoset $x_{i,h} \in \mathcal{X}_{i,h}$ (for the $i^{\text{th}}$ player) into two types, based on whether the player has already deviated $K$ times from past recommendations:

(1) A *Type-I rechistory* ($\leq K-1$ deviations happened) at $x_{i,h}$ is any action sequence $b_{1:h-1} \in \mathcal{A}_i^{h-1}$ such that $\sum_{k=1}^{h-1} \mathbf{1}\{a_k \neq b_k\} \leq K - 1$, where $(a_1, \ldots, a_{h-1})$ is the unique sequence of actions leading to $x_{i,h}$. Let $\Omega_i^{(\text{I}),K}(x_{i,h})$ denote the set of all Type-I rechistories at $x_{i,h}$.

(2) A *Type-II rechistory* ($K$ deviations happened) at $x_{i,h}$ is any action sequence $b_{1:h'} \in \mathcal{A}_i^{h'}$ with length $h' < h$ such that $\sum_{k=1}^{h'-1} \mathbf{1}\{a_k \neq b_k\} = K - 1$ and $a_{h'} \neq b_{h'}$, where $(a_1, \ldots, a_{h-1})$ is the unique sequence of actions leading to $x_{i,h}$. Let $\Omega_i^{(\text{II}),K}(x_{i,h})$ denote the set of all Type-II rechistories at $x_{i,h}$.

We now define a $K$-EFCE strategy modification ($0 \leq K \leq \infty$) for the $i^{\text{th}}$ player.

**Definition 1** ($K$-EFCE strategy modification). *A $K$-EFCE strategy modification $\phi$ (for the $i^{th}$ player) is a mapping $\phi$ of the following form: At any infoset $x_{i,h} \in \mathcal{X}_{i,h}$, for any Type-I rechistory $b_{1:h-1} \in \Omega_i^{(\text{I}),K}(x_{i,h})$, $\phi$ swaps any recommended action $b_h$ into $\phi(x_{i,h}, b_{1:h-1}, b_h) \in \mathcal{A}_i$; for any Type-II rechistory $b_{1:h'} \in \Omega_i^{(\text{II}),K}(x_{i,h})$, $\phi$ directly takes action $\phi(x_{i,h}, b_{1:h'}) \in \mathcal{A}_i$.*

---

[4]This is implementable (and slightly more general than) when the full game (transitions and rewards) is known.

Let $\Phi_i^K$ denote the set of all possible $K$-EFCE strategy modifications for any $0 \leq K \leq \infty$. Formally, for any $\phi \in \Phi_i^K$ and any pure policy $\pi_i \in \Pi_i$, we define the modified policy $\phi \diamond \pi_i$ as in Algorithm 1.

We parse the modified policy $\phi \diamond \pi_i$ (Algorithm 1) as follows. Upon receiving the infoset $x_{i,h}$ at each step $h$, the player has the rechistory $\mathbf{b}$ containing all past observed recommended actions. Then, if $\mathbf{b}$ is Type-I, i.e. at most $K - 1$ deviations have happened (Line 4), then the player observes the current recommended action $b_h = \pi_{i,h}(x_{i,h})$, takes a potentially swapped action $a_h = \phi(x_{i,h}, \mathbf{b}, b_h)$ (Line 6), and appends $b_h$ to the recommendation history (Line 7). Otherwise, $(x_{i,h}, \mathbf{b})$ is Type-II, i.e. $K$ deviations have already happened. In this case, the player does not observe the recommended action, and instead takes an action $a_h = \phi(x_{i,h}, \mathbf{b})$, and does not update $\mathbf{b}$ (Line 10).

We now define $K$-EFCE as the equilibrium induced by the $K$-EFCE strategy modification set $\Phi_i^K$. With slight abuse of notation, we define $\phi \diamond \overline{\pi}$ for any *correlated policy* $\overline{\pi}$ to be the policy $(\phi \diamond \pi_i) \times \pi_{-i}$ where $\pi \sim \overline{\pi}$ is the product policy sampled from $\overline{\pi}$.

**Definition 2** ($K$-EFCE)**.** *A correlated policy $\overline{\pi}$ is an $\varepsilon$-approximate $K$ Extensive-Form Correlated Equilibrium (K-EFCE) if*

$$K\text{-EFCEGap}(\overline{\pi}) := \max_{i \in [m]} \max_{\phi \in \Phi_i^K} \left( V_i^{\phi \diamond \overline{\pi}} - V_i^{\overline{\pi}} \right) \leq \varepsilon.$$

*We say $\overline{\pi}$ is an (exact) $K$-EFCE if $K\text{-EFCEGap}(\overline{\pi}) = 0$.*

### 3.2 Properties of $K$-EFCE

The $K$-EFCE is closely related to various existing definitions of correlated equilibria in IIEFGs. We show that the special case of $K = 1$ is equivalent to the existing definition of EFCE based on trigger policies (Proposition C.1); The $K$-EFCE are indeed stricter equilibria as $K$ increases (Proposition C.2); The two extreme cases $K = 0$ and $K = \infty$ are equivalent to (Normal-Form) Coarse Correlated Equilibrium and the "Behavioral Correlated Equilibria" of [34][5], respectively (Proposition C.3). Due to the space limit, the full statements and proofs are deferred to Appendix C.

## 4 Computing $K$-EFCE from full feedback

**Algorithm description**    We first present our algorithm for computing $K$-EFCE in the full-feedback setting. Our algorithm $K$ Extensive-Form Regret Minimization ($K$-EFR), described in Algorithm 2, is an uncoupled no-regret algorithm aiming to minimize the following *K-EFCE regret*

$$R_{i,K}^T := \max_{\phi \in \Phi_i^K} \sum_{t=1}^T \left( V_i^{\phi \diamond \pi_i^t \times \pi_{-i}^t} - V_i^{\pi^t} \right). \tag{6}$$

By standard online-to-batch conversion, achieving sublinear $K$-EFCE regret for every player implies that the average joint policy over all players is an approximate $K$-EFCE (Lemma E.1).

At a high level, our Algorithm 2 builds upon the EFR algorithm of Morrill et al. [34] to minimize the $K$-EFCE regret $R_{i,K}^T$, by maintaining a *regret minimizer* $\mathcal{R}_{x_{i,h}}$ (using algorithm REGALG) at each infoset $x_{i,h} \in \mathcal{X}_i$ that is responsible for outputting the policy $\pi_i^t(\cdot | x_{i,h}) \in \Delta_{\mathcal{A}_i}$ (Line 8) which combine to give the overall policy $\pi_i^t$ for the $t$-th iteration.

Core to our algorithm is the requirement that $\mathcal{R}_{x_{i,h}} \sim$ REGALG should be able to minimize regrets with *time-selection functions and strategy modifications* (also known as the *wide range regret*) [31, 6]. Specifically, $\mathcal{R}_{x_{i,h}}$ needs to control the regret

$$\max_{\varphi \in \Psi^s} \sum_{t=1}^T \prod_{k=1}^{h-1} \pi_i^t(b_k | x_k) \left( \left\langle \pi_{i,h}^t(\cdot | x_{i,h}) - \varphi \diamond \pi_{i,h}^t(\cdot | x_{i,h}), L_{i,h}^t(x_{i,h}, \cdot) \right\rangle \right) \tag{7}$$

for all possible Type-I rechistories $b_{1:h-1} \in \Omega_i^{(\text{I}),K}(x_{i,h})$ simultaneously, where $\prod_{k=1}^{h-1} \pi_i^t(b_k | x_k) =: S_{b_{1:h-1}}^t$ is the *time-selection function* (i.e. a weight function) associated with this $b_{1:h-1}$ (cf. Line 5), and $\Psi^s = \{\psi : \mathcal{A}_i \to \mathcal{A}_i\}$ is the set of all *swap modifications* from the action set $\mathcal{A}_i$ onto itself. (An analogous regret for Type-II rechistories is also controlled by $\mathcal{R}_{x_{i,h}}$.) Controlling these "local"

---

---

[5]Up to a minor difference that our $\infty$-EFCE only defines the equilibrium in terms of the overall game value, where the BCE additionally requries similar equilibrium properties to hold in certain subgames.

**Algorithm 2** $K$-EFR with full feedback ($i^{\text{th}}$ player's version)

---

**Input:** Algorithm REGALG for minimizing wide range regret; learning rates $\{\eta_{x_{i,h}}\}_{x_{i,h} \in \mathcal{X}_i}$.

1: Initialize regret minimizers $\{\mathcal{R}_{x_{i,h}}\}_{x_{i,h} \in \mathcal{X}_i}$ with REGALG and learning rate $\eta_{x_{i,h}}$.
2: **for** iteration $t = 1, \ldots, T$ **do**
3:     **for** $h = 1, \ldots, H$ **do**
4:         **for** $x_{i,h} \in \mathcal{X}_{i,h}$ **do**
5:             Compute $S^t_{b_{1:h-1}} = \prod_{k=1}^{h-1} \pi^t_{i,k}(b_k | x_k)$ for all $b_{1:h-1} \in \Omega^{(\mathrm{I}),K}_i(x_{i,h})$.
6:             Compute $S^t_{b_{1:h'}} = \prod_{k=1}^{h'} \pi^t_{i,k}(b_k | x_k)$ for all $b_{1:h'} \in \Omega^{(\mathrm{II}),K}_i(x_{i,h})$.
7:             $\mathcal{R}_{x_{i,h}}.\text{OBSERVE\_TIMESELECTION}(\{S^t_{b_{1:h-1}}\}_{b_{1:h-1} \in \Omega^{(\mathrm{I}),K}_i(x_{i,h})} \cup \{S^t_{b_{1:h'}}\}_{b_{1:h'} \in \Omega^{(\mathrm{II}),K}_i(x_{i,h})})$.

8:             Set policy $\pi^t_i(\cdot | x_{i,h}) \leftarrow \mathcal{R}_{x_{i,h}}.\text{RECOMMEND}()$.
9:     Observe counterfactual losses $\{L^t_h(x_{i,h}, a_h)\}_{h, x_{i,h}, a_h}$ (depending on $\pi^t_i$ and $\pi^t_{-i}$; cf. (5)).
10:     **for all** $x_{i,h} \in \mathcal{X}_i$ **do**
11:         $\mathcal{R}_{x_{i,h}}.\text{OBSERVE\_LOSS}(\{L^t_h(x_{i,h}, a)\}_{a \in \mathcal{A}_i})$.

**Output:** Policies $\{\pi^t_i\}_{t=1}^T$.

---

regrets at each $x_{i,h}$ guarantees that the overall $K$-EFCE regret is bounded, by the $K$-EFCE regret decomposition (cf. Lemma E.2).

To control this wide range regret, we instantiate REGALG as WRHEDGE (Algorithm 4; cf. Appendix A.2), which is similar as the wide regret minimization algorithm in [27], with a slight modification of the initial weights suitable for our purpose (cf. (11)). The learning rate is set as

$$\eta_{x_{i,h}} = \sqrt{\binom{H}{K \wedge H} X_i A_i^{K \wedge H} \log A_i / (H^2 T)} \tag{8}$$

for all $x_{i,h} \in \mathcal{X}_i$. With this algorithm in place, at each iteration, $\mathcal{R}_{x_{i,h}}$ observes all time selection functions (Line 7), computes the policy for the current iteration (Line 8), and then observes the loss vector $L^t_{i,h}(x_{i,h}, \cdot)$ (Line 9) that is useful for updating the policy in the next iteration.

**Theoretical guarantee**    We are now ready to present the theoretical guarantee for $K$-EFR.

**Theorem 3** (Computing $K$-EFCE from full feedback). *For any $0 \leq K \leq \infty$, $\varepsilon \in (0, H]$, let all players run Algorithm 2 together in a self-play fashion where* REGALG *is instantiated as Algorithm 4 with learning rates specified in (8). Let $\pi^t = \{\pi^t_i\}_{i \in [m]}$ denote the joint policy of all players at the $t$'th iteration. Then the average policy $\overline{\pi} = \mathrm{Unif}(\{\pi^t\}_{t=1}^T)$ satisfies $K$-EFCEGap$(\overline{\pi}) \leq \varepsilon$, as long as the number of iterations*

$$T \geq \mathcal{O}\left(\binom{H}{K \wedge H}\left(\max_{i \in [m]} X_i A_i^{K \wedge H}\right) \iota / \varepsilon^2\right),$$

*where $\iota = \log(\max_{i \in [m]} A_i)$ is a log factor and $\mathcal{O}(\cdot)$ hides $\mathrm{poly}(H)$ factors.*

In the special case of $K = 1$, Theorem 3 shows that $K$-EFR can compute an $\varepsilon$-approximate 1-EFCE within $\widetilde{\mathcal{O}}(\max_{i \in [m]} X_i A_i / \varepsilon^2)$ iterations. This improves over the existing $\widetilde{\mathcal{O}}(\max_{i \in [m]} X_i^2 A_i^2 / \varepsilon^2)$ iteration complexity of Celli et al. [11], Farina et al. [20] by a factor of $X_i A_i$. Also, compared with the iteration complexity of the optimistic algorithm of [3] which is at least $\widetilde{\mathcal{O}}(\max_{i \in [m]} X_i^{4-\delta} A_i^{4/3} / \varepsilon^{4/3})$[6], we achieve lower $X_i$ dependence (though worse $\varepsilon$ dependence).

For $1 < K \leq \infty$, Theorem 3 gives a sharp $\widetilde{\mathcal{O}}(\binom{H}{K \wedge H}(\max_{i \in [m]} X_i A_i^{K \wedge H}) / \varepsilon^2)$ iteration complexity for computing $K$-EFCE. This improves over the $\widetilde{\mathcal{O}}(\binom{H}{K \wedge H}(\max_{i \in [m]} X_i^2 A_i^{K \wedge H}) / \varepsilon^2)$ rate of EFR [34] instantiated to the $K$-EFCE problem. Also, note that although the term $A_i^{K \wedge H}$ is exponential in $K$ (for $K \leq H$), this is sensible since it is roughly the same scale as the number of possible recommendation histories, which is also the "degree of freedom" within a $K$-EFCE strategy modification. Apart from learning equilibria, Algorithm 2 also achieves a low $K$-EFCE regret when controlling the $i^{\text{th}}$ player only and facing potentially adversarial opponents:

---

[6]More precisely, Anagnostides et al. [3, Corollary 4.17] proves an $\widetilde{\mathcal{O}}((X_i \max_{\pi_i \in \Pi_{\max}} \|\pi_i\|_1^2 A / \varepsilon)^{4/3})$ iteration complexity, which specializes to the above rate, as for any $\delta > 0$ a game with $\max_{\pi_i \in \Pi_{\max}} \|\pi_i\|_1 \geq X_i^{1-\delta}$ can be constructed.

---

**Algorithm 3** Loss estimator for Type-II rechistories via Balanced Sampling ($i^{\text{th}}$ player's version)

---

**Input:** Policy $\pi_i^t, \pi_{-i}^t$. Balanced exploration policies $\{\pi_i^{\star,h}\}_{h\in[H]}$.

1: **for** $K \leq h' < h \leq H$, $W \subseteq [h']$ with $|W| = K$ and ending in $h'$ **do**

2:   Set policy $\pi_i^{t,(h,h',W)} \leftarrow (\pi_{i,k}^{\star,h})_{k\in W\cup\{h'+1,\ldots,h\}} \cdot (\pi_{i,k}^t)_{k\in[h']\setminus W} \cdot \pi_{i,(h+1):H}^t$.

3:   Play $\pi_i^{t,(h,h',W)} \times \pi_{-i}^t$ for one episode, observe trajectory

$$(x_{i,1}^{t,(h,h',W)}, a_{i,1}^{t,(h,h',W)}, r_{i,1}^{t,(h,h',W)}, \ldots, x_{i,H}^{t,(h,h',W)}, a_{i,H}^{t,(h,h',W)}, r_{i,H}^{t,(h,h',W)}).$$

4: **for** all $(x_{i,h}, b_{1:h'}) \in \Omega_i^{(\text{II}),K}$ **do**

5:   Find $(x_{i,1}, a_1) \prec \cdots \prec (x_{i,h-1}, a_{h-1}) \prec x_{i,h}$.

6:   Set $W \leftarrow \{k \in [h'] : b_k \neq a_k\}$

7:   Construct loss estimator for all $a \in \mathcal{A}_i$

$$\widetilde{L}_{(x_{i,h},b_{1:h'})}^t(a) \leftarrow \frac{\mathbf{1}\left\{(x_{i,h}^{t,(h,h',W)}, a_{i,h}^{t,(h,h',W)}) = (x_{i,h}, a)\right\}}{\pi_{i,1:h}^{t,(h,h',W)}(x_{i,h}, a)} \cdot \sum_{h''=h}^{H} \left(1 - r_{i,h''}^{t,(h,h',W)}\right). \tag{9}$$

**Output:** Loss estimators $\left\{\widetilde{L}_{(x_{i,h},b_{1:h'})}^t(\cdot)\right\}_{(x_{i,h},b_{1:h'})\in\Omega_i^{(\text{II}),K}}$.

---

$R_{i,K}^T \leq \widetilde{\mathcal{O}}(\sqrt{\binom{H}{K\wedge H}X_i A_i^{K\wedge H}T})$ (Corollary F.1). In particular, the $\widetilde{\mathcal{O}}(\sqrt{X_i T})$ scaling is optimal up to log factors, due to the fact that $R_{i,K}^T \geq R_{i,0}^T$ (i.e. the vanilla regret) and the known lower bound $R_{i,0}^T \geq \Omega(\sqrt{X_i T})$ in IIEFGs [42].

**Proof overview** Our Theorem 3 follows from a sharp analysis on the $K$-EFCE regret of Algorithm 2, by incorporating (i) a decomposition of the $K$-EFCEGap into local regrets at each infoset with tight leading coefficients (Lemma E.2), and (ii) loss-dependent upper bounds for the wide range regret of WRHEDGE (Lemma A.2), which when plugged into the aforementioned regret decomposition yields the improved dependence in $(X_i A_i^{K\wedge H})$ over the analysis of Morrill et al. [34] (Lemma F.1 & F.2), and also the $X_i A_i$ factor improvement over the results of [11, 20] in the special case of $K = 1$. The full proof can be found in Appendix F.

## 5 Learning $K$-EFCE from bandit feedback

We now present Balanced $K$-EFR, a sample-based variant of $K$-EFR that achieves a sharp sample complexity in the more challenging bandit feedback setting. Our algorithm relies on the following *balanced exploration policy* [19, 5]. Recall that $|\mathcal{C}_h(x_{i,h'}, a_{i,h'})|$ is the number of descendants of $(x_{i,h'}, a_{i,h'})$ within the $h$-th layer of the $i^{\text{th}}$ player's game tree (cf. Section 2).

**Definition 4** (Balanced exploration policy). *For any $1 \leq h \leq H$, we define $\pi_i^{\star,h}$, the ($i^{\text{th}}$ player's) balanced exploration policy for layer $h$ as*

$$\pi_{i,h'}^{\star,h}(a_{h'}|x_{h'}) := |\mathcal{C}_h(x_{i,h'}, a_{i,h'})|/|\mathcal{C}_h(x_{i,h'})| \quad \text{for all } (x_{i,h'}, a_{i,h'}) \in \mathcal{X}_{i,h'} \times \mathcal{A}_i, \ h' \leq h-1,$$

*and $\pi_{i,h'}^{\star,h}(a_{i,h'}|x_{i,h'}) := 1/A_i$ for $h' \geq h$.*

Note that there are $H$ such policies, one for each layer $h$. We remark that the construction of $\pi_i^{\star,h}$ requires knowledge about the descendant relationships among the $i^{\text{th}}$ player's infosets, which is a mild requirement (e.g. can be efficiently obtained from one traversal of the $i^{\text{th}}$ player's game tree; see Appendix H.2 for a detailed discussion about this requirement).

**Algorithm description (sampling part)** Our Balanced $K$-EFR (deferred to Algorithm 7) builds upon the full feedback version of $K$-EFR (Algorithm 2). The main new ingredient within Algorithm 7 is to use sample-based loss estimators obtained by two *balanced sampling* algorithms (Algorithm 3 & 6), one for each type of rechistories. Here we present the sampling algorithm for Type-II rechistories in Algorithm 3; The sampling algorithm for Type-I rechistories (Algorithm 6) is designed similarly and deferred to Appendix G.1 due to space limit. Algorithm 3 performs two main steps:

- Line 1-3 (Sampling): Construct policies $\{\pi_i^{t,(h,h',W)}\}$ that are *interlaced concatenations* of the current $\pi_i^t$ and the balanced policy $\pi_i^{\star,h}$, and play one episode using each policy against $\pi_{-i}^t$.

- Line 7: Construct loss estimators $\{\widetilde{L}_{x_{i,h},b_{1:h'}}(a)\}_{x_{i,h},b_{1:h'},a}$ by (9), which for each $x_{i,h}$ and $b_{1:h'} \in \Omega_h^{(\mathrm{II}),K}(x_{i,h})$ is an unbiased estimator of counterfactual losses $\{L_h^t(x_{i,h},a)\}_{a \in \mathcal{A}_i}$ that will be used by Algorithm 7 to be fed into the regret minimization algorithm REGALG.

We remark that the sampling policies $\{\pi_i^{t,(h,h',W)}\}$ in Algorithm 3 are *interlaced concatenations* of $\pi_i^t$ and $\pi_i^{\star,h}$ along time steps $h$, where the policy to take at each $h$ is determined by $W$. These policies are generalizations of the sampling policies in the Balanced CFR algorithm of Bai et al. [5] (which can be thought of as a simple non-interlacing concatenation). They allow *time-selection aware sampling*: Each loss estimator $\widetilde{L}_{(x_{i,h},b_{1:h'})}^t(\cdot)$ achieves low variance relative to the corresponding time selection function $S_{b_{1:h'}}^t$. Further, there is an *efficient sharing* of sampling policies, as here roughly $\binom{H}{K \wedge H} X_i A_i^{K \wedge H}$ loss estimators (one for each $(x_{i,h}, b_{1:h'})$) are constructed using only (a much lower number of) $H\binom{H}{K \wedge H}$ policies.

**Stochastic wide-range regret minimization**  Algorithm 7 requires the wide-range regret minimization algorithm REGALG to additionally handle the stochastic setting, i.e. minimize the wide-range regret (e.g. (7)) when fed with our sample-based loss estimators. Here, we instantiate REGALG to be SWRHEDGE (Algorithm 5), a stochastic variant of WRHEDGE, with hyperparameters

$$\eta_{x_{i,h}} = \sqrt{\binom{H}{K \wedge H} X_i A_i^{K \wedge H+1} \log(8 \textstyle\sum_{i \in [m]} X_i A_i/p)/(H^3 T)}, \quad \overline{L} = H. \tag{10}$$

SWRHEDGE is a non-trivial extension of WRHEDGE to the stochastic setting, as in each round it admits *multiple* sample-based loss estimators, one for each time selection function, with the same mean (cf. Line 8). This is needed since Algorithm 3 uses different sampling policies to construct the loss estimator $\widetilde{L}_{x_{i,h},b_{1:h'}}^t(\cdot)$ for each $b_{1:h'} \in \Omega_i^{(\mathrm{II}),K}(x_{i,h})$ (cf. (9)).

**Theoretical guarantee**  We now present our main result for the bandit feedback setting.

**Theorem 5** (Learning $K$-EFCE from bandit feedback). *For any $0 \le K \le \infty$, $\varepsilon \in (0, H]$ and $p \in [0, 1)$, letting all players run Algorithm 7 together in a self-play fashion for $T$ iterations, with REGALG instantiated as SWRHEDGE (Algorithm 5) with hyperparameters in (10). Let $\pi^t = \{\pi_i^t\}_{i \in [m]}$ denote the joint policy of all players at the $t$'th iteration. Then, with probability at least $1 - p$, the correlated policy $\overline{\pi} = \mathrm{Unif}(\{\pi^t\}_{t=1}^T)$ satisfies $K$-EFCEGap$(\overline{\pi}) \le \varepsilon$, as long as $T \ge \mathcal{O}(H^3\binom{H}{K \wedge H}(\max_{i \in [m]} X_i A_i^{K \wedge H+1})\iota/\varepsilon^2)$. The total number of episodes played is*

$$3mH\binom{H}{K \wedge H} \cdot T = \mathcal{O}\Big(m\binom{H}{K \wedge H}^2 \big(\max_{i \in [m]} X_i A_i^{K \wedge H+1}\big)\iota/\varepsilon^2\Big),$$

*where $\iota = \log(8 \sum_{i \in [m]} X_i A_i/p)$ is a log factor and $\mathcal{O}(\cdot)$ hides $\mathrm{poly}(H)$ factors.*

To our best knowledge, Theorem 5 provides the first result for learning $K$-EFCE under bandit feedback. The sample complexity $\widetilde{\mathcal{O}}(\binom{H}{K \wedge H}^2 \max_{i \in [m]}(X_i A_i^{K \wedge H+1})/\varepsilon^2)$ (ignoring $m$, $H$ factors) has only an $\binom{H}{K \wedge H} A_i$ additional factor over the iteration complexity in the full feedback setting (Theorem 3), which is natural—The $\binom{H}{K \wedge H}$ comes from the number of episodes sampled within each iteration (Lemma G.1), and the $A_i$ arises from estimating loss vectors from bandit feedback. In particular, the special case of $K = 1$ provides the first result for learning EFCEs from bandit feedback, with sample complexity $\widetilde{\mathcal{O}}(\max_{i \in [m]} X_i A_i^2/\varepsilon^2)$. We remark that the linear in $X_i$ dependence at all $K \ge 0$ is optimal, as the sample complexity lower bound for the $K = 0$ case (learning NFCCEs) is already $\Omega(\max_{i \in [m]} X_i A_i/\varepsilon^2)$ [5][7]. Also, the policies $\{\pi_i^t\}_{t=1}^T$ maintained in Algorithm 7 also achieves sublinear $K$-EFCE regret. However, strictly speaking, this is not a regret bound of our algorithm, as the sampling policies $\pi_i^{t,(h,h',W)}$ actually used are not $\pi_i^t$.

**Proof overview**  The proof of Theorem 5 builds on the analysis in the full-feedback case, and further relies on several new techniques in order to achieve the sharp linear in $\max_{i \in [m]} X_i$ sample

---

[7]The sample complexity lower bound in [5] is stated for learning Nash Equilibria in two-player zero-sum IIEFGs, but can be directly extended to learning NFCCEs in multi-player general-sum IIEFGs.

complexity: (1) A regret bound for the SWRHEDGE algorithm under the same-mean condition (Lemma A.3), which may be of independent interest; (2) Crucial use of the *balancing property* of $\pi_i^{\star,h}$ (Lemma B.4) to control the variance of the loss estimators $\widetilde{L}_{(x_{i,h}, b_{1:h'})}^t(\cdot)$, which in turn produces sharp bounds on the regret terms and additional concentration terms (Lemma G.4-G.9). The full proof can be found in Appendix G.3.

## 6 Conclusion

This paper proposes $K$-EFCE, a generalized definition of Extensive-Form Correlated Equilibria in Imperfect-Information Games, and designs sharp algorithms for computing $K$-EFCE under full feedback and learning a $K$-EFCE under bandit feedback. Our algorithms perform wide-range regret minimization over each infoset to minimize the overall $K$-EFCE regret, and introduce new efficient sampling policies to handle bandit feedback. We believe our work opens up many future directions, such as accelerated techniques for computing $K$-EFCE from full feedback, learning other notions of equilibria from bandit feedback, as well as empirical investigations of our algorithms.

## Acknowledgment

The authors thank Brian Hu Zhang for discussions regarding the relationship between the $\infty$-EFCE and the BCE. S.M. is supported by NSF grant DMS-2210827.

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
