# Appendix

## Table of Contents

# A  Technical tools

## A.1  Technical lemmas

The following Freedman's inequality can be found in [1, Lemma 9].

**Lemma A.1** (Freedman's inequality). *Suppose random variables $\{X_t\}_{t=1}^T$ is a martingale difference sequence, i.e. $X_t \in \mathcal{F}_t$ where $\{\mathcal{F}_t\}_{t\geq 1}$ is a filtration, and $\mathbb{E}[X_t|\mathcal{F}_{t-1}] = 0$. Suppose $X_t \leq R$ almost surely for some (non-random) $R > 0$. Then for any $\lambda \in (0, 1/R]$, we have with probability at least $1 - \delta$ that*

$$\sum_{t=1}^T X_t \leq \lambda \cdot \sum_{t=1}^T \mathbb{E}\left[X_t^2|\mathcal{F}_{t-1}\right] + \frac{\log(1/\delta)}{\lambda}.$$

## A.2  Wide-range regret minimization

**Time selection functions**  For each element $b$ in a finite set $\mathcal{B}$, we let $\{S_b^t\}_{t\geq 1} \subseteq [0, 1]$ represent the time selection functions (viewing it as a function of $t$). We let $\mathcal{B}$ be the non-intersection union of $\mathcal{B}^s$ and $\mathcal{B}^e$, where $\mathcal{B}^s$ is called the set of swap time selection indexes and $\mathcal{B}^e$ is called the set of external time selection indexes.

**Strategy modifications**  We denote the set of swap modification function set $\Psi^s = \{\psi : [A] \rightarrow [A]\}$ and external modification function set $\Psi^e = \{\psi : [A] \rightarrow [A] : \exists a \in [A], \text{s.t. } \psi(b) = a, \forall b \in [A]\}$. Given a modification $\psi : [A] \rightarrow [A]$ and a strategy $p^t \in \Delta([A])$, we denote $\psi \diamond p^t \in \Delta([A])$ to be the modified strategy with $(\psi \diamond p^t)(a) = \sum_{b \in \psi^{-1}(a)} p^t(b)$. Given a modification $\psi : [A] \rightarrow [A]$, we define $M_\psi \in \{0, 1\}^{A \times A}$ to be its associated matrix, with its $(b, \psi(b))$'th element $M_\psi(b, \psi(b)) = 1$ for every $b \in [A]$, and otherwise equal to 0.

**Interaction protocal and learning goals**  We consider the following interaction protocol: at each iteration $t$, the learner receives the time selection set $\{S_b^t\}_{b \in \mathcal{B}} \subseteq [0, 1]$, outputs a vector $p^t \in \Delta([A])$, and receives the loss $\ell^t \in [0, \infty]^A$. The goal of the learner is to control the regret $\sum_{t=1}^T S_b^t(\langle p^t, \ell^t \rangle - \langle \psi \diamond p^t, \ell^t \rangle)$ for each pair $(b, \psi) \in \mathcal{I} := (\mathcal{B}^s \times \Psi^s) \cup (\mathcal{B}^e \times \Psi^e)$.

Specializing the results in [27], we design the WRHEDGE algorithm (Algorithm 4) achieving our goal. The regret bound is presented in the lemma below, whose proof is based on [27], with a slight modification on the initial weights and with a refined analysis.

---

**Algorithm 4** Wide-Range Regret Minimization with Hedge (WRHEDGE)

---

**Input:** Learning rate $\eta > 0$. Swap index set $\mathcal{B}^s$ and external index set $\mathcal{B}^e$; Swap strategy modification set $\Psi^s$ and external strategy modification set $\Psi^e$.

1: Let $\mathcal{I} := (\mathcal{B}^s \times \Psi^s) \cup (\mathcal{B}^e \times \Psi^e)$. Initialize $S_b^0 \leftarrow 0$ for all $b \in \mathcal{B}^s \cup \mathcal{B}^e$, and

$$q^0(b, \psi) \leftarrow \frac{|\Psi^s|\mathbf{1}\{b \in \mathcal{B}^e\} + |\Psi^e|\mathbf{1}\{b \in \mathcal{B}^s\}}{\sum_{(b', \psi') \in \mathcal{I}} [|\Psi^s|\mathbf{1}\{b' \in \mathcal{B}^e\} + |\Psi^e|\mathbf{1}\{b' \in \mathcal{B}^s\}]} \qquad (11)$$

for all $(b, \psi) \in \mathcal{I}$.

2: **for** iteration $t = 1, \ldots, T$ **do**
3:   (OBSERVE_TIMESELECTION) Receive time selection functions $\{S_b^t\}_{b \in \mathcal{B}^s \cup \mathcal{B}^e}$.
4:   Update distribution over $(b, \psi) \in \mathcal{I}$:

$$q^t(b, \psi) \propto q^{t-1}(b, \psi) \exp\left\{\eta \exp(-\eta\|\ell^{t-1}\|_\infty) S_b^{t-1}\langle p^{t-1}, \ell^{t-1} \rangle - \eta S_b^{t-1}\langle \psi \diamond p^{t-1}, \ell^{t-1} \rangle\right\}.$$

5:   Set $p^t \in \Delta([A])$ as a solution to the equation $p^{t\top} = p^{t\top}\left(\frac{\sum_{(b, \psi) \in \mathcal{I}} S_b^t q^t(b, \psi) M_\psi}{\sum_{(b, \psi) \in \mathcal{I}} S_b^t q^t(b, \psi)}\right)$.
6:   **if** RECOMMEND is called **then**
7:     Output the vector $p^t$.
8:   (OBSERVE_LOSS) Receive loss vector $\ell^t \in \mathbb{R}^A$.

---

**Lemma A.2** (Wide-range regret bound of WRHEDGE). *Let $\{\ell^t(a)\}_{a\in[A],t\in[T]}$ and $\{S^t_b\}_{b\in\mathcal{B}^s\cup\mathcal{B}^e,t\in[T]}$ be arbitrary arrays of loss functions and time selection functions. Assume that $S^t_b\in[0,1]$ and $\ell^t\in[0,\infty]^A$ for any $b\in\mathcal{B}^s\cup\mathcal{B}^e$, $a\in[A]$ and $t\in[T]$. Let $p^t$ be given as in Algorithm 4 with learning rate $\eta\in(0,\infty)$. Then we have*

$$\sup_{\psi\in\Psi^s}\sum_{t=1}^T S^t_b\Big(\langle p^t,\ell^t\rangle - \langle\psi\diamond p^t,\ell^t\rangle\Big) \leq \sum_{t=1}^T \eta\|\ell^t\|_\infty S^t_b\langle p^t,\ell^t\rangle + \frac{\log[(|\mathcal{B}^s|+|\mathcal{B}^e|)|\Psi^s|]}{\eta}, \quad \forall b\in\mathcal{B}^s,$$

$$\sup_{\psi\in\Psi^e}\sum_{t=1}^T S^t_b\Big(\langle p^t,\ell^t\rangle - \langle\psi\diamond p^t,\ell^t\rangle\Big) \leq \sum_{t=1}^T \eta\|\ell^t\|_\infty S^t_b\langle p^t,\ell^t\rangle + \frac{\log[(|\mathcal{B}^s|+|\mathcal{B}^e|)|\Psi^e|]}{\eta}, \quad \forall b\in\mathcal{B}^e.$$

*Proof.* For $(b,\psi)\in\mathcal{I}$, we define the cumulative loss w.r.t. $\{S^t_b\}_{t\geq 1}$ till time $t$ as

$$L^t_b := \sum_{t'=1}^t S^{t'}_b \left\langle p^{t'},\ell^{t'}\right\rangle,$$

and define the cumulative loss w.r.t. $(\{S^t_b\}_{t\geq 1},\psi)$ till time $t$ as

$$L^t(b,\psi) := \sum_{t'=1}^t S^{t'}_b \left\langle \psi\diamond p^{t'},\ell^{t'}\right\rangle.$$

We further define the weight of $(\{S^t_b\}_{t\geq 1},\psi)$ at the end of time $t$ as

$$w^t(b,\psi) := (|\Psi^s|\mathbf{1}\{b\in\mathcal{B}^e\}+|\Psi^e|\mathbf{1}\{b\in\mathcal{B}^s\})\exp\left\{\eta\sum_{t'=1}^t \exp(-\eta\|\ell^{t'}\|_\infty)S^{t'}_b\left\langle p^{t'},\ell^{t'}\right\rangle - \eta L^t(b,\psi)\right\},$$

and hence $w^0(b,\psi)$ is given by $|\Psi^s|\mathbf{1}\{b\in\mathcal{B}^e\} + |\Psi^e|\mathbf{1}\{b\in\mathcal{B}^s\}$. We further let $W^t := \sum_{(b,\psi)\in\mathcal{I}} w^t(b,\psi)$. Then the quantity $q^t(b,\psi)$ in Algorithm 4 is simply equal to $w^{t-1}(b,\psi)/W^{t-1}$.

We next show that $W^t \leq W^{t-1}$ for all $t \geq 1$. In fact, by $\exp(-\eta x) \leq 1 - (1 - \exp(-\eta\|\ell^t\|_\infty))x/\|\ell^t\|_\infty$ and $\exp(\eta x) \leq 1 + (\exp(\eta\|\ell^t\|_\infty) - 1)x/\|\ell^t\|_\infty$ for any $\eta \in (0,\infty)$ and $x\in[0,\|\ell^t\|_\infty]$, we have

$$W^t = \sum_{(b,\psi)\in\mathcal{I}} w^t(b,\psi) = \sum_{(b,\psi)\in\mathcal{I}} w^{t-1}(b,\psi)\exp\left\{\eta S^t_b\left\langle\exp(-\eta\|\ell^t\|_\infty)p^t - \psi\diamond p^t,\ell^t\right\rangle\right\}$$

$$\leq \sum_{(b,\psi)\in\mathcal{I}} w^{t-1}(b,\psi)\left(1 - \frac{(1-\exp(-\eta\|\ell^t\|_\infty))S^t_b}{\|\ell^t\|_\infty}\left\langle\psi\diamond p^t,\ell^t\right\rangle\right)\cdot\left(1 + \frac{(1-\exp(-\eta\|\ell^t\|_\infty))S^t_b}{\|\ell^t\|_\infty}\left\langle p^t,\ell^t\right\rangle\right)$$

$$\overset{(i)}{\leq} W^{t-1} - \frac{1-\exp(-\eta\|\ell^t\|_\infty)}{\|\ell^t\|_\infty}W^{t-1}\sum_{(b,\psi)\in\mathcal{I}} q^t(b,\psi)S^t_b\left\langle\psi\diamond p^t,\ell^t\right\rangle$$

$$+ \frac{1-\exp(-\eta\|\ell^t\|_\infty)}{\|\ell^t\|_\infty}W^{t-1}\sum_{(b,\psi)\in\mathcal{I}} q^t(b,\psi)S^t_b\left\langle p^t,\ell^t\right\rangle$$

$$\overset{(ii)}{=} W^{t-1}.$$

Here, (i) follows from $q^t(b,\psi) = w^{t-1}(b,\psi)/W^{t-1}$ and $\ell^t\in[0,1]^A$; (ii) uses the fact that $p^t$ solves

$$p^{t\top} = p^{t\top}\left(\frac{\sum_{(b,\psi)\in\mathcal{I}} S^t_b q^t(b,\psi)M_\psi}{\sum_{(b,\psi)\in\mathcal{I}} S^t_b q^t(b,\psi)}\right)$$

in line 5 in the algorithm, which gives

$$\sum_{(b,\psi)\in\mathcal{I}} q^t(b,\psi)S^t_b p^t = \sum_{(b,\psi)\in\mathcal{I}} q^t(b,\psi)S^t_b(\psi\diamond p^t).$$

This proves that $W^t \leq W^{t-1}$ for all $t \geq 1$.

Therefore, $W^t$ is non-increasing in $t$, and thus for all $(b, \psi) \in \mathcal{I}$,

$$\left(|\Psi^s|\mathbf{1}\{b \in \mathcal{B}^e\} + |\Psi^e|\mathbf{1}\{b \in \mathcal{B}^s\}\right)\exp\left\{\eta\sum_{t'=1}^{t}\exp(-\eta\|\ell^{t'}\|_\infty)S_b^{t'}\left\langle p^{t'}, \ell^{t'}\right\rangle - \eta L^t(b, \psi)\right\}$$

$$= w^t(b, \psi) \leq \sum_{(b', \psi') \in \mathcal{I}} w^0(b', \psi') = |\Psi^s||\Psi^e|(|\mathcal{B}^e| + |\mathcal{B}^s|),$$

which gives

$$\sum_{t'=1}^{t}\exp(-\eta\|\ell^{t'}\|_\infty)S_b^{t'}\left\langle p^{t'}, \ell^{t'}\right\rangle - L^t(b, \psi) \leq \frac{\log[|\Psi^s|(|\mathcal{B}^e| + |\mathcal{B}^s|)]}{\eta}, \quad \forall b \in \mathcal{B}^s,$$

$$\sum_{t'=1}^{t}\exp(-\eta\|\ell^{t'}\|_\infty)S_b^{t'}\left\langle p^{t'}, \ell^{t'}\right\rangle - L^t(b, \psi) \leq \frac{\log[|\Psi^e|(|\mathcal{B}^e| + |\mathcal{B}^s|)]}{\eta}, \quad \forall b \in \mathcal{B}^e.$$

Note that we have $1 \leq \exp\left(-\eta\|\ell^t\|_\infty\right) + \eta\|\ell^t\|_\infty$. So we can get that, for any $b \in \mathcal{B}^s$,

$$L^t(b) - L^t(b, \psi) = \sum_{t=1}^{T}S_b^t\left\langle p^t, \ell^t\right\rangle - L^t(b, \psi)$$

$$\leq \sum_{t=1}^{T}\exp(-\eta\|\ell^t\|_\infty)S_b^t\left\langle p^{t'}, \ell^{t'}\right\rangle - L^t(b, \psi) + \sum_{t=1}^{T}\eta\|\ell^t\|_\infty S_b^t\left\langle p^t, \ell^t\right\rangle$$

$$\leq \sum_{t=1}^{T}\eta\|\ell^t\|_\infty S_b^t\left\langle p^t, \ell^t\right\rangle + \frac{\log[|\Psi^s|(|\mathcal{B}^e| + |\mathcal{B}^s|)]}{\eta}.$$

Note that the left side is exactly $\sum_{t=1}^{T}S_b^t\left(\langle p^t, \ell^t\rangle - \langle\psi \diamond p^t, \ell^t\rangle\right)$. Consequently,

$$\sum_{t=1}^{T}S_b^t\left(\left\langle p^t, \ell^t\right\rangle - \left\langle\psi \diamond p^t, \ell^t\right\rangle\right) \leq \sum_{t=1}^{T}\eta\|\ell^t\|_\infty S_b^t\left\langle p^t, \ell^t\right\rangle + \frac{\log[|\Psi^s|(|\mathcal{B}^e| + |\mathcal{B}^s|)]}{\eta}, \quad \forall b \in \mathcal{B}^s.$$

We have similar results for $b \in \mathcal{B}^e$. Taking supremum over all $\psi \in \Psi^e$ or $\psi \in \Psi^s$ proves Lemma A.2.  $\square$

## A.3 Stochastic wide-range regret minimization

In this section, we consider a stochastic variant of wide-range regret minimization. More specifically, we consider the following interaction protocol: at each iteration $t$, the learner receives the time selection set $\{S_b^t\}_{b \in \mathcal{B}} \subseteq [0, 1]$, outputs a vector $p^t \in \Delta([A])$, and receives the unbiased stochastic loss $\widetilde{\ell}_b^t$ for each $b \in \mathcal{B}^e \cup \mathcal{B}^s$, with $\mathbb{E}[\widetilde{\ell}_b^t|\mathcal{F}_{t-1}] = \ell^t$ (the expected loss is independent of $b$). Here, the $\sigma$-field $\mathcal{F}_{t-1}$ is generated by all the random variables before the $t$-th round. The goal of the learner is still to control the regret $\sum_{t=1}^{T}S_b^t(\langle p^t, \widetilde{\ell}_b^t\rangle - \langle\psi \diamond p^t, \widetilde{\ell}_b^t\rangle)$ for each pair $(b, \psi) \in \mathcal{I} := (\mathcal{B}^s \times \Psi^s) \cup (\mathcal{B}^e \times \Psi^e)$.

Our algorithm Stochastic Wide-Range Regret Minimization with Hedge (SWRHEDGE) is given in Algorithm 5. The regret bound for SWRHEDGE is given by the lemma below.

**Lemma A.3** (Wide-range regret bound for SWRHEDGE). *Let $\{\ell^t\}_{t \in [T]}$, $\{M_b^t\}_{b \in \mathcal{B}^s \cup \mathcal{B}^e, t \in [T]}$, and $\{w_b\}_{b \in \mathcal{B}^s \cup \mathcal{B}^e}$ be arbitrary arrays of loss functions, rescaled time selection functions, and weighting functions. Let $0 < \overline{L} < \infty$ be a parameter (that will serve as an upper bound of all $w_b M_b^t\|\widetilde{\ell}_b^t\|_\infty$). Assume that (i) $M_b^t \geq 0$, $w_b > 0$, and $\widetilde{\ell}_b^t \in [0, \overline{L}/(w_b M_b^t)]^A$ for any $b \in \mathcal{B}^s \cup \mathcal{B}^e$, $a \in [A]$ and $t \in [T]$; (ii) $\mathbb{E}\left[\widetilde{\ell}_b^t|\mathcal{F}_{t-1}\right] = \ell^t$ for all $b \in \mathcal{B}^s \cup \mathcal{B}^e$. Let $p^t$ be given as in Algorithm 5 with learning rate $\eta \in (0, \infty)$ and time selection function $\{S_b^t\}_{b \in \mathcal{B}^s \cup \mathcal{B}^e} = \{w_b M_b^t\}_{b \in \mathcal{B}^s \cup \mathcal{B}^e}$. Then with probability at least $1 - p$, we have*

$$\sup_{\psi \in \Psi^s}\sum_{t=1}^{T}M_b^t\left(\left\langle p^t, \widetilde{\ell}_b^t\right\rangle - \left\langle\psi \diamond p^t, \widetilde{\ell}_b^t\right\rangle\right) \leq \sum_{t=1}^{T}\eta\overline{L}M_b^t\left\langle p^t, \widetilde{\ell}_b^t\right\rangle + \frac{\log[(|\mathcal{B}^s| + |\mathcal{B}^e|)|\Psi^s|/p]}{\eta w_b}, \quad \forall b \in \mathcal{B}^s,$$

$$\sup_{\psi \in \Psi^e}\sum_{t=1}^{T}M_b^t\left(\left\langle p^t, \widetilde{\ell}_b^t\right\rangle - \left\langle\psi \diamond p^t, \widetilde{\ell}_b^t\right\rangle\right) \leq \sum_{t=1}^{T}\eta\overline{L}M_b^t\left\langle p^t, \widetilde{\ell}_b^t\right\rangle + \frac{\log[(|\mathcal{B}^s| + |\mathcal{B}^e|)|\Psi^e|/p]}{\eta w_b}, \quad \forall b \in \mathcal{B}^e.$$

**Algorithm 5** Stochastic Wide Range Regret Minimization with Hedge (SWRHEDGE)

---

**Input:** Learning rate $\eta > 0$ and parameter $\overline{L}$. Swap index set $\mathcal{B}^s$ and external index set $\mathcal{B}^e$; Swap strategy modification set $\Psi^s$ and external strategy modification set $\Psi^e$.

1: Let $\mathcal{I} := (\mathcal{B}^s \times \Psi^s) \cup (\mathcal{B}^e \times \Psi^e)$. Initialize $S_b^0 \leftarrow 0$ for all $b \in \mathcal{B}^s \cup \mathcal{B}^e$, and

$$q^0(b, \psi) \leftarrow \frac{|\Psi^s|\mathbf{1}\{b \in \mathcal{B}^e\} + |\Psi^e|\mathbf{1}\{b \in \mathcal{B}^s\}}{\sum_{(b',\psi')\in\mathcal{I}}[|\Psi^s|\mathbf{1}\{b' \in \mathcal{B}^e\} + |\Psi^e|\mathbf{1}\{b' \in \mathcal{B}^s\}]}$$

for all $(b, \psi) \in \mathcal{I}$.

2: **for** iteration $t = 1, \ldots, T$ **do**

3:    (OBSERVE_TIMESELECTION) Receive time selection functions $\{S_b^t\}_{b\in\mathcal{B}^s\cup\mathcal{B}^e}$.

4:    Update distribution over $(b, \psi) \in \mathcal{I}$:

$$q^t((b, \psi)) \propto q^{t-1}((b, \psi)) \exp\left\{\eta \exp(-\eta\overline{L})S_b^{t-1}\langle p^{t-1}, \widetilde{\ell}_b^{t-1}\rangle - \eta S_b^{t-1}\langle \psi \diamond p^{t-1}, \widetilde{\ell}_b^{t-1}\rangle\right\}.$$

5:    Set $p^t \in \Delta([A])$ as a solution to the equation $p^{t\top} = p^{t\top}\left(\frac{\sum_{(b,\psi)\in\mathcal{I}} S_b^t q^t(b,\psi) M_\psi}{\sum_{(b,\psi)\in\mathcal{I}} S_b^t q^t(b,\psi)}\right)$.

6:    **if** RECOMMEND is called **then**

7:      Output the vector $p^t$.

8:    (OBSERVE_LOSS) Receive loss vectors $\{\widetilde{\ell}_b^t\}_{b\in\mathcal{B}^s\cup\mathcal{B}^e}$ (where $\mathbb{E}\left(\widetilde{\ell}_b^t|\mathcal{F}_{t-1}\right) = \ell^t$ doesn't depend on $b$).

---

*Proof.* For $(b, \psi) \in \mathcal{I}$, we define the cumulative loss w.r.t. $\{S_b^t\}_{t\geq 1}$ till time $t$ as

$$L^t(b) := \sum_{t'=1}^t S_b^{t'}\left\langle p^{t'}, \widetilde{\ell}_b^{t'}\right\rangle,$$

and define the cumulative loss w.r.t. $(\{S_b^t\}_{t\geq 1}, \psi)$ till time $t$ as

$$L^t(b, \psi) := \sum_{t'=1}^t S_b^{t'}\left\langle \psi \diamond p^{t'}, \widetilde{\ell}_b^{t'}\right\rangle.$$

We further define the weight of $(\{S_b^t\}_{t\geq 1}, \psi)$ at the end of time $t$ as

$$w^t(b, \psi) := (|\Psi^s|\mathbf{1}\{b \in \mathcal{B}^e\} + |\Psi^e|\mathbf{1}\{b \in \mathcal{B}^s\}) \exp\left\{\eta \exp(-\eta\overline{L})L^t(b) - \eta L^t(b, \psi)\right\},$$

and hence $w^0(b, \psi)$ is given by $|\Psi^s|\mathbf{1}\{b \in \mathcal{B}^e\} + |\Psi^e|\mathbf{1}\{b \in \mathcal{B}^s\}$. We further let $W^t := \sum_{(b,\psi)\in\mathcal{I}} w^t(b, \psi)$. Then the quantity $q^t(b, \psi)$ in Algorithm 5 is simply equal to $w^{t-1}(b, \psi)/W^{t-1}$.

We next show that $\mathbb{E}[W^t|\mathcal{F}_{t-1}] \leq W^{t-1}$ for all $t \geq 1$. In fact, by $\exp(-\eta x) \leq 1 - (1 - \exp(-\eta\overline{L}))x/\overline{L}$ and $\exp(\eta x) \leq 1 + (\exp(\eta\overline{L}) - 1)x/\overline{L}$ for any $\eta \in (0, \infty]$ and $x \in [0, \overline{L}]$, we have

$$W^t = \sum_{(b,\psi)\in\mathcal{I}} w^t(b, \psi) = \sum_{(b,\psi)\in\mathcal{I}} w^{t-1}(b, \psi) \exp\left\{\eta S_b^t\left\langle \exp(-\eta\overline{L})p^t - \psi \diamond p^t, \widetilde{\ell}_b^t\right\rangle\right\}$$

$$\leq \sum_{(b,\psi)\in\mathcal{I}} w^{t-1}(b, \psi)\left(1 - \frac{(1 - \exp(-\eta\overline{L}))S_b^t}{\overline{L}}\left\langle \psi \diamond p^t, \widetilde{\ell}_b^t\right\rangle\right) \cdot \left(1 + \frac{(1 - \exp(-\eta\overline{L}))S_b^t}{\overline{L}}\left\langle p^t, \widetilde{\ell}_b^t\right\rangle\right)$$

$$\overset{(i)}{\leq} W^{t-1} - \frac{1 - \exp(-\eta\overline{L})}{\overline{L}}W^{t-1}\sum_{(b,\psi)\in\mathcal{I}} q^t(b, \psi)S_b^t\left\langle \psi \diamond p^t, \widetilde{\ell}_b^t\right\rangle$$

$$+ \frac{1 - \exp(-\eta\overline{L})}{\overline{L}}W^{t-1}\sum_{(b,\psi)\in\mathcal{I}} q^t(b, \psi)S_b^t\left\langle p^t, \widetilde{\ell}_b^t\right\rangle.$$

Here, (i) follows from $q^t(b, \psi) = w^{t-1}(b, \psi)/W^{t-1}$ and $S_b^t \|\widetilde{\ell}_b^t\|_\infty \leq \overline{L}$ for any $b$. Using the above inequality, $\mathbb{E}\left(\widetilde{\ell}_b^t | \mathcal{F}_{t-1}\right) = \ell^t$ for any $b$, and the fact that $p^t$ solves

$$p^{t\top} = p^{t\top} \left( \frac{\sum_{(b,\psi)\in\mathcal{I}} S_b^t q^t(b, \psi) M_\psi}{\sum_{(b,\psi)\in\mathcal{I}} S_b^t q^t(b, \psi)} \right)$$

in line 5 in the algorithm, which gives

$$\sum_{(b,\psi)\in\mathcal{I}} q^t(b, \psi) S_b^t p^t = \sum_{(b,\psi)\in\mathcal{I}} q^t(b, \psi) S_b^t(\psi \diamond p^t),$$

we have $\mathbb{E}[W^t | \mathcal{F}_{t-1}] \leq W^{t-1}$. Taking expectation and using the tower property of conditional expectation yields that

$$\mathbb{E}[W^t] \leq W^0.$$

Therefore, by Markov inequality, we have with probability at least $1 - p$ that

$$W^t \leq W^0/p.$$

On this event, we have for all $(b, \psi) \in \mathcal{I}$ that

$$(|\Psi^s| \mathbf{1}\{b \in \mathcal{B}^e\} + |\Psi^e| \mathbf{1}\{b \in \mathcal{B}^s\}) \exp\{\eta \exp(-\eta\overline{L}) L^t(b) - \eta L^t(b, \psi)\}$$
$$= w^t(b, \psi) \leq W^t \leq W^0/p \leq \sum_{(b',\psi')\in\mathcal{I}} w^0(b', \psi')/p = |\Psi^s||\Psi^e|(|\mathcal{B}^e| + |\mathcal{B}^s|)/p.$$

As a result,

$$\exp\{\eta \exp(-\eta\overline{L}) L^t(b) - \eta L^t(b, \psi)\} \leq |\Psi^s|(|\mathcal{B}^e| + |\mathcal{B}^s|)/p, \quad \forall b \in \mathcal{B}^s,$$
$$\exp\{\eta \exp(-\eta\overline{L}) L^t(b) - \eta L^t(b, \psi)\} \leq |\Psi^e|(|\mathcal{B}^e| + |\mathcal{B}^s|)/p, \quad \forall b \in \mathcal{B}^e.$$

Note that we have $1 \leq \exp(-\eta\overline{L}) + \eta\overline{L}$. So we can get that, for any $b \in \mathcal{B}^s$,

$$L^t(b) - L^t(b, \psi) \leq \exp(-\eta\overline{L}) L^t(b) - L^t(b, \psi) + \eta\overline{L} L^t(b)$$
$$\leq \frac{\log[|\Psi^s|(|\mathcal{B}^e| + |\mathcal{B}^s|)/p]}{\eta} + \eta\overline{L} L^t(b).$$

Note that the left side is exactly $\sum_{t=1}^T S_b^t \left( \langle p^t, \widetilde{\ell}_b^t \rangle - \langle \psi \diamond p^t, \widetilde{\ell}_b^t \rangle \right)$. Consequently,

$$\sum_{t=1}^T S_b^t \left( \langle p^t, \widetilde{\ell}_b^t \rangle - \langle \psi \diamond p^t, \widetilde{\ell}_b^t \rangle \right) \leq \sum_{t=1}^T \eta\overline{L} S_b^t \langle p^t, \widetilde{\ell}_b^t \rangle + \frac{\log[|\Psi^s|(|\mathcal{B}^e| + |\mathcal{B}^s|)/p]}{\eta}, \quad \forall b \in \mathcal{B}^s.$$

Because $S_b^t = M_b^t w_b$, dividing by $w_b$ gives that

$$\sum_{t=1}^T M_b^t \left( \langle p^t, \widetilde{\ell}_b^t \rangle - \langle \psi \diamond p^t, \widetilde{\ell}_b^t \rangle \right) \leq \sum_{t=1}^T \eta\overline{L} M_b^t \langle p^t, \widetilde{\ell}_b^t \rangle + \frac{\log[|\Psi^s|(|\mathcal{B}^e| + |\mathcal{B}^s|)/p]}{\eta w_b}, \quad \forall b \in \mathcal{B}^s.$$

We have similar results for $b \in \mathcal{B}^e$. Taking supreme over all $\psi \in \Psi^e$ or $\psi \in \Psi^s$ proves Lemma A.3. $\qquad\square$

# B   Properties of the game

## B.1   Basic properties

Given the sequence-form transitions $p_{1:h}$ as in Eq. (2) and the sequence-form policies of the opponents $\{\pi_{j,1:h}\}_{j\neq i}$ as in Eq. (3), we define the marginal reaching probability $p_{1:h}^{\pi_{-i}}(s_h)$ and $p_{1:h}^{\pi_{-i}}(x_{i,h})$ as follows:

$$p_{1:h}^{\pi_{-i}}(s_h) = p_{1:h}(s_h) \prod_{j\in[m],j\neq i} \pi_{j,1:h}(s_h, a_{j,h}), \tag{12}$$

$$p_{1:h}^{\pi_{-i}}(x_{i,h}) = \sum_{s_h \in x_{i,h}} p_{1:h}^{\pi_{-i}}(s_h). \tag{13}$$

The following three results give properties of the marginal reaching probability $p_{1:h}^{\pi_{-i}}(x_{i,h})$, and the counterfactual loss functions $L_{i,h}^t$.

**Lemma B.1** (Properties of $p_{1:h}^{\pi_{-i}}(x_h)$)**.** *The following holds for any $\pi_{-i} = \{\pi_j\}_{j \neq i} \in \otimes_{j \neq i} \Pi_j$:*

(a) *For any policy $\pi_i \in \Pi_i$, we have*

$$\sum_{(x_h, a_h) \in \mathcal{X}_{i,h} \times \mathcal{A}_i} \pi_{i,1:h}(x_h, a_h) p_{1:h}^{\pi_{-i}}(x_h) = 1.$$

(b) $0 \leq p_{1:h}^{\pi_{-i}}(x_h) \leq 1$ *for all $h \in [H], x_h \in \mathcal{X}_{i,h}$.*

*Proof.* For (a), notice that

$$\pi_{i,1:h}(x_h, a_h) p_{1:h}^{\pi_{-i}}(x_h) = \sum_{s_h \in x_h} p_{1:h}(s_h) \cdot \pi_{i,1:h}(x_h, a_h) \cdot \prod_{j \neq i} \pi_{j,1:h-1}(x_{j,h}(s_{h-1}), a_{j,h-1})$$

$$= \sum_{s_h \in x_h} \mathbb{P}^{\pi_i, \pi_{-i}}(\text{visit } (s_h, a_h)) = \mathbb{P}^{\pi_i, \pi_{-i}}(\text{visit } (x_h, a_h)).$$

Summing over all $(x_h, a_h) \in \mathcal{X}_{i,h} \times \mathcal{A}_i$, the right hand side sums to one, thereby showing (a).

For (b), fix any $x_h \in \mathcal{X}_{i,h}$. Clearly $p_{1:h}^{\pi_{-i}}(x_h) \geq 0$. Choose any $a_h \in \mathcal{A}_i$, and choose policy $\pi_i^{x_h, a_h} \in \Pi_i$ such that $\pi_{i,1:h}^{x_h, a_h}(x_h, a_h) = 1$ (such $\pi_i^{x_h, a_h}$ exists, for example, by deterministically taking all actions prescribed in infoset $x_h$ at all ancestors of $x_h$). For this $\pi_i^{x_h, a_h}$, using (a), we have

$$p_{1:h}^{\pi_{-i}}(x_h) = \pi_{i,1:h}^{x_h, a_h}(x_h, a_h) \cdot p_{1:h}^{\pi_{-i}}(x_h) \leq \sum_{(x_h', a_h') \in \mathcal{X}_{i,h} \times \mathcal{A}_i} \pi_{i,1:h}^{x_h, a_h}(x_h', a_h') \cdot p_{1:h}^{\pi_{-i}}(x_h') = 1.$$

This shows part (b). $\qquad\square$

**Corollary B.1.** *For any policy $\pi_i \in \Pi_i$ and $h \in [H]$, we have*

$$\sum_{(x_h, a_h) \in \mathcal{X}_{i,h} \times \mathcal{A}_i} \pi_{i,1:h}(x_h, a_h) \ell_{i,h}^t(x_h, a_h) \leq 1.$$

*Proof.* Notice by definition

$$\ell_h^t(x_h, a_h) = \sum_{s_h \in x_h, (a_{j,h})_{j \neq i} \in \otimes_{j \neq i} \mathcal{A}_j} p_{1:h}(s_h) \prod_{j \neq i} \pi_{j,1:h}^t(x_{j,h}(s_h), a_{j,h})(1 - r_h(s_h, \mathbf{a}_h)) \leq p_{1:h}^{\pi_{-i}}(x_h),$$

and the result is implied by Lemma B.1 (b). $\qquad\square$

**Lemma B.2.** *For any $h \in [H]$, the counterfactual loss function $L_{i,h}^t$ defined in (5) satisfies the bound*

(a) *For any policy $\pi_i \in \Pi_i$, we have*

$$\sum_{(x_h, a_h) \in \mathcal{X}_{i,h} \times \mathcal{A}_i} \pi_{i,1:h}(x_h, a_h) L_{i,h}^t(x_h, a_h) \leq H - h + 1.$$

(b) *For any $(h, x_h, a_h)$, we have*

$$0 \leq L_{i,h}^t(x_h, a_h) \leq p_{1:h}^{\pi_{-i}^t}(x_h) \cdot (H - h + 1).$$

*Proof.* Part (a) follows from the fact that

$$\sum_{(x_h, a_h) \in \mathcal{X}_{i,h} \times \mathcal{A}_i} \pi_{i,1:h}(x_h, a_h) L_{i,h}^t(x_h, a_h) = \mathbb{E}_{\pi_i, \pi_{-i}^t} \left[ \sum_{h'=h}^H r_{h'} \right] \leq H - h + 1,$$

where the first equality follows from the definition of the loss functions $\ell_h$ and $L_h$ in (4), (5).

For part (b), the nonnegativity follows clearly by definition. For the upper bound, take any policy $\pi_i^{x_h, a_h} \in \Pi_i$ such that $\pi_{i,1:h}^{x_h, a_h}(x_h, a_h) = 1$. We then have

$$L_{i,h}^t(x_h, a_h) = \pi_{i,1:h}^{x_h, a_h}(x_h, a_h) L_{i,h}^t(x_h, a_h) = \mathbb{E}_{\pi_i^{x_h, a_h}, \pi_{-i}^t} \left[ \mathbf{1}\{\text{visit } x_h, a_h\} \cdot \sum_{h'=h}^H r_{h'} \right]$$

$$= \mathbb{P}_{\pi_i^{x_h,a_h}, \pi_{-i}^t}(\text{visit } x_h, a_h) \cdot \mathbb{E}_{\pi_i^{x_h,a_h}, \pi_{-i}^t}\left[\sum_{h'=h}^H r_{h'} \middle| \text{visit } x_h, a_h\right]$$

$$\leq \pi_{i,1:h}^{x_h,a_h}(x_h, a_h) p_{1:h}^{\pi_{-i}^t}(x_h) \cdot (H - h + 1) = p_{1:h}^{\pi_{-i}^t}(x_h) \cdot (H - h + 1).$$

This proves the lemma. $\qquad\square$

## B.2  Balanced exploration policy

Here we collect properties of the balanced exploration policy $\pi_i^{\star,h}$ (cf. Definition 4). Most results below have appeared in [5, Appendix C.2] in the two-player zero-sum setting. Here we present them again in our setting of multi-player general-sum IIEFGs.

We begin by providing an interpretation of the balanced exploration policy $\pi_{i,1:h}^{\star,h}$: its inverse $1/\pi_{i,1:h}^{\star,h}$ can be viewed as the (product) of a "transition probability" over the game tree for the $i$'th player.

For any $1 \leq h \leq H$ and $1 \leq k \leq h-1$, define $p_{i,k}^{\star,h}(x_{k+1}|x_k, a_k) = |\mathcal{C}_h(x_{k+1})|/|\mathcal{C}_h(x_k, a_k)|$ (we use the convention that $|\mathcal{C}_h(x_h)| = 1$). By this definition, $p_{i,k}^{\star,h}(\cdot|x_k, a_k)$ is a probability distribution over $\mathcal{C}_h(x_k, a_k)$ and can be interpreted as a balanced transition probability from $(x_k, a_k)$ to $x_{k+1}$. The sequence-form of this balanced transition probability takes the form

$$p_{i,1:h}^{\star,h}(x_h) = \frac{|\mathcal{C}_h(x_1)|}{X_{i,h}} \prod_{k=1}^{h-1} p_{i,k}^{\star,h}(x_{k+1}|x_k, a_k) = \frac{|\mathcal{C}_h(x_1)|}{X_{i,h}} \prod_{k=1}^{h-1} \frac{|\mathcal{C}_h(x_{k+1})|}{|\mathcal{C}_h(x_k, a_k)|}. \tag{14}$$

**Lemma B.3.** *For any $(x_h, a_h) \in \mathcal{X}_{i,h} \times \mathcal{A}_i$, the sequence form of the transition $p_{i,1:h}^{\star,h}(x_h)$ and the sequence form of balanced exploration policy $\pi_{i,1:h}^{\star,h}(x_h, a_h)$ are related by*

$$p_{i,1:h}^{\star,h}(x_h) = \frac{1}{X_{i,h}A_i \cdot \pi_{i,1:h}^{\star,h}(x_h, a_h)}. \tag{15}$$

*Furthermore, for any $i$-th player's policy $\pi_i \in \Pi_i$ and any $h \in [H]$, we have*

$$\sum_{(x_h,a_h)\in\mathcal{X}_{i,h}\times\mathcal{A}_i} \pi_{i,1:h}(x_h, a_h) p_{i,1:h}^{\star,h}(x_h) = 1. \tag{16}$$

*Proof.* By the definition of the balanced transition probability as in Eq. (14) and the balanced exploration policy as in Definition 4, we have

$$\frac{1}{X_{i,h}A_i \cdot \pi_{i,1:h}^{\star,h}(x_h, a_h)} = \frac{1}{X_{i,h}A_i} \prod_{k=1}^{h-1} \frac{|\mathcal{C}_h(x_k)|}{|\mathcal{C}_h(x_k, a_k)|} \times A_i = \frac{|\mathcal{C}_h(x_1)|}{X_{i,h}} \prod_{k=1}^{h-1} \frac{|\mathcal{C}_h(x_{k+1})|}{|\mathcal{C}_h(x_k, a_k)|} = p_{i,1:h}^{\star,h}(x_h),$$

where the second equality used the property that $|\mathcal{C}_h(x_h)| = 1$. This proves Eq. (15). The proof of Eq. (16) is similar to the proof of Lemma B.1 (a). $\qquad\square$

**Lemma B.4** (Balancing property of $\pi_i^{\star,h}$)**.** *For any $i^{th}$ player's policy $\pi_i \in \Pi_i$ and any $h \in [H]$, we have*

$$\sum_{(x_h,a_h)\in\mathcal{X}_{i,h}\times\mathcal{A}_i} \frac{\pi_{i,1:h}(x_h, a_h)}{\pi_{i,1:h}^{\star,h}(x_h, a_h)} = X_{i,h}A_i.$$

*Proof.* Lemma B.4 follows as a direct consequence of Eq. (15) and (16) in Lemma B.3. $\qquad\square$

Lemma B.4 states that $\pi_i^{\star,h}$ is a good exploration policy in the sense that the distribution ratio between it and *any* $\pi_i \in \Pi_i$ has bounded $L_1$ norm. Further, the bound $X_{i,h}A_i$ is non-trivial—For example, if we replace $\pi_{i,1:h}^{\star,h}$ with the uniform policy $\pi_{i,1:h}^{\text{unif}}(x_h, a_h) = 1/A_i^h$, the left-hand side can be as large as $X_{i,h}A_i^h$ in the worst case.

## C Relationship between $K$-EFCE and exsiting equilibria

**Equivalence between 1-EFCE and trigger definition of EFCE**   At the special case $K = 1$, our (exact) 1-EFCE is equivalent to the existing definition of EFCE based on *trigger policies* [22, 11], which defines an $\varepsilon$-approximate EFCE as any correlated policy $\overline{\pi}$ such that the following trigger gap is at most $\varepsilon$:

$$\mathrm{TriggerGap}(\overline{\pi}) := \max_{i \in [m]} \max_{(x_i, a) \in \mathcal{X}_i \times \mathcal{A}_i} \max_{\widehat{\pi}_i \in \Pi_i} \left( \mathbb{E}_{\pi \sim \overline{\pi}} V_i^{\mathrm{trig}(\pi_i, \widehat{\pi}_i, (x_i, a)) \times \pi_{-i}} - \mathbb{E}_{\pi \sim \overline{\pi}} V_i^{\pi} \right) \leq \varepsilon. \quad (17)$$

Here the trigger policy $\mathrm{trig}(\pi_i, \widehat{\pi}_i, (x_i, a)) \in \Pi_i$ (with triggering sequence $(x_i, a)$) is the unique policy that plays $\pi_i \in \Pi_i$, unless infoset $x_i$ is visited and action $a$ is recommended, in which case the sequence $(x_i, a)$ is "triggered" and the player plays $\widehat{\pi}_i \in \Pi_i$ thereafter.

**Proposition C.1** (Equivalence of 1-EFCE and trigger definition). *For any correlated policy $\overline{\pi}$, we have*

$$\mathrm{TriggerGap}(\overline{\pi}) \leq \text{1-EFCEGap}(\overline{\pi}) \leq (\max_{i \in [m]} X_i A_i) \cdot \mathrm{TriggerGap}(\overline{\pi}),$$

*In particular,* $\text{1-EFCEGap}(\overline{\pi}) = 0$ *if and only if* $\mathrm{TriggerGap}(\overline{\pi}) = 0$.

The proof can be found in Appendix C.1. Proposition C.1 has two main implications: (1) An exact EFCE defined by the trigger gap is equivalent to an exact 1-EFCE (cf. Definition 2). Therefore the two definitions yields the same set of exact equilibria. (2) For $\varepsilon > 0$, $\text{1-EFCEGap}(\overline{\pi}) \leq \varepsilon$ implies $\mathrm{TriggerGap}(\overline{\pi}) \leq \varepsilon$, but the converse only holds with an extra $\max_{i \in [m]} X_i A_i$ factor, and thus 1-EFCEGap is a stricter metric for approximate equilibria than TriggerGap. This distinction is inherent instead of a proof artifact: Our 1-EFCE strategy modification (Algorithm 1) is able to *implement multiple trigger policies simultaneously*, as long as their triggering sequences are not ancestors or descendants of each other.

**Containment relationship**   We next show that $K$-EFCE are indeed stricter equilibria as $K$ increases, i.e. any (approximate) $(K + 1)$-EFCE is also an (approximate) $K$-EFCE, but not the converse. This justifies the necessity of considering $K$-EFCE for all values of $K \geq 1$ and shows that they are strict strengthenings of the 1-EFCE. Note that as we consider games with a finite horizon $H$, we have $K$-EFCEGap $=$ $H$-EFCEGap for all $K \geq H$ (including $K = \infty$). The proof of Proposition C.2 can be found in Appendix C.2.

**Proposition C.2** (Containment relationship). *For any correlated policy $\overline{\pi}$, we have*

$$0\text{-EFCEGap}(\overline{\pi}) \leq 1\text{-EFCEGap}(\overline{\pi}) \leq \cdots \leq K\text{-EFCEGap}(\overline{\pi}) \leq (K + 1)\text{-EFCEGap}(\overline{\pi})$$
$$\leq \cdots \leq \infty\text{-EFCEGap}(\overline{\pi}).$$

*In other words, $K$-EFCE are stricter equilibria as $K$ increases: Any $\varepsilon$-approximate $(K + 1)$-EFCE is also an $\varepsilon$-approximate $K$-EFCE for any $\varepsilon \geq 0$ and $K \geq 0$.*

*Moreover, the converse bounds do not hold, even if multiplicative factors are allowed: For any $0 \leq K < \infty$, there exists a game with $H = K + 1$ and a correlated policy $\overline{\pi}$ for which*

$$K\text{-EFCEGap}(\overline{\pi}) = 0 \quad \text{but} \quad (K + 1)\text{-EFCEGap}(\overline{\pi}) \geq 1/3 > 0.$$

**Relationship with other correlated equilibria**   The two endpoints $K = 0$ and $K = \infty$ of $K$-EFCE are closely related to other existing definitions of correlated equilibria in IIEFGs. Concretely, 0-EFCE is equivalent to Normal-Form Coarse Correlated Equilibria (NFCCE), whereas $\infty$-EFCE is equivalent to using the "Behavioral Correlated Equilibria" considered in [34], which is strictly weaker than Normal-Form Correlated Equilibria (NFCE) that is more computationally challenging to learn [17, 11].

In order to introduce the definition of NFCE, we reload the definition of a correlated policy to be a probability measure on all *pure product policies* instead of general product policies. We let $\Pi_i^{\mathrm{pure}}$ denote the set of all possible pure policies for player $i$. Note that this does not affect our definition of $K$-EFCE introduced in Section 3.

We first present the definitions of Normal-Form Correlated Equilibria (NFCE) and Normal-Form Coarse Correlated Equilibria (NFCCE) (from e.g. [17]). For consistency with our $K$-EFCE definition, we define both equilibria through defining their set of strategy modifications.

**Definition C.1** (NFCE strategy modification). *A NFCE strategy modification $\phi$ (for the $i^{th}$ player) is a mapping $\phi(\cdot, \cdot) : \mathcal{X}_i \times \Pi_i^{\text{pure}} \to \mathcal{A}_i$ . Let $\Phi_i^{\text{NFCE}}$ denote the set of all possible NFCE strategy modification for the $i^{th}$ player. For any $\phi \in \Phi_i^{\text{NFCE}}$, and any pure policy $\pi_i \in \Pi_i^{\text{pure}}$, we define the modified policy $\phi \diamond \pi_i$ as following: at infoset $x_{i,h}$, the modified policy $\phi \diamond \pi_i$ takes action $\phi(x_{i,h}, \pi_i)$.*

**Definition C.2** (NFCCE strategy modification). *A NFCCE strategy modification $\phi$ (for the $i^{th}$ player) is a mapping $\phi(\cdot) : \mathcal{X}_i \to \mathcal{A}_i$. Let $\Phi_i^{\text{NFCCE}}$ denote the set of all possible NFCCE strategy modification for the $i^{th}$ player. For any $\phi \in \Phi_i^{\text{NFCCE}}$, and any pure policy $\pi_i \in \Pi_i^{\text{pure}}$, we define the modified policy $\phi \diamond \pi_i$ as following: at infoset $x_{i,h}$, the modified policy $\phi \diamond \pi_i$ take action $\phi(x_{i,h})$.*

At a high level, NFCE has the "strongest" form of strategy modifications, which can observe the entire pure policy $\pi_i$ (i.e. full set of recommendations on every infoset). NFCCE has the "weakest" form of strategy modifications, which cannot observe any recommendation at all (so that each $\phi \in \Phi_i^{\text{NFCCE}}$ is equivalent to a pure policy in $\Pi_i^{\text{pure}}$).

**Definition C.3** (NFCE and NFCCE). *An $\varepsilon$-approximate {NFCE, NFCCE} of a POMG is a correlated policy $\overline{\pi}$ such that*

$$\{\text{NFCE}, \text{NFCCE}\}\text{Gap}(\overline{\pi}) := \max_{i \in [m]} \max_{\phi \in \Phi_i^{\{\text{NFCE}, \text{NFCCE}\}}} \left( \mathbb{E}_{\pi \sim \overline{\pi}} V_i^{(\phi \diamond \pi_i) \times \pi_{-i}} - \mathbb{E}_{\pi \sim \overline{\pi}} V_i^{\pi} \right) \leq \varepsilon.$$

*We say $\overline{\pi}$ is an (exact) {NFCE, NFCCE} if the above holds with $\varepsilon = 0$.*

**Proposition C.3** (Relationship between $K$-EFCE and NFCE, NFCCE). *For any correlated policy $\overline{\pi}$, we have*

  (a) $\infty$-EFCEGap($\overline{\pi}$) $\leq$ NFCEGap($\overline{\pi}$), *i.e.* NFCE *is stricter than* $\infty$-EFCE *(NFCEGap($\overline{\pi}$) $\leq \varepsilon$ implies $\infty$-EFCEGap($\overline{\pi}$) $\leq \varepsilon$).*

  *Further, the converse bound does not hold even if multiplicative factors are allowed: there exists a game with $H = 2$ and a correlated policy $\overline{\pi}$ for which*

$$\infty\text{-EFCEGap}(\overline{\pi}) = 0 \quad \text{but} \quad \text{NFCEGap}(\overline{\pi}) > 1/20.$$

  (b) $0$-EFCEGap($\overline{\pi}$) = NFCCEGap($\overline{\pi}$), *i.e.* $0$-EFCE *is equivalent to* NFCCE.

The proof can be found in Section C.3.

**Equivalence between $\infty$-EFCE and BCE deviations** Next, we give an (informal) argument of the equivalence between "behavioral deviations" considered in [34] and our $\infty$-EFCE strategy modifications.

A "behavioral deviation" $\phi$ for one player states that at each infoset, the player can choose from three options: (i) follow the recommendation action, (ii) choose a action without ever seeing the recommendation action, or (iii) choose an action after seeing the recommendation action. Further, the choice of these three options as well as the action to deviate to may depend on the infoset as well as the recommendation history. This is exactly equivalent to the $\infty$-EFCE strategy modification defined in Definition 1.

We remark though, despite the equivalence between the strategy modifications of $\infty$-EFCE and BCE, the resulting equilibria defined as the BCE in Morrill et al. [34] is slightly stricter than the $\infty$-EFCE—The definition of Morrill et al. [34] requires a BCE $\pi$ to satisfy that $\pi_i$ does not gain in game value from all the above deviation functions in not only the full game, but also in certain subgames induced by $\pi_{-i}$; by contrast, our $\infty$-EFCE only requires such a property in the full game.

## C.1 Proof of Proposition C.1

*Proof.* It suffices to consider all trigger policy $\text{trig}(\pi_i, \widehat{\pi}_i, (x_i, a))$ where $\widehat{\pi}_i$ is a pure policy (at each infoset $x_{h'}$, $\widehat{\pi}_i$ chooses action $\widehat{\pi}_i(x_{h'})$ deterministically.). We prove the two claims separately.

**Step 1.** We first show that $\text{TriggerGap}(\overline{\pi}) \leq 1\text{-EFCEGap}(\overline{\pi})$ for any correlated policy $\overline{\pi}$. We first claim that, for any trigger policy $\text{trig}(\pi_i, \widehat{\pi}_i, (x_{i,h}^{\star}, a^{\star}))$ where $\pi_i, \widehat{\pi}_i \in \Pi_i$, $x_{i,h}^{\star} \in \mathcal{X}_{i,h}$, and $a^{\star} \in \mathcal{A}_i$, there exists an $1$-EFCE strategy modification $\phi^{\star} \in \Phi_i^1$ such that, for any opponent's policy $\pi_{-i} \in \Pi_{-i}$, we have

$$V_i^{\text{trig}(\pi_i, \widehat{\pi}_i, (x_{i,h}^{\star}, a^{\star})) \times \pi_{-i}} = V_i^{(\phi^{\star} \diamond \pi_i) \times \pi_{-i}}.$$

Given this claim, for any correlated policy $\overline{\pi}$, we have as desired

$$\mathrm{TriggerGap}(\overline{\pi}) = \max_{i \in [m]} \max_{(x_i, a) \in \mathcal{X}_i \times \mathcal{A}_i} \max_{\widehat{\pi}_i \in \Pi_i} \left( \mathbb{E}_{\pi \sim \overline{\pi}} V_i^{\mathsf{trig}(\pi_i, \widehat{\pi}_i, (x_i, a)) \times \pi_{-i}} - \mathbb{E}_{\pi \sim \overline{\pi}} V_i^{\pi} \right)$$

$$\leq \max_{i \in [m]} \max_{\phi \in \Phi_i^1} \left( \mathbb{E}_{\pi \sim \overline{\pi}} V_i^{(\phi \diamond \pi_i) \times \pi_{-i}} - \mathbb{E}_{\pi \sim \overline{\pi}} V_i^{\pi} \right) = \text{1-EFCEGap}(\overline{\pi}).$$

To prove such a claim, we can choose the 1-EFCE strategy modification $\phi^\star$ to be the following: (1) At any Type-I rechistory with infoset $x = x_{i,h}^\star$, $\phi^\star$ swaps $a_\star$ to $\widehat{\pi}_i(x_{i,h}^\star)$ and swaps $a$ to $a$ (i.e., keep it unchanged) for any $a \neq a_\star$; (2) At any Type-I rechistory with infoset $x$ such that $x \neq x_{i,h}^\star$ and $x \not\succ (x_{i,h}^\star, a^\star)$, $\phi^\star$ does not swap the recommended action (swap the recommended action to itself); (3) At any Type-I rechistory and Type-II rechistory with infoset $x \succ (x_{i,h}^\star, a^\star)$, $\phi^\star$ chooses action $\widehat{\pi}_i(x)$ (no matter seeing recommendation or not); (4) For any rechistory that does not fall into the above categories, $\phi^\star$ can be arbitrarily defined since those rechistories will not be encountered by the design of $\phi^\star$ as above. It is easy to see that such an 1-EFCE strategy modification $\phi^\star$ applied on any $\pi_i$ implements the trigger policy $\mathsf{trig}(\pi_i, \widehat{\pi}_i, (x_{i,h}^\star, a^\star))$ so that their value functions are equal. This proves the claim.

**Step 2.** We next show that $\text{1-EFCEGap}(\overline{\pi}) \leq \max_{i \in [m]} X_i A_i \cdot \mathrm{TriggerGap}(\overline{\pi})$ for any correlated policy $\overline{\pi} \in \Delta(\Pi)$. For any 1-EFCE strategy modification $\phi \in \Phi_i^1$ and any $\pi_i \in \Pi_i$, by classifying $x_h$ according to the first $h$ such that $\phi \diamond \pi_i(x_i) \neq \pi_i(x_i)$, we have the decomposition of identity

$$1 = \sum_{h=1}^H \sum_{x_h \in \mathcal{X}_{i,h}} \sum_{a_h \in \mathcal{A}_i} \mathbf{1}\left\{x_h \text{ visited, } a_{1:h} \text{ recomd., and } \phi(x_h, a_{1:h}) \neq a_h\right\}$$

$$+ \sum_{x_H \in \mathcal{X}_{i,H}} \sum_{a_H \in \mathcal{A}_i} \mathbf{1}\left\{x_H \text{ visited, } a_{1:H} \text{ recomd., and } \phi(x_H, a_{1:H}) = a_H\right\}.$$

As a consequence, for any $\phi \in \Phi_i^1$, $\pi_i \in \Pi_i$ and $\pi_{-i} \in \Pi_{-i}$, we have

$$V_i^{(\phi \diamond \pi_i) \times \pi_{-i}} - V_i^{\pi} = (\mathbb{E}_{(\phi \diamond \pi_i) \times \pi_{-i}} - \mathbb{E}_{\pi}) \left[ \sum_{k=1}^H r_{i,k} \right]$$

$$= (\mathbb{E}_{(\phi \diamond \pi_i) \times \pi_{-i}} - \mathbb{E}_{\pi}) \left[ \sum_{h=1}^H \sum_{x_h \in \mathcal{X}_{i,h}} \sum_{a_h \in \mathcal{A}_i} \mathbf{1}\left\{x_h \text{ visited, } a_{1:h} \text{ recomd., and } \phi(x_h, a_{1:h}) \neq a_h\right\} \sum_{k=1}^H r_{i,k} \right]$$

$$+ (\mathbb{E}_{(\phi \diamond \pi_i) \times \pi_{-i}} - \mathbb{E}_{\pi}) \left[ \sum_{x_H \in \mathcal{X}_{i,H}} \sum_{a_H \in \mathcal{A}_i} \mathbf{1}\left\{x_H \text{ visited, } a_{1:H} \text{ recomd., and } \phi(x_H, a_{1:H}) = a_H\right\} \sum_{k=1}^H r_{i,k} \right]$$

$$= \sum_{h=1}^H \sum_{x_h \in \mathcal{X}_{i,h}} \sum_{a_h \in \mathcal{A}_i} (\mathbb{E}_{(\phi \diamond \pi_i) \times \pi_{-i}} - \mathbb{E}_{\pi}) \left[ \mathbf{1}\left\{x_h \text{ visited, } a_{1:h} \text{ recomd., and } \phi(x_h, a_{1:h}) \neq a_h\right\} \sum_{k=1}^H r_{i,k} \right],$$

where the last equality used two facts: $(i)$ fixing $x_1, a_1, \cdots, x_H, a_H$, supposing that $\phi(x_h, a_{1:h}) = a_h$ for all $h \leq H$, then the probability of $x_H$ is visited and $a_{1:H}$ are recommended are the same under $(\phi \diamond \pi_i) \times \pi_{-i}$ and $\pi$; $(ii)$ the randomness of $\sum_{k=1}^H r_{i,k}$ is independent of policy when fixing $x_1, a_1, \cdots, x_H, a_H$. So the second quantity of left hand side of that equality is zero.

For any $(x_h, a_h) \in \mathcal{X}_{i,h} \times \mathcal{A}_i$, $\phi \in \Phi_i^1$ and $\pi_i \in \Pi_i$, we define the trigger policy $\mathsf{trig}(\pi_i, (\phi \diamond \pi_i), (x_h, a_h))$ to be a policy that plays $\pi_i$ before triggered by $(x_h, a_h)$ and plays $\phi \diamond \pi_i$ after triggered by $(x_h, a_h)$. Supposing that $\phi(x_{h'}, a_{1:h'}) = a_{h'}, \forall h' < h$ and $\phi(x_h, a_{1:h}) \neq a_h$, the probability of $x_h$ is visited and $a_{1:h}$ are recommended are the same the same under $(\phi \diamond \pi_i) \times \pi_{-i}$ and $\mathsf{trig}(\pi_i, (\phi \diamond \pi_i), (x_h, a_h)) \times \pi_{-i}$, which gives

$$\mathbb{E}_{(\phi \diamond \pi_i) \times \pi_{-i}} \left[ \mathbf{1}\left\{x_h \text{ visited, } a_{1:h} \text{ recomd., and } \phi(x_h, a_{1:h}) \neq a_h\right\} \sum_{h=1}^H r_{i,h} \right]$$

$$= \mathbb{E}_{\mathsf{trig}(\pi_i, (\phi \diamond \pi_i), (x_h, a_h)) \times \pi_{-i}} \left[ \mathbf{1}\left\{x_h \text{ visited, } a_{1:h} \text{ recomd., and } \phi(x_h, a_{1:h}) \neq a_h\right\} \sum_{h=1}^H r_{i,h} \right]. \tag{18}$$

Consequently, we have

$$\mathbb{E}_{\pi \sim \overline{\pi}}\left(V_i^{(\phi \diamond \pi_i) \times \pi_{-i}} - V_i^{\pi}\right)$$

$$= \sum_{h=1}^{H} \sum_{x_h \in \mathcal{X}_{i,h}} \sum_{a_h \in \mathcal{A}_i} \mathbb{E}_{\pi \sim \overline{\pi}}\left(\mathbb{E}_{(\phi \diamond \pi_i) \times \pi_{-i}} - \mathbb{E}_{\pi}\right)\left[\mathbf{1}\left\{x_h \text{ visited, } a_{1:h} \text{ recomd., and } \phi(x_h, a_{1:h}) \neq a_h\right\} \sum_{h=1}^{H} r_{i,h}\right]$$

$$\overset{(i)}{=} \sum_{h=1}^{H} \sum_{x_h \in \mathcal{X}_{i,h}} \sum_{a_h \in \mathcal{A}_i} \mathbf{1}\left\{\phi(x_h, a_{1:h}) \neq a_h\right\}$$

$$\times \mathbb{E}_{\pi \sim \overline{\pi}}\left(\mathbb{E}_{\text{trig}(\pi_i, (\phi \diamond \pi_i), (x_h, a_h)) \times \pi_{-i}} - \mathbb{E}_{\pi}\right)\left[\mathbf{1}\left\{x_h \text{ visited } and \text{ } a_{1:h} \text{ recomd.}\right\} \sum_{h=1}^{H} r_{i,h}\right]$$

$$\overset{(ii)}{=} \sum_{h=1}^{H} \sum_{x_h \in \mathcal{X}_{i,h}} \sum_{a_h \in \mathcal{A}_i} \mathbf{1}\left\{\phi(x_h, a_{1:h}) \neq a_h\right\} \mathbb{E}_{\pi \sim \overline{\pi}}\left(\mathbb{E}_{\text{trig}(\pi_i, (\phi \diamond \pi_i), (x_h, a_h)) \times \pi_{-i}} - \mathbb{E}_{\pi}\right)\left[\sum_{h=1}^{H} r_{i,h}\right]$$

$$\overset{(iii)}{\leq} \sum_{h=1}^{H} \sum_{x_h \in \mathcal{X}_{i,h}} \sum_{a_h \in \mathcal{A}_i} \text{TriggerGap}(\overline{\pi}) = X_i A_i \cdot \text{TriggerGap}(\overline{\pi}).$$

Here in (i) we used equation (18); in (iii) we bound the indicator by 1 and use the fact that TriggerGap is non-negative (by the observation that in the definition (17), we can choose $x_i$ to be some leaf infoset $x_{i,H}$ and choose $\widehat{\pi}_i(x_{i,H}) = a$ so that $\text{trig}(\pi_i, \widehat{\pi}_i, (x_i, a)) = \pi_i$); in (ii) we use the fact that $\text{trig}(\pi_i, (\phi \diamond \pi_i), (x_h, a_h))$ and $\pi$ are identical on any infoset $x$ such that $x \neq x_h$ and $x \not\succ (x_h, a_h)$, so that

$$\left(\mathbb{E}_{\text{trig}(\pi_i, (\phi \diamond \pi_i), (x_h, a_h)) \times \pi_{-i}} - \mathbb{E}_{\pi}\right)\left[\mathbf{1}\left\{a_{1:h} \text{ are not recommended or } x_h \text{ is not visited}\right\} \sum_{h=1}^{H} r_{i,h}\right] = 0.$$

Finally, take supermum over $\phi \in \Phi_i^1$ and then take supermum over $i \in [m]$, we get

$$1\text{-EFCEGap}(\overline{\pi}) = \max_{i \in [m]} \max_{\phi \in \Phi_i^1} \mathbb{E}_{\pi \sim \overline{\pi}}\left(V_i^{(\phi \diamond \pi_i) \times \pi_{-i}} - V_i^{\pi}\right) \leq \max_{i \in [m]} X_i A_i \cdot \text{TriggerGap}(\overline{\pi}).$$

This proves the lemma. $\qquad\square$

### C.2 Proof of Proposition C.2

*Proof.* We prove the containment result and strict containment result separately as follows.

**Proof of** $K\text{-EFCEGap}(\overline{\pi}) \leq (K+1)\text{-EFCEGap}(\overline{\pi})$ We claim that, for any $K \geq 0$ and strategy modification $\phi \in \Phi_i^K$, there exists $\phi' \in \Phi_i^{K+1}$ such that for any policy $\pi_i \in \Pi_i$, we have that $\phi \diamond \pi_i$ and $\phi' \diamond \pi_i$ gives the same policy. Given this claim, we have

$$\max_{\phi \in \Phi_i^K} \mathbb{E}_{\pi \sim \overline{\pi}} V_i^{(\phi \diamond \pi_i) \times \pi_{-i}} \leq \max_{\phi \in \Phi_i^{K+1}} \mathbb{E}_{\pi \sim \overline{\pi}} V_i^{(\phi \diamond \pi_i) \times \pi_{-i}}.$$

This implies $K\text{-EFCEGap}(\overline{\pi}) \leq (K+1)\text{-EFCEGap}(\overline{\pi})$ for all $K \geq 0$.

To prove such a claim, we can choose the strategy modification $\phi' \in \Phi_i^{K+1}$ to be the following: (1) For any rechistory $(x_{i,h}, b_{1:h-1}) \in \Omega_i^{(\text{I}),K} \cap \Omega_i^{(\text{I}),K+1}$ and any action $a_h \in \mathcal{A}_i$, we set $\phi'(x_{i,h}, b_{1:h-1}, a_h) = \phi(x_{i,h}, b_{1:h-1}, a_h)$; (2) For any rechistory $(x_{i,h}, b_{1:h-1}) \in \Omega_i^{(\text{I}),K+1} \setminus \Omega_i^{(\text{I}),K}$ and any action $a_h \in \mathcal{A}_i$, we set $\phi'(x_{i,h}, b_{1:h-1}, a_h) = \phi(x_{i,h}, b_{1:h_{\star\star}})$ where $h_{\star\star} = \inf\{k \leq h : \sum_{h''=1}^{k} \mathbf{1}\{a_{h''} \neq b_{h''}\} = K\}$; (3) For any rechistory $(x_{i,h}, b_{1:h'}) \in \Omega_i^{(\text{II}),K+1}$, we set $\phi'(x_{i,h}, b_{1:h'}) = \phi(x_{i,h}, b_{1:h_{\star\star}})$ where $h_{\star\star} = \inf\{k \leq h' : \sum_{h''=1}^{k} \mathbf{1}\{a_{h''} \neq b_{h''}\} = K\}$. It is easy to see that for any $\pi_i \in \Pi_i$, we have that $\phi' \diamond \pi_i$ is the same as $\phi \diamond \pi_i$. This proves the claim.

**Example of a game and a $\overline{\pi}$ with $K\text{-EFCEGap}(\overline{\pi}) = 0$ but $(K+1)\text{-EFCEGap}(\overline{\pi}) \geq 1/3$**

For any $K \geq 0$, we consider a two-player game with $H = K + 1$ steps and perfect information. The action spaces are $\mathcal{A}_1 = \{1, 2\}$ for the first player and $\mathcal{A}_2 = \{1, 2\}$ for the second player in each time

step. The state space $\mathcal{S} = \cup_{h=1}^{H} \mathcal{S}_h$ can be identified as $\mathcal{S}_h = \mathcal{A}_1^{h-1} \times \mathcal{A}_2^{h-1}$ for $h = 1, 2, \ldots, K+1$ and both players' infosets are the same as the state space $\mathcal{X}_1 = \mathcal{X}_2 = \mathcal{S}$. If action $(a_h, b_h)$ is taken at $s_h = (a_{1:h-1}, b_{1:h-1}) \in \mathcal{S}_h$, then the environment will transit to the next state given by $s_{h+1} = (a_{1:h}, b_{1:h})$. The reward for the second player is always 0 at every time step. We design the reward for the first player (denoting as $r_h$ in short) as following:

- The reward $r_h(\cdot, \cdot) = 0$ when $h \leq K$ for every state actions.

- The reward at $s_{K+1} = (a_{1:K}, b_{1:K})$ is defined as

$$r_{K+1}(s_{K+1}, a_{K+1}, b_{K+1}) = \mathbf{1}\{a_1 \neq b_1, \ldots, a_{K+1} \neq b_{K+1}\}$$
$$+ \frac{1}{2} \cdot \mathbf{1}\{a_1 = b_1, \ldots, a_{K+1} = b_{K+1}\}.$$

Let $\Pi_\star = \{(\pi_\star, \pi_\star) : \pi_\star \in \Pi_1\}$ where $\Pi_1$ is the set of pure policies of the first player. That means, $\Pi_\star$ is the set of pure policies such that two players take the same action (either 1 or 2) at each state. We define $\overline{\pi}$ as the uniform distribution over such a policy space $\Pi_\star$. We claim that $K$-EFCEGap$(\overline{\pi}) = 0$ but $(K+1)$-EFCEGap$(\overline{\pi}) \geq 1/3$. Since the reward of the second player is always 0, we only need to consider the value function gap of the first player. Note that the value function of the first player for the correlated policy $\overline{\pi}$ is $1/2$.

We first consider the $(K+1)$-EFCEGap. If the first player deviates from the recommended action in every time step (this is an allowed strategy modification in $\Phi_i^{K+1}$), she can receive reward 1 so that her received value is 1. As a consequence, we have

$$(K+1)\text{-EFCEGap}(\overline{\pi}) \geq 1 - 1/2 > 1/3.$$

We then consider the $K$-EFCEGap. If the first player chooses to deviate at any step, she need to play a different action from the second player at all time steps to receive an reward 1, otherwise she will receive reward 0. However, she is only allowed to see the recommendation $K$ times. There is at least one time step such that she cannot see the recommendation and she need to guess what is the second player's action. The probability that her guess coincides with the other player's action is $1/2$ no matter how she guess. So by deviating from the recommended action, the first player can receive a value at most $1/2$. That means, $K$-EFCEGap$(\overline{\pi}) \leq 1/2 - 1/2 = 0$. This finishes the proof of the proposition. $\qquad\square$

### C.3 Proof of Proposition C.3

(a) We first show that $\infty$-EFCEGap$(\overline{\pi}) \leq$ NFCEGap$(\overline{\pi})$. Indeed, for any $\infty$-EFCE strategy modification $\phi \in \Phi_i^{\infty\text{-EFCE}}$, we let $\phi' \in \Phi_i^{\text{NFCE}}$ such that for any $x_{i,h} \in \mathcal{X}_i$ and $\pi_i \in \Pi_i^{\text{pure}}$, $\phi'(x_{i,h}, \pi_i) := \phi(x_{i,h}, b_{1:h-1})$ where $b_{1:h-1} := (\pi_i(x_{i,1}), \ldots, \pi_i(x_{i,h-1}))$ and $x_{i,1} \preceq \cdots \prec x_{i,h-1} \prec x_{i,h}$ are the unique history of infosets leading to $x_{i,h}$. By comparing the execution of $\phi \diamond \pi$ (cf. Algorithm 1) and $\phi' \diamond \pi$ (cf. Definition C.1), the policy $\phi \diamond \pi_i$ exactly implements (i.e. is the same as) $\phi' \diamond \pi_i$. This gives

$$\infty\text{-EFCEGap}(\overline{\pi}) = \max_{\phi \in \Phi_i^{\infty\text{-EFCE}}} \mathbb{E}_{\pi \sim \overline{\pi}} V_i^{(\phi \diamond \pi_i) \times \pi_{-i}} \leq \max_{\phi \in \Phi_i^{\text{NFCE}}} \mathbb{E}_{\pi \sim \overline{\pi}} V_i^{(\phi \diamond \pi_i) \times \pi_{-i}} = \text{NFCEGap}(\overline{\pi}).$$

We next prove the second claim (converse bound does not hold), by constructing the following example.

**Example 1** (There exists an $\infty$-EFCE which is not $\varepsilon$-approximate NFCE with $\varepsilon = 1/20$): We consider a two-player game with 2 steps and perfect information. The set of infosets $\mathcal{X}_{i,h}$ for both players gives $\mathcal{X}_{1,1} = \mathcal{X}_{2,1} = \{s_0\}$ and $\mathcal{X}_{1,2} = \mathcal{X}_{2,2} = \{s_{1,1}, s_{1,2}, s_{2,1}, s_{2,2}\}$. The action spaces are $\mathcal{A}_1 = \{a_1, a_2\}$ and $\mathcal{A}_2 = \{b_1, b_2\}$. If action pair $(a_i, b_j)$ $(i, j = 1, 2)$ is chosen at $s_0$, $s_{i,j}$ would be reached with probability 1. The reward for the second player is always 0. And we design the reward for the first player as following:

- The reward at $s_0$ depends only on the action of the first player: the reward is $1/2$ if $a_1$ is chosen and 0 if $a_2$ is chosen.

- The rewards at $s_{1,1}$ and $s_{1,2}$ are always 0. The rewards at $s_{2,1}$ and $s_{2,2}$ depends only on the action of the second player: the rewards are all 1 if $b_1$ is chosen and 0 if $b_2$ is chosen.

Suppose $\overline{\pi}$ is the uniform distribution of all the deterministic policies that takes $(a_1, b_1)$ or $(a_2, b_2)$ at each infosets (there are $2^5 = 32$ such policies). We would verify that $\overline{\pi}$ is a $\infty$-EFCE but not a $1/20$-NFCE. Since the reward of the second player is always $0$, we only need to consider the first player.

We first consider NFCE strategy modifications. On one hand, the first player only has motivation to modify his action at $s_0$ to $a_2$ if he observes that the recommendation at $s_0$, $s_{2,1}$ and $s_{2,2}$ are all $a_1$ (which happens iff the recommendation for his opponent are all $b_1$). Otherwise, he does not have motivation to change his action. If the first player choose such a modification, he will modify only $1/8$ of the deterministic policies, and for each deterministic policy $\pi_i$ that are modified, the reward of the first player is increased by $1/2$. So using this NFCE strategy modification, the first player's value function are increased by $1/16$, which gives that

$$\text{NFCEGap}(\overline{\pi}) \geq 1/16 > 1/20.$$

On the other hand, for any $\infty$-EFCE strategy modification, taking $a_2$ at $s_0$ always has utility $1/2$ since the actions taken by the second player at $\mathcal{X}_2$ are all uniformly distributed conditional on the recommendation at $s_0$. The utility of taking $a_1$ at $s_0$ is also $1/2$. This means that any $\infty$-EFCE strategy modification of $\overline{\pi}$ has value function $1/2$, so does $\overline{\pi}$. Consequently, $\overline{\pi}$ is an exact $\infty$-EFCE, i.e. $\infty$-EFCEGap$(\overline{\pi}) = 0$. $\diamond$

(b) Consider $K$-EFCE with $K = 0$. From the definition of strategy modifications and the executing of modified policy (Algorithm 1), for $\phi \in \Phi_i^0$ and policy $\pi_i \in \Pi_i^{\text{pure}}$, $\phi \diamond \pi_i$ takes action $\phi(x_i, \emptyset)$ at $x_i$. So $\phi$ is equivalent to a modification $\phi' \in \Phi_i^{\text{NFCCE}}$ which satisfies $\phi'(x_i) = \phi(x_i, \emptyset)$ for all $x_i \in \mathcal{X}_i$. Here, the equivalence means that $\phi \diamond \pi_i = \phi' \diamond \pi_i$ for any $\pi_i \in \Pi_i^{\text{pure}}$. This gives

$$0\text{-EFCEGap}(\overline{\pi}) = \max_{\phi \in \Phi_i^0} \mathbb{E}_{\pi \sim \overline{\pi}} V_i^{(\phi \diamond \pi_i) \times \pi_{-i}} = \max_{\phi \in \Phi_i^{\text{NFCCE}}} \mathbb{E}_{\pi \sim \overline{\pi}} V_i^{(\phi \diamond \pi_i) \times \pi_{-i}} = \text{NFCCEGap}(\overline{\pi}),$$

which is the desired result. $\square$

# D  Properties of $K$-EFCE strategy modifications

For any $\phi \in \Phi_i^K$, we define its "probabilistic" expression $\mu^\phi$ as follows: For any $a_h \in \mathcal{A}_i$,

$$\mu_h^\phi(a_h | x_{i,h}, b_{1:h-1}, b_h) := \mathbf{1}\{a_h = \phi(x_{i,h}, b_{1:h-1}, b_h)\} \quad \text{for all } (x_{i,h}, b_{1:h-1}), b_h \in \Omega_{i,h}^{(\text{I}),K} \times \mathcal{A}_i,$$

$$\mu_h^\phi(a_h | x_{i,h}, b_{1:h'}) := \mathbf{1}\{a_h = \phi(x_{i,h}, b_{1:h'})\} \qquad \text{for all } (x_{i,h}, b_{1:h'}) \in \Omega_{i,(h_\star,h)}^{(\text{II}),K}.$$

In words, $\mu_h^\phi(\cdot | x_{i,h}, b_{1:h-1}, b_h) \in \Delta(\mathcal{A}_i)$ is the pure policy that takes action $\phi(x_{i,h}, b_{1:h-1}, b_h)$ deterministically, for any Type-I rechistory $(x_{i,h}, b_{1:h-1})$ and recommendation $b_h \in \mathcal{A}_i$; $\mu_h^\phi(a_h | x_{i,h}, b_{1:h'})$ is the pure policy that takes action $\phi(x_{i,h}, b_{1:h'})$ for any Type-II rechistory $(x_{i,h}, b_{1:h'})$. For convenience, we abuse notation slightly to let

$$\phi_h(\cdot | x_{i,h}, b_{1:h}) := \mu_h^\phi(\cdot | x_{i,h}, b_{1:h-1}, b_h), \quad \phi_h(\cdot | x_{i,h}, b_{1:h'}) := \mu_h^\phi(\cdot | x_{i,h}, b_{1:h'}).$$

Moreover, we use $\mathsf{D}(a_{1:k}, b_{1:k})$ to denote the Hamming distance of two action sequences $a_{1:k}, b_{1:k} \in \mathcal{A}_i^k$:

$$\mathsf{D}(a_{1:k}, b_{1:k}) := \sum_{h=1}^k \mathbf{1}\{a_h \neq b_h\},$$

and define the following notation as shorthand for the indicator that $a_{1:h}$ and $b_{1:h}$ differs in $\{\leq K-1, K\}$ elements:

$$\delta^{\leq K-1}(a_{1:h-1}, b_{1:h-1}) := \mathbf{1}\{\mathsf{D}(a_{1:h-1}, b_{1:h-1}) \leq K-1\};$$

$$\delta^K(a_{1:h-1}, b_{1:h-1}) := \mathbf{1}\{\mathsf{D}(a_{1:h-1}, b_{1:h-1}) = K\}.$$

**Lemma D.1.** *For any $\phi \in \Phi_i^K$ and any (potentially mixed) policy $\pi_i$ for the $i^{th}$ player, $\phi \diamond \pi_i$ is also a (potentially mixed) policy for the $i^{th}$ player, with sequence-form expression*

$$(\phi \diamond \pi_i)_{1:h}(x_h, a_h) = \sum_{b_{1:h}} \delta^{\leq K-1}(a_{1:h-1}, b_{1:h-1}) \prod_{k=1}^h \phi_k(a_k | x_k, b_{1:k}) \prod_{k=1}^h \pi_i(b_k | x_k)$$

$$+\sum_{b_{1:h}} \delta^K(a_{1:h-1}, b_{1:h-1}) \prod_{k=1}^{h} \phi_k(a_k|x_k, b_{1:k\wedge\tau_K}) \prod_{k=1}^{\tau_K} \pi_i(b_k|x_k),$$

where

$$\tau_K := \inf\left\{ h' \le h-1 : \sum_{h''=1}^{h'} \mathbf{1}\{a_{h''} \ne b_{h''}\} \ge K \right\} \tag{19}$$

is the time step of the K-th deviation, and the event $\{\tau_K \le k\}$ can be determined by $(a_{1:k}, b_{1:k})$ for any $k \ge 1$. Furthermore, we have

$$\sum_{(x_h, a_h) \in \mathcal{X}_{i,h} \times \mathcal{A}_i} \frac{(\phi \diamond \pi_i)_{1:h}(x_h, a_h)}{\pi_{i,1:h}^{\star,h}(x_h, a_h)} = X_{i,h} A_i.$$

*Proof.* Suppose the ancestors of $x_h$ are $x_1 \prec x_2 \prec \cdots \prec x_{h-1} \prec x_h$ and the actions leading to $x_h$ are $a_1, \ldots, a_{h-1}$. The sequence-form expression $(\phi \diamond \pi_i)(x_h, a_h)$ is the probability of $(\phi \diamond \pi_i)$ choose $a_k$ at $x_k$ for all $k \in [h]$. We further denote the recommended action $b_k = \pi_i(x_k)$ for all $k \in [h]$.

If $|\{h' \in [h-1] : a_{h'} \ne b_{h'}\}| \le K-1$, the conditional probability of $(\phi \diamond \pi)$ choosing $a_k$ at $x_k$ for all $k \in [h]$ is $\prod_{k=1}^{h} \phi_k(a_k|x_k, b_{1:k})$, as the player would always swap the action; If $|\{h' \in [h] : a_{h'} \ne b_{h'}\}| \ge K$, the conditional probability of $(\phi \diamond \pi)$ choosing $a_k$ at $x_k$ for all $k \in [h]$ is $\prod_{k=1}^{h} \phi_k(a_k|x_k, b_{1:k\wedge\tau_K})$. So by the law of total probability, we have

$$(\phi \diamond \pi_i)_{1:h}(x_h, a_h) = \sum_{b_{1:h}} \delta^{\le K-1}(a_{1:h-1}, b_{1:h-1}) \prod_{k=1}^{h} \phi_k(a_k|x_k, b_{1:k}) \prod_{k=1}^{h} \pi_i(b_k|x_k)$$

$$+ \sum_{b_{1:h}} \mathbf{1}\{|\{h' \in [h] : a_{h'} \ne b_{h'}\}| \ge K\} \prod_{k=1}^{h} \phi_k(a_k|x_k, b_{1:k\wedge\tau_K}) \prod_{k=1}^{h} \pi_i(b_k|x_k).$$

Notice that $\prod_{k=1}^{h} \phi_k(a_k|x_k, b_{1:k\wedge\tau_K})$ only depend on $b_{1:\tau_K}$ and $\sum_{b_{\tau_K:h}} \prod_{k=1}^{h} \pi_i(b_k|x_k) = \prod_{k=1}^{\tau_K} \pi_i(b_k|x_k)$, so the second summation admits a simpler form:

$$\sum_{b_{1:h}} \mathbf{1}\{|\{h' \in [h] : a_{h'} \ne b_{h'}\}| \ge K\} \prod_{k=1}^{h} \phi_k(a_k|x_k, b_{1:k\wedge\tau_K}) \prod_{k=1}^{h} \pi_i(b_k|x_k)$$

$$= \sum_{(h', b_{1:h'}): \sum_1^{h'} \mathbf{1}\{a_k \ne b_k\}=K \text{ and } b_{h'} \ne a_{h'}} \prod_{k=1}^{h} \phi_k(a_k|x_k, b_{1:k\wedge h'}) \prod_{k=1}^{h'} \pi_i(b_k|x_k)$$

$$= \sum_{b_{1:h}} \delta^K(a_{1:h-1}, b_{1:h-1}) \prod_{k=1}^{h} \phi_k(a_k|x_k, b_{1:k\wedge\tau_K}) \prod_{k=1}^{\tau_K} \pi_i(b_k|x_k).$$

The last equality is because we can append $a_{h'+1:h}$ to $b_{1:h'}$ to get a new $b_{1:h}$ which doesn't change the value of the summation. Then the first part of this lemma is proved. The second part actually is a direct corollary of Lemma B.4. □

The lemma above has the following corollary.

**Corollary D.1.** *For the $i^{th}$ player and any pure policy $\pi \in \Pi$, fix any $\phi \in \Phi_i^K$ and $x_h \in \mathcal{X}_{i,h}$ with $(a_1, \ldots, a_{h-1})$ being the unique history of actions leading to $x_h$. Then the probability that $x_h$ is reached by the $i^{th}$ player under policy $(\phi \diamond \pi_i) \times \pi_{-i}$ is*

$$\mathbb{P}_{\phi \diamond \pi_i \times \pi_{-i}}\left( x_h \text{ is reached by the } i^{th} \text{ player} \right)$$

$$= \sum_{b_{1:h-1}} \delta^{\le K-1}(a_{1:h-1}, b_{1:h-1}) \prod_{k=1}^{h-1} \phi_k(a_k|x_k, b_{1:k}) \cdot \prod_{k=1}^{h-1} \pi_i(b_k|x_k) p_{1:h}^{\pi_{-i}}(x_{i,h})$$

$$+ \sum_{b_{1:h-1}} \delta^K(a_{1:h-1}, b_{1:h-1}) \prod_{k=1}^{h-1} \phi_k(a_k|x_k, b_{1:k \wedge \tau_K}) \cdot \prod_{k=1}^{\tau_K} \pi_i(b_k|x_k) p_{1:h}^{\pi_{-i}}(x_{i,h}).$$

*Proof.* We have

$$\mathbb{P}_{\phi \diamond \pi_i \times \pi_{-i}} \left( x_h \text{ is reached by the } i^{th} \text{ player} \right)$$
$$= \sum_{a \in \mathcal{A}_i} (\phi \diamond \pi_i)_{1:h}(x_h, a) p_{1:h}^{\pi_{-i}}(x_{i,h}),$$

where $p_{1:h}^{\pi_{-i}}(x_{i,h})$ is defined in equation (12). So applying Lemma D.1 yields the desired result. $\square$

As each step $h$, one (and only one) $x_h \in \mathcal{X}_{i,h}$ is visited, so the summation of the above reaching probability over $x_h$ is 1. This directly yields the following corollary.

**Corollary D.2.** *For the $i^{th}$ player and any policy $\pi \in \Pi$, fix any $\phi \in \Phi_i^K$, we have*

$$\sum_{x_h \in \mathcal{X}_{i,h}} \sum_{b_{1:h-1}} \delta^{\leq K-1}(a_{1:h-1}, b_{1:h-1}) \prod_{k=1}^{h-1} \phi_k(a_k|x_k, b_{1:k}) \cdot \prod_{k=1}^{h-1} \pi_i(b_k|x_k) p_{1:h}^{\pi_{-i}}(x_{i,h})$$

$$+ \sum_{x_h \in \mathcal{X}_{i,h}} \sum_{b_{1:h-1}} \delta^K(a_{1:h-1}, b_{1:h-1}) \prod_{k=1}^{h-1} \phi_k(a_k|x_k, b_{1:k \wedge \tau_K}) \cdot \prod_{k=1}^{\tau_K} \pi_i(b_k|x_k) p_{1:h}^{\pi_{-i}}(x_{i,h}) = 1.$$

# E   Regret decomposition for $K$-EFCE regret

This section presents the properties of $K$-EFCE regret, which will be useful for the proofs of our main results. Let $\{\pi^t\}_{t \in [T]}$ be a sequence of policies. Recall the $K$-EFCE regret for the $i^{th}$ player (6):

$$R_{i,K}^T = \max_{\phi \in \Phi_i^K} \sum_{t=1}^T \left( V_i^{\phi \diamond \pi_i^t \times \pi_{-i}^t} - V_i^{\pi^t} \right). \tag{20}$$

**Lemma E.1** (Online-to-batch for $K$-EFCE). *Let $\{\pi^t = (\pi_i^t)_{i \in [m]}\}_{t \in [T]}$ be a sequence of product policies for all players over $T$ rounds. Then, for the average (correlated) policy $\overline{\pi} = \mathrm{Unif}(\{\pi^t\}_{t=1}^T)$, we have*

$$K\text{-EFCEGap}(\overline{\pi}) = \max_{i \in [m]} R_{i,K}^T / T.$$

*Proof.* This follows directly by the definition of $K$-EFCEGap:

$$K\text{-EFCEGap}(\overline{\pi}) = \max_{i \in [m]} \max_{\phi \in \Phi_i^K} \left( V_i^{\phi \diamond \overline{\pi}} - V_i^{\overline{\pi}} \right)$$
$$= \max_{i \in [m]} \max_{\phi \in \Phi_i^K} \mathbb{E}_{\pi \sim \overline{\pi}} \left[ V_i^{\phi \diamond \pi_i \times \pi_{-i}} - V_i^{\pi} \right]$$
$$= \max_{i \in [m]} \max_{\phi \in \Phi_i^K} \frac{1}{T} \sum_{t=1}^T \left[ V_i^{\phi \diamond \pi_i^t \times \pi_{-i}^t} - V_i^{\pi^t} \right]$$
$$= \max_{i \in [m]} R_{i,K}^T / T.$$

This proves the lemma. $\square$

For $(x_{i,h}, b_{1:h-1}, \varphi) \in \Omega_i^{(I),K} \times \Psi^s$, we define the immediate local swap regret as

$$\widehat{R}_{(x_h, b_{1:h-1}), \varphi}^T := \sum_{t=1}^T \prod_{k=1}^{h-1} \pi_i^t(b_k|x_k) \left( \left\langle \pi_{i,h}^t(\cdot|x_{i,h}) - \varphi \diamond \pi_{i,h}^t(\cdot|x_{i,h}), L_{i,h}^t(x_{i,h}, \cdot) \right\rangle \right),$$

and the (overall) local swap regret as

$$\widehat{R}^{T,\text{swap}}_{(x_h,b_{1:h-1})} := \max_{\varphi} \widehat{R}^{T}_{(x_h,b_{1:h-1}),\varphi}. \tag{21}$$

For $(x_{i,h}, b_{1:h'}, a) \in \Omega_i^{(\text{II}),K} \times \mathcal{A}_i$, we define the immediate local external regret as

$$\widehat{R}^{T}_{(x_h,b_{1:h'}),a} := \sum_{t=1}^{T} \prod_{k=1}^{h'} \pi_i^t(b_k|x_k) \Big( \langle \pi_{i,h}^t(\cdot|x_{i,h}), L_{i,h}^t(x_{i,h},\cdot) \rangle - L_{i,h}^t(x_{i,h},a) \Big),$$

and the (overall) local external regret as

$$\widehat{R}^{T,\text{ext}}_{(x_h,b_{1:h'})} := \max_{a \in \mathcal{A}_i} \widehat{R}^{T}_{(x_h,b_{1:h'}),a}. \tag{22}$$

$K$-EFCE **regret decomposition**    Our main result in this section is the following regret decomposition that decomposes the $K$-EFCE regret $R_{i,K}^T$ into combinations of local regrets at each rechistory.

**Lemma E.2** (Regret decomposition for $K$-EFCE regret). *We have $R_{i,K}^T \leq \sum_{h=1}^{H} R_h^T$, with*

$$R_h^T := \max_{\phi \in \Phi_i^K} \sum_{x_h \in \mathcal{X}_{i,h}} G_h^{T,\text{swap}}(x_h; \phi) + \max_{\phi \in \Phi_i^K} \sum_{x_h \in \mathcal{X}_{i,h}} G_h^{T,\text{ext}}(x_h; \phi),$$

*where*

$$G_h^{T,\text{swap}}(x_h; \phi) := \sum_{b_{1:h-1}} \delta^{\leq K-1}(a_{1:h-1}, b_{1:h-1}) \prod_{k=1}^{h-1} \phi_k(a_k|x_k, b_{1:k}) \widehat{R}^{T,\text{swap}}_{(x_h,b_{1:h-1})}, \tag{23}$$

*and*

$$G_h^{T,\text{ext}}(x_h; \phi) := \sum_{b_{1:h-1}} \delta^K(a_{1:h-1}, b_{1:h-1}) \prod_{k=1}^{h-1} \phi_k(a_k|x_k, b_{1:k \wedge \tau_K}) \widehat{R}^{T,\text{ext}}_{(x_h,b_{1:\tau_K})}$$

$$= \sum_{h'=K}^{h-1} \sum_{b_{1:h'}} \mathbf{1}\{\tau_K = h'\} \prod_{k=1}^{h-1} \phi_k(a_k|x_k, b_{1:k \wedge h'}) \widehat{R}^{T,\text{ext}}_{(x_h,b_{1:h'})}. \tag{24}$$

*Above, $a_{1:h-1}$ is the unique sequence of actions leading to $x_h$, and $\tau_K$ (cf. definition in (19)) depends on $a_{1:h-1}$, $b_{1:h-1}$.*

*Proof of Lemma E.2.* We begin by performing the following performance decomposition

$$R_{i,K}^T = \max_{\phi \in \Phi_i^K} \sum_{t=1}^{T} (V_i^{\phi \diamond \pi_i^t \times \pi_{-i}^t} - V_i^{\pi^t})$$

$$= \max_{\phi \in \Phi_i^K} \sum_{t=1}^{T} \left( \mathbb{E}_{\phi \diamond \pi_i^t \times \pi_{-i}^t} \left[ \sum_{h=1}^{H} r_{i,h} \right] - \mathbb{E}_{\pi^t} \left[ \sum_{h=1}^{H} r_{i,h} \right] \right)$$

$$= \max_{\phi \in \Phi_i^K} \sum_{t=1}^{T} \sum_{h=1}^{H} \left( \mathbb{E}_{((\phi \diamond \pi_i^t)_{1:h}\pi_{i,h+1:H}^t) \times \pi_{-i}^t} \left[ \sum_{k=1}^{H} r_{i,k} \right] - \mathbb{E}_{((\phi \diamond \pi_i^t)_{1:h-1}\pi_{i,h:H}^t) \times \pi_{-i}^t} \left[ \sum_{k=1}^{H} r_{i,k} \right] \right)$$

$$= \max_{\phi \in \Phi_i^K} \sum_{t=1}^{T} \sum_{h=1}^{H} \left( \mathbb{E}_{((\phi \diamond \pi_i^t)_{1:h}\pi_{i,h+1:H}^t) \times \pi_{-i}^t} \left[ \sum_{k=h}^{H} r_{i,k} \right] - \mathbb{E}_{((\phi \diamond \pi_i^t)_{1:h-1}\pi_{i,h:H}^t) \times \pi_{-i}^t} \left[ \sum_{k=h}^{H} r_{i,k} \right] \right)$$

$$= \max_{\phi \in \Phi_i^K} \sum_{h=1}^{H} \sum_{t=1}^{T} \left( \mathbb{E}_{((\phi \diamond \pi_i^t)_{1:h-1}\pi_{i,h:H}^t) \times \pi_{-i}^t} \left[ \sum_{k=h}^{H} (1 - r_{i,k}) \right] - \mathbb{E}_{((\phi \diamond \pi_i^t)_{1:h}\pi_{i,h+1:H}^t) \times \pi_{-i}^t} \left[ \sum_{k=h}^{H} (1 - r_{i,k}) \right] \right).$$

Here, $((\phi \diamond \pi_i^t)_{1:h}\pi_{i,h+1:H}^t) \times \pi_{-i}^t$ refers to the policy that the $i^{\text{th}}$ player uses $\phi \diamond \pi_i^t$ for the first $h$ step, and then uses $\pi_i^t$ where as other players always use $\pi_{-i}^t$. The last step use the fact that $((\phi \diamond \pi_i^t)_{1:h}\pi_{i,h+1:H}^t) \times \pi_{-i}^t$ and $((\phi \diamond \pi_i^t)_{1:h-1}\pi_{i,h:H}^t) \times \pi_{-i}^t$ are the same for the first $h-1$ steps, so the expected reward in first $h-1$ steps are the same, too. Therefore, define

$$\widetilde{R}_h^T = \max_{\phi \in \Phi_i^K} \sum_{t=1}^{T} \left( \mathbb{E}_{((\phi \diamond \pi_i^t)_{1:h-1}\pi_{i,h:H}^t) \times \pi_{-i}^t} \left[ \sum_{k=h}^{H} (1 - r_{i,k}) \right] - \mathbb{E}_{((\phi \diamond \pi_i^t)_{1:h}\pi_{i,h+1:H}^t) \times \pi_{-i}^t} \left[ \sum_{k=h}^{H} (1 - r_{i,k}) \right] \right),$$

we have $R_{i,K}^T \leq \sum_{h=1}^H \widetilde{R}_h^T$.

We next show that

$$\widetilde{R}_h^T \leq \max_{\phi \in \Phi_i^K} \sum_{x_h \in \mathcal{X}_{i,h}} G_h^{T,\mathrm{swap}}(x_h; \phi) + \max_{\phi \in \Phi_i^K} \sum_{x_h \in \mathcal{X}_{i,h}} G_h^{T,\mathrm{ext}}(x_h; \phi) = R_h^T,$$

which yields the desired result. Fix any $h \in [H]$ and $\phi \in \Phi_i^K$, according to the execution of modified policy $\phi \diamond \pi_i$ as in Algorithm 1, we have

$$\mathbb{E}_{((\phi \diamond \pi_i^t)_{1:h} \pi_{i,h+1:H}^t) \times \pi_{-i}^t} \left[ \sum_{k=h}^H (1 - r_{i,k}) \right]$$

$$= \sum_{x_h \in \mathcal{X}_{i,h}} \sum_{b_{1:h-1}} \prod_{k=1}^{h-1} \pi_i^t(b_k | x_k) \underbrace{\prod_{k=1}^{h-1} \phi_k(a_k | x_k, b_{1:k \wedge \tau_K})}_{\text{probability of taking 'right' actions leading to } x_h} \underbrace{\left\langle \phi_h(\cdot | x_h, b_{1:h \wedge \tau_K}), L_{i,h}^t(x_h, \cdot) \right\rangle}_{\text{counterfactual loss}},$$

where we assume $\tau_K := \inf\{h' \leq h - 1 : \sum_{h''=1}^{h'} \mathbf{1}\{a_{h''} \neq b_{h''}\} \geq K\}$. Similarly,

$$\mathbb{E}_{((\phi \diamond \pi_i^t)_{1:h-1} \pi_{i,h:H}^t) \times \pi_{-i}^t} \left[ \sum_{k=h}^H (1 - r_{i,k}) \right]$$

$$= \sum_{x_h \in \mathcal{X}_{i,h}} \sum_{b_{1:h-1}} \prod_{k=1}^{h-1} \pi_i^t(b_k | x_k) \underbrace{\prod_{k=1}^{h-1} \phi_k(a_k | x_k, b_{1:k \wedge \tau_K})}_{\text{probability of taking 'right' actions leading to } x_h} \underbrace{\left\langle \pi_i^t(\cdot | x_h), L_{i,h}^t(x_h, \cdot) \right\rangle}_{\text{counterfactual loss}}.$$

Substituting these into $\widetilde{R}_h^T$, we have

$$\widetilde{R}_h^T = \max_{\phi \in \Phi_i^K} \sum_{t=1}^T \left( \mathbb{E}_{((\phi \diamond \pi_i^t)_{1:h-1} \pi_{i,h:H}^t) \times \pi_{-i}^t} \left[ \sum_{k=h}^H (1 - r_{i,k}) \right] - \mathbb{E}_{((\phi \diamond \pi_i^t)_{1:h} \pi_{i,h+1:H}^t) \times \pi_{-i}^t} \left[ \sum_{k=h}^H (1 - r_{i,k}) \right] \right)$$

$$= \max_{\phi \in \Phi_i^K} \sum_{t=1}^T \sum_{x_h \in \mathcal{X}_{i,h}} \sum_{b_{1:h-1}} \left( \prod_{k=1}^{h-1} \pi_i^t(b_k | x_k) \prod_{k=1}^{h-1} \phi_k(a_k | x_k, b_{1:k \wedge \tau_K}) \left\langle \pi_i^t(\cdot | x_h) - \phi_h(\cdot | x_h, b_{1:h \wedge \tau_K}), L_{i,h}^t(x_h, \cdot) \right\rangle \right)$$

$$= \max_{\phi \in \Phi_i^K} \sum_{x_h \in \mathcal{X}_{i,h}} \sum_{b_{1:h-1}} \left( \prod_{k=1}^{h-1} \phi_k(a_k | x_k, b_{1:k \wedge \tau_K}) \sum_{t=1}^T \prod_{k=1}^{h-1} \pi_i^t(b_k | x_k) \left\langle \pi_i^t(\cdot | x_h) - \phi_h(\cdot | x_h, b_{1:h \wedge \tau_K}), L_{i,h}^t(x_h, \cdot) \right\rangle \right).$$

For fixed $\phi$, $x_h$, based on whether $\mathsf{D}(a_{1:h-1}, b_{1:h-1}) \leq K - 1$ or not, we have

$$\widetilde{R}_h^T = \max_{\phi \in \Phi_i^K} \sum_{x_h \in \mathcal{X}_{i,h}} \sum_{b_{1:h-1}} \left( \prod_{k=1}^{h-1} \phi_k(a_k | x_k, b_{1:k \wedge \tau_K}) \sum_{t=1}^T \prod_{k=1}^{h-1} \pi_i^t(b_k | x_k) \left\langle \pi_i^t(\cdot | x_h) - \phi_h(\cdot | x_h, b_{1:h \wedge \tau_K}), L_{i,h}^t(x_h, \cdot) \right\rangle \right)$$

$$= \max_{\phi \in \Phi_i^K} (\mathrm{I}_h + \mathrm{II}_h),$$

where

$$\mathrm{I}_h := \sum_{x_h \in \mathcal{X}_{i,h}} \sum_{b_{1:h-1}} \delta^{\leq K-1}(a_{1:h-1}, b_{1:h-1}) \prod_{k=1}^{h-1} \phi_k(a_k | x_k, b_{1:k})$$

$$\times \sum_{t=1}^T \prod_{k=1}^{h-1} \pi_i^t(b_k | x_k) \left\langle \pi_i^t(\cdot | x_h) - \phi_h(\cdot | x_h, b_{1:h \wedge \tau_K}), L_{i,h}^t(x_h, \cdot) \right\rangle,$$

and

$$\mathrm{II}_h := \sum_{x_h \in \mathcal{X}_{i,h}} \sum_{b_{1:h-1}} \mathbf{1}\{\mathsf{D}(a_{1:h-1}, b_{1:h-1}) \geq K\} \prod_{k=1}^{h-1} \phi_k(a_k | x_k, b_{1:k \wedge \tau_K})$$

$$\times \sum_{t=1}^T \prod_{k=1}^{h-1} \pi_i^t(b_k | x_k) \left\langle \pi_i^t(\cdot | x_h) - \phi_h(\cdot | x_h, b_{1:h \wedge \tau_K}), L_{i,h}^t(x_h, \cdot) \right\rangle.$$

For $I_h$, since all non-zero terms in the summation satisfy $D(a_{1:h-1}, b_{1:h-1}) \leq K - 1$, at step $h$, the number of deviations is less than $K$, i.e. $\tau_K \geq h$. Moreover, max over $\phi \in \Phi_i^K$ can be separated into max over all $\phi_h(\cdot|x_h, b_{1:h \wedge \tau_K})$, so we have

$$\max_{\phi \in \Phi_i^K} I_h = \max_{\phi \in \Phi_i^K} \sum_{x_h \in \mathcal{X}_{i,h}} \sum_{b_{1:h-1}} \delta^{\leq K-1}(a_{1:h-1}, b_{1:h-1}) \prod_{k=1}^{h-1} \phi_k(a_k|x_k, b_{1:k \wedge \tau_K})$$
$$\times \sum_{t=1}^{T} \prod_{k=1}^{h-1} \pi_i^t(b_k|x_k) \Big\langle \pi_i^t(\cdot|x_h) - \phi_h(\cdot|x_h, b_{1:h \wedge \tau_K}), L_{i,h}^t(x_h, \cdot) \Big\rangle$$

$$\leq \max_{\phi \in \Phi_i^K} \sum_{x_h \in \mathcal{X}_{i,h}} \sum_{b_{1:h-1}} \delta^{\leq K-1}(a_{1:h-1}, b_{1:h-1}) \prod_{k=1}^{h-1} \phi_k(a_k|x_k, b_{1:k})$$
$$\times \max_{\varphi} \sum_{t=1}^{T} \prod_{k=1}^{h-1} \pi_i^t(b_k|x_k) \Big\langle \pi_i^t(\cdot|x_h) - (\varphi \diamond \pi_i^t)(\cdot|x_h), L_{i,h}^t(x_h, \cdot) \Big\rangle$$

$$\overset{(i)}{=} \max_{\phi \in \Phi_i^K} \sum_{x_h \in \mathcal{X}_{i,h}} \sum_{b_{1:h-1}} \delta^{\leq K-1}(a_{1:h-1}, b_{1:h-1}) \prod_{k=1}^{h-1} \phi_k(a_k|x_k, b_{1:k}) \widehat{R}_{(x_h, b_{1:h-1})}^{T, \mathrm{swap}}$$

$$= \max_{\phi \in \Phi_i^K} \sum_{x_h \in \mathcal{X}_{i,h}} G_h^{T, \mathrm{swap}}(x_h).$$

Above, (i) follows by the definition of the local swap regret in (21). For $II_h$, since all non-zero terms in the summation satisfy $D(a_{1:h-1}, b_{1:h-1}) \geq K$, all such $(a_{1:h-1}, b_{1:h-1})$ have already deviated $K$ times at some step $h' \in [K, h-1]$, i.e. $\tau_K = h'$. In this case, the recommended action at $x_h$ (i.e. $b_h \sim \pi_i^t(\cdot|x_h)$) cannot be observed. Thus for $II_h$, we have

$$\max_{\phi \in \Phi_i^K} II_h = \max_{\phi \in \Phi_i^K} \sum_{x_h \in \mathcal{X}_{i,h}} \sum_{b_{1:h-1}} \mathbf{1}\{D(a_{1:h-1}, b_{1:h-1}) \geq K\} \prod_{k=1}^{h-1} \phi_k(a_k|x_k, b_{1:k \wedge \tau_K})$$
$$\times \sum_{t=1}^{T} \prod_{k=1}^{h-1} \pi_i^t(b_k|x_k) \Big\langle \pi_i^t(\cdot|x_h) - \phi_h(\cdot|x_h, b_{1:h \wedge \tau_K}), L_{i,h}^t(x_h, \cdot) \Big\rangle$$

$$= \max_{\phi \in \Phi_i^K} \sum_{x_h \in \mathcal{X}_{i,h}} \sum_{b_{1:h-1}} \sum_{h'=K}^{h-1} \mathbf{1}\{\tau_K = h'\} \prod_{k=1}^{h-1} \phi_k(a_k|x_k, b_{1:k \wedge h'})$$
$$\times \sum_{t=1}^{T} \prod_{k=1}^{h-1} \pi_i^t(b_k|x_k) \Big\langle \pi_i^t(\cdot|x_h) - \phi_h(\cdot|x_h, b_{1:h \wedge h'}), L_{i,h}^t(x_h, \cdot) \Big\rangle$$

$$= \max_{\phi \in \Phi_i^K} \sum_{x_h \in \mathcal{X}_{i,h}} \sum_{h'=K}^{h-1} \sum_{b_{1:h-1}} \mathbf{1}\{\tau_K = h'\} \prod_{k=1}^{h-1} \phi_k(a_k|x_k, b_{1:k \wedge h'})$$
$$\times \sum_{t=1}^{T} \prod_{k=1}^{h-1} \pi_i^t(b_k|x_k) \Big\langle \pi_i^t(\cdot|x_h) - \phi_h(\cdot|x_h, b_{1:h \wedge h'}), L_{i,h}^t(x_h, \cdot) \Big\rangle$$

$$\overset{(i)}{=} \max_{\phi \in \Phi_i^K} \sum_{x_h \in \mathcal{X}_{i,h}} \sum_{h'=K}^{h-1} \sum_{b_{1:h'}} \mathbf{1}\{\tau_K = h'\} \prod_{k=1}^{h-1} \phi_k(a_k|x_k, b_{1:k \wedge h'})$$
$$\times \sum_{t=1}^{T} \prod_{k=1}^{h'} \pi_i^t(b_k|x_k) \Big\langle \pi_i^t(\cdot|x_h) - \phi_h(\cdot|x_h, b_{1:h \wedge h'}), L_{i,h}^t(x_h, \cdot) \Big\rangle$$

$$\leq \max_{\phi \in \Phi_i^K} \sum_{x_h \in \mathcal{X}_{i,h}} \sum_{h'=K}^{h-1} \sum_{b_{1:h'}} \mathbf{1}\{\tau_K = h'\} \prod_{k=1}^{h-1} \phi_k(a_k|x_k, b_{1:k \wedge h'})$$
$$\times \max_{a} \sum_{t=1}^{T} \prod_{k=1}^{h'} \pi_i^t(b_k|x_k) \big(L_{i,h}^t(x_h, \pi_i^t(x_h)) - L_{i,h}^t(x_h, a)\big)$$

$$\overset{(ii)}{=} \max_{\phi \in \Phi_i^K} \sum_{h'=K}^{h-1} \sum_{b_{1:h'}} \mathbf{1}\{\tau_K = h'\} \prod_{k=1}^{h-1} \phi_k(a_k|x_k, b_{1:k \wedge h'}) \widehat{R}_{(x_h, b_{1:h'}), x_h}^{T, \mathrm{ext}}.$$

Here, (i) uses the fact that for fixed $x_h$, $h'$ and $b_{1:h'}$ with $\tau_K = h'$, there exists only one $b_{h'+1:h-1}$ satisfying $\tau_K = h$ and $\sum_{b_{h'+1:h-1}} \prod_{k=h'+1}^{h-1} \pi_i^t(b_k|x_k) = 1$; (ii) follows by definition of the local external regret in (22). Furthermore, for fixed $x_h$, we can expand $b_{1:h'}$ to $b_{1:h} := (b_{1:h'}, a_{h^*+1}, \cdots, a_{h-1})$ such that $\mathsf{D}(a_{1:h-1}, b_{1:h-1}) = K$. Then we can rewrite

$$
\sum_{h'=K}^{h-1} \sum_{b_{1:h'}} \mathbf{1}\left\{\tau_K = h'\right\} \prod_{k=1}^{h-1} \phi_k(a_k|x_k, b_{1:k \wedge h'}) \widehat{R}^{T,\text{ext}}_{(x_h, b_{1:h'}), x_h}
$$

$$
= \sum_{h'=K}^{h-1} \sum_{b_{1:h-1}} \mathbf{1}\left\{\tau_K = h', \mathsf{D}(a_{1:h-1}, b_{1:h-1}) = K\right\} \prod_{k=1}^{h-1} \phi_k(a_k|x_k, b_{1:k \wedge h'}) \widehat{R}^{T,\text{ext}}_{(x_h, b_{1:h'}), x_h}
$$

$$
= \sum_{b_{1:h-1}} \delta^K(a_{1:h-1}, b_{1:h-1}) \prod_{k=1}^{h-1} \phi_k(a_k|x_k, b_{1:k \wedge \tau_K}) \widehat{R}^{T,\text{ext}}_{(x_h, b_{1:\tau_K}), x_h} = G_h^{T,\text{ext}}(x_h).
$$

Consequently,

$$
\max_{\phi \in \Phi_i^K} \mathrm{II}_h \leq \max_{\phi \in \Phi_i^K} \sum_{x_h \in \mathcal{X}_{i,h}} G_h^{T,\text{ext}}(x_h).
$$

Finally, combining the above bounds for $\max_{\phi \in \Phi_i^K} \mathrm{I}_h$ and $\max_{\phi \in \Phi_i^K} \mathrm{II}_h$ gives the desired result:

$$
\widetilde{R}_h^T = \max_{\phi \in \Phi_i^K} (\mathrm{I}_h + \mathrm{II}_h) \leq \max_{\phi \in \Phi_i^K} \sum_{x_h \in \mathcal{X}_{i,h}} G_h^{T,\text{swap}}(x_h) + \max_{\phi \in \Phi_i^K} \sum_{x_h \in \mathcal{X}_{i,h}} G_h^{T,\text{ext}}(x_h) = R_h^T.
$$

$\square$

# F   Proofs for Section 4

This section is devoted to proving Theorem 3.

The proof follows by bounding the $K$-EFCE regret (6):

$$
R_{i,K}^T = \max_{\phi \in \Phi_i^K} \sum_{t=1}^T \left( V_i^{\phi \diamond \pi_i^t \times \pi_{-i}^t} - V_i^{\pi^t} \right)
$$

for all players $i \in [m]$, and then converting to a bound on $K$-EFCEGap$(\overline{\pi})$ by the online-to-batch conversion (Lemma E.1).

By the regret decomposition for $R_{i,K}^T$ (Lemma E.2), we have $R_{i,K}^T \leq \sum_{h=1}^H R_h^T$, where

$$
R_h^T := \max_{\phi \in \Phi_i^K} \sum_{x_h \in \mathcal{X}_{i,h}} G_h^{T,\text{swap}}(x_h; \phi) + \max_{\phi \in \Phi_i^K} \sum_{x_h \in \mathcal{X}_{i,h}} G_h^{T,\text{ext}}(x_h; \phi).
$$

The following two lemmas bound two terms

$$
\sum_{h=1}^H \max_{\phi \in \Phi_i^K} \sum_{x_h \in \mathcal{X}_{i,h}} G_h^{T,\text{swap}}(x_h; \phi) \quad \text{and} \quad \sum_{h=1}^H \max_{\phi \in \Phi_i^K} \sum_{x_h \in \mathcal{X}_{i,h}} G_h^{T,\text{ext}}(x_h; \phi).
$$

Their proofs are presented in Section F.1 & F.2 respectively.

**Lemma F.1** (Bound on summation of $G_h^{T,\text{swap}}(x_{i,h})$ with full feedback). *If we choose learning rates as*

$$
\eta_{x_h} = \sqrt{\binom{H}{K \wedge H} X_i A_i^{K \wedge H} \log A_i / (H^2 T)}
$$

*for all $x_h \in \mathcal{X}_{i,h}$ (same with (8)). Then we have*

$$
\sum_{h=1}^H \max_{\phi \in \Phi_i^K} \sum_{x_h \in \mathcal{X}_{i,h}} G_h^{T,\text{swap}}(x_h; \phi) \leq \sqrt{H^4 \binom{H}{K \wedge H} X_i A_i^{K \wedge H} T \log A_i}.
$$

**Lemma F.2** (Bound on summation of $G_h^{T,\text{ext}}(x_{i,h})$ with full feedback)**.** *If we choose learning rates as*

$$\eta_{x_h} = \sqrt{\binom{H}{K \wedge H} X_i A_i^{K \wedge h} \log A_i / (H^2 T)}$$

*for all $x_h \in \mathcal{X}_{i,h}$ (same with (8)). Then we have*

$$\sum_{h=1}^{H} \max_{\phi \in \Phi_i^K} \sum_{x_h \in \mathcal{X}_{i,h}} G_h^{T,\text{ext}}(x_h; \phi) \leq \mathcal{O}\left(\sqrt{H^4 \binom{H}{K \wedge H} X_i A_i^{K \wedge H} T \log A_i}\right).$$

Combining Lemma F.1 & F.2, we obtain regret bound

$$R_{i,K}^T \leq \sum_{h=1}^{H} R_h^T \leq \sum_{h=1}^{H} \left( \max_{\phi \in \Phi_i^K} \sum_{x_h \in \mathcal{X}_{i,h}} G_h^{T,\text{swap}}(x_h; \phi) + \max_{\phi \in \Phi_i^K} \sum_{x_h \in \mathcal{X}_{i,h}} G_h^{T,\text{ext}}(x_h; \phi) \right)$$

$$= \mathcal{O}\left(\sqrt{H^4 \binom{H}{K \wedge H} A_i^{K \wedge H} X_i T \log A_i}\right). \tag{25}$$

In particular, as long as

$$T \geq \mathcal{O}\left( H^4 \binom{H}{K \wedge H} \left( \max_{i \in [m]} X_i A_i^{K \wedge H} \right) \log A_i / \varepsilon^2 \right),$$

we have by the online-to-batch lemma (Lemma E.1) that the average policy $\overline{\pi} = \text{Unif}(\{\pi^t\}_{t=1}^T)$ satisfies

$$K\text{-EFCEGap}(\overline{\pi}) = \frac{\max_{i \in [m]} R_{i,K}^T}{T} \leq \max_{i \in [m]} \mathcal{O}\sqrt{H^4 \binom{H}{K \wedge H} \left( \max_{i \in [m]} X_i A_i^{K \wedge H} \right) \log A_i / T} \leq \varepsilon. \tag{26}$$

This proves Theorem 3. $\qquad\square$

We remark that the above proof does not depend on the particular choice of $\pi_{-i}^t$, and thus the regret bound (25) also holds even if we control the $i^{\text{th}}$ player only, and $\pi_{-i}^t$ are arbitrary (potentially adversarial depending on all information before iteration $t$ starts). This directly gives the following corollary.

**Corollary F.1** ($K$-EFCE regret bound for $K$-EFR against adversarial opponents)**.** *For any $0 \leq K \leq \infty$, $\varepsilon \in (0, H]$, suppose the $i^{\text{th}}$ player runs Algorithm 2 together against arbitrary (potentially adversarial) opponents $\left\{\pi_{-i}^t\right\}_{t=1}^T$, where REGALG is instantiated as Algorithm 4 with learning rates specified in (8). Then the $i^{\text{th}}$ player achieves $K$-EFCE regret bound:*

$$R_{i,K}^T = \max_{\phi \in \Phi_i^K} \sum_{t=1}^{T} \left( V_i^{\phi \diamond \pi_i^t \times \pi_{-i}^t} - V_i^{\pi^t} \right) \leq \mathcal{O}\left(\sqrt{H^4 \binom{H}{K \wedge H} X_i A_i^{K \wedge H} T \log A_i}\right).$$

## F.1 Proof of Lemma F.1

*Proof.* Recall that $G_h^{T,\text{swap}}(x_h; \phi)$ is defined as

$$G_h^{T,\text{swap}}(x_h; \phi) := \sum_{b_{1:h-1}} \delta^{\leq K-1}(a_{1:h-1}, b_{1:h-1}) \prod_{k=1}^{h-1} \phi_k(a_k|x_k, b_{1:k}) \widehat{R}_{(x_h, b_{1:h-1})}^{T,\text{swap}},$$

where for each $h \in [H]$ and $(x_h, b_{1:h-1}) \in \Omega_i^{(\text{I}),K}$,

$$\widehat{R}_{(x_h, b_{1:h-1})}^{T,\text{swap}} = \max_{\varphi} \sum_{t=1}^{T} \prod_{k=1}^{h-1} \pi_i^t(b_k|x_k) \left( \left\langle \pi_{i,h}^t(\cdot|x_{i,h}) - \varphi \diamond \pi_{i,h}^t(\cdot|x_{i,h}), L_{i,h}^t(x_{i,h}, \cdot) \right\rangle \right).$$

For $x_h \in \mathcal{X}_{i,h}$, we first apply regret minimization lemma (Lemma A.2) on $\mathcal{R}_{x_h}$ to give an upper bound on $\widehat{R}_{(x_h, b_{1:h-1})}^{T,\text{swap}}$. Recall in Algorithm 2 with REGALG instantiated as Algorithm 4, each $\mathcal{R}_{x_h}$

observes time selection functions $S_{b_{1:h-1}}^t = \prod_{k=1}^{h-1} \pi_i^t(b_k|x_k)$ for $(x_h, b_{1:h-1}) \in \Omega_i^{(\mathrm{I}),K}$ and loss vector $L_{i,h}^t(x_h, \cdot)$. Suppose $\mathcal{R}_{x_h}$ uses learning rate $\eta_{x_h} = \eta$ for all $x_h \in \mathcal{X}_i$. Then, by the regret bound with respect to a time selection function index and strategy modification pair (Lemma A.2), we have

$$\max_\varphi \sum_{t=1}^T \prod_{k=1}^{h-1} \pi_i^t(b_k|x_k) \Big( \big\langle \pi_{i,h}^t(\cdot|x_{i,h}) - \varphi \diamond \pi_{i,h}^t(\cdot|x_{i,h}), L_{i,h}^t(x_{i,h}, \cdot) \big\rangle \Big)$$

$$\leq \eta \sum_{t=1}^T \prod_{k=1}^{h-1} \pi_i^t(b_k|x_k) \| L_{i,h}^t(x_h, \cdot) \|_\infty \big\langle \pi_i^t(\cdot|x_h), L_{i,h}^t(x_h, \cdot) \big\rangle + \frac{(A_i + H) \log A_i}{\eta}$$

$$\leq \eta H^2 \sum_{t=1}^T \prod_{k=1}^{h-1} \pi_i^t(b_k|x_k) p_{1:h}^{\pi_{-i}^t}(x_h) + \frac{(A_i + H) \log A_i}{\eta}.$$

Above, we used (i) $\| L_{i,h}^t(x_h, \cdot) \|_\infty \leq H p_{1:h}^{\pi_{-i}^t}(x_h)$, and (ii) $(|\mathcal{B}^s| + |\mathcal{B}^e|)|\Psi^s| \leq A_i^{H+A_i}$ since the number of rechistories is no more than $A_i^H$. Then we can get

$$\max_{\phi \in \Phi_i^K} \sum_{x_h \in \mathcal{X}_{i,h}} G_h^{T,\mathrm{swap}}(x_h; \phi)$$

$$= \max_{\phi \in \Phi_i^K} \sum_{x_h \in \mathcal{X}_{i,h}} \sum_{b_{1:h-1}} \delta^{\leq K-1}(a_{1:h-1}, b_{1:h-1}) \prod_{k=1}^{h-1} \phi_k(a_k|x_k, b_{1:k})$$

$$\times \max_\varphi \sum_{t=1}^T \prod_{k=1}^{h-1} \pi_i^t(b_k|x_k) \Big( L_{i,h}^t(x_h, \pi_i^t(x_h)) - L_{i,h}^t(x_h, \varphi \diamond \pi_i^t(x_h)) \Big)$$

$$\leq \eta H^2 \max_{\phi \in \Phi_i^K} \sum_{x_h \in \mathcal{X}_{i,h}} \sum_{b_{1:h-1}} \delta^{\leq K-1}(a_{1:h-1}, b_{1:h-1}) \prod_{k=1}^{h-1} \phi_k(a_k|x_k, b_{1:k}) \sum_{t=1}^T \prod_{k=1}^{h-1} \pi_i^t(b_k|x_k) p_{1:h}^{\pi_{-i}^t}(x_h)$$

$$+ \frac{(A_i + H) \log A_i}{\eta} \max_{\phi \in \Phi_i^K} \sum_{x_h \in \mathcal{X}_{i,h}} \sum_{b_{1:h-1}} \delta^{\leq K-1}(a_{1:h-1}, b_{1:h-1}) \prod_{k=1}^{h-1} \phi_k(a_k|x_k, b_{1:k}).$$

Letting

$$\mathrm{I}_h := \max_{\phi \in \Phi_i^K} \sum_{x_h \in \mathcal{X}_{i,h}} \sum_{b_{1:h-1}} \delta^{\leq K-1}(a_{1:h-1}, b_{1:h-1}) \prod_{k=1}^{h-1} \phi_k(a_k|x_k, b_{1:k});$$

$$\mathrm{II}_h := \max_{\phi \in \Phi_i^K} \sum_{x_h \in \mathcal{X}_{i,h}} \sum_{b_{1:h-1}} \delta^{\leq K-1}(a_{1:h-1}, b_{1:h-1}) \prod_{k=1}^{h-1} \phi_k(a_k|x_k, b_{1:k}) \sum_{t=1}^T \prod_{k=1}^{h-1} \pi_i^t(b_k|x_k) p_{1:h}^{\pi_{-i}^t}(x_h).$$

For fixed $\phi \in \Phi_i^K$ and $x_h \in \mathcal{X}_{i,h}$, by counting the number of $b_{1:h-1}$ such that $\mathsf{D}(a_{1:h-1}, b_{1:h-1}) \leq K - 1$, we have

$$\sum_{b_{1:h-1}} \delta^{\leq K-1}(a_{1:h-1}, b_{1:h-1}) \prod_{k=1}^{h-1} \phi_k(a_k|x_k, b_{1:k})$$

$$\leq \sum_{b_{1:h-1}} \delta^{\leq K-1}(a_{1:h-1}, b_{1:h-1}) \leq \binom{h-1}{(K-1) \wedge (h-1)} A_i^{(K-1) \wedge (h-1)}.$$

Consequently,

$$\mathrm{I}_h \leq X_{i,h} \binom{h-1}{(K-1) \wedge (h-1)} A_i^{K \wedge h - 1}. \tag{27}$$

Note that by Corollary D.2, for fixed $t$,

$$\sum_{x_h \in \mathcal{X}_{i,h}} \sum_{b_{1:h-1}} \delta^{\leq K-1}(a_{1:h-1}, b_{1:h-1}) \prod_{k=1}^{h-1} \phi_k(a_k|x_k, b_{1:k}) \prod_{k=1}^{h-1} \pi_i^t(b_k|x_k) p_{1:h}^{\pi_{-i}^t}(x_{i,h}) \leq 1.$$

Consequently, we have

$$\max_{\phi \in \Phi_i^K} \sum_{x_h \in \mathcal{X}_{i,h}} \sum_{b_{1:h-1}} \delta^{\leq K-1}(a_{1:h-1}, b_{1:h-1}) \prod_{k=1}^{h-1} \phi_k(a_k|x_k, b_{1:k}) \sum_{t=1}^T \prod_{k=1}^{h-1} \pi_i^t(b_k|x_k) p_{1:h}^{\pi_{-i}^t}(x_{i,h}) \leq T.$$

This yields that

$$\text{II}_h \leq T. \tag{28}$$

Taking summation over $h \in [H]$, we have

$$\sum_{h=1}^{H} \max_{\phi \in \Phi_i^K} \sum_{x_h \in \mathcal{X}_{i,h}} G_h^{T,\text{swap}}(x_h; \phi) \leq \sum_{h=1}^{H} \left( \frac{(A_i + H)\log A_i}{\eta} \text{I}_h + H^2 \eta \text{II}_h \right)$$

$$\leq H^3 \eta T + \sum_{h=1}^{H} \frac{(A_i + H)\log A_i}{\eta} X_{i,h} \binom{h-1}{(K-1)\wedge(h-1)} A_i^{K \wedge h - 1}$$

$$\leq H^3 \eta T + \frac{(A_i + H)\log A_i}{\eta} X_i \binom{H-1}{K \wedge H - 1} A_i^{K \wedge H - 1}$$

$$\leq H^3 \eta T + \frac{H \log A_i}{\eta} X_i \binom{H}{K \wedge H} A_i^{K \wedge H}.$$

As we chose $\eta = \sqrt{\binom{H}{K \wedge H} X_i A_i^{K \wedge H} \log A_i / (H^2 T)}$ per (8), we have

$$\sum_{h=1}^{H} \max_{\phi \in \Phi_i^K} \sum_{x_h \in \mathcal{X}_{i,h}} G_h^{T,\text{swap}}(x_h; \phi) \leq \sqrt{H^4 \binom{H}{K \wedge H} X_i A_i^{K \wedge H} T \log A_i}.$$

$\square$

## F.2 Proof of Lemma F.2

*Proof.* Recall that $G_h^{T,\text{ext}}(x_h; \phi)$ is defined as

$$G_h^{T,\text{ext}}(x_h; \phi) := \sum_{b_{1:h-1}} \delta^K(a_{1:h-1}, b_{1:h-1}) \prod_{k=1}^{h-1} \phi_k(a_k | x_k, b_{1:k \wedge \tau_K}) \widehat{R}_{(x_h, b_{1:\tau_K})}^{T,\text{ext}}.$$

Since $(x_h, b_{1,\tau_K}) \in \Omega_i^{(\text{II}),K}$, we have

$$\widehat{R}_{(x_h, b_{1,\tau_K})}^{T,\text{ext}} = \max_a \sum_{t=1}^{T} \prod_{k=1}^{\tau_K} \pi_i^t(b_k | x_k) \left( \left\langle \pi_{i,h}^t(\cdot | x_{i,h}), L_{i,h}^t(x_{i,h}, \cdot) \right\rangle - L_{i,h}^t(x_{i,h}, a) \right).$$

For $x_h \in \mathcal{X}_{i,h}$, we can apply regret minimization lemma (Lemma A.2) on $\mathcal{R}_{x_h}$ give an upper bound on $\widehat{R}_{(x_h, b_{1:h-1})}^{T,\text{swap}}$. Recall in Algorithm 2 with REGALG instantiated as Algorithm 4, each regret minimizer $\mathcal{R}_{x_h}$ observes time selection functions $S_{b_{1:h_{\tau_K}}}^t = \prod_{k=1}^{h_{\tau_K}} \pi_i^t(b_k | x_k)$ for $(x_h, b_{1:h_{\tau_K}}) \in \Omega_i^{(\text{II}),K}$ and losses $L_{i,h}^t(x_h, \cdot)$. Suppose all $\mathcal{R}_{x_h}$ use the same learning rate $\eta$, by the bound on regret with respect to a time selection function index and strategy modification pair (Lemma A.2), we have

$$\max_a \sum_{t=1}^{T} \prod_{k=1}^{\tau_K} \pi_i^t(b_k | x_k) \left( \left\langle \pi_{i,h}^t(\cdot | x_{i,h}), L_{i,h}^t(x_{i,h}, \cdot) \right\rangle - L_{i,h}^t(x_{i,h}, a) \right)$$

$$\leq \eta \sum_{t=1}^{T} \prod_{k=1}^{\tau_K} \pi_i^t(b_k | x_k) \| L_{i,h}^t(x_h, \cdot) \|_\infty \left\langle \pi_i^t(\cdot | x_h), L_{i,h}^t(x_h, \cdot) \right\rangle + \frac{\log((|\mathcal{B}^s| + |\mathcal{B}^e|)|\Psi^e|)}{\eta}$$

$$\leq \eta H^2 \sum_{t=1}^{T} \prod_{k=1}^{\tau_K} \pi_i^t(b_k | x_k) p_{1:h}^{\pi_{-i}^t}(x_h) + \frac{2H \log A_i}{\eta}.$$

Here, we use (i) $\| L_{i,h}^t(x_h, \cdot) \|_\infty \leq H p_{1:h}^{\pi_{-i}^t}(x_h)$, and (ii) $(|\mathcal{B}^s| + |\mathcal{B}^e|)|\Psi^e| \leq A_i^{H+1}$ since the number of rechistories is no more than $A_i^H$. Then we can get

$$\max_{\phi \in \Phi_i^K} G_h^{T,\text{ext}}(x_h; \phi)$$

$$= \max_{\phi \in \Phi_i^K} \sum_{x_h \in \mathcal{X}_{i,h}} \sum_{b_{1:h-1}} \delta^K(a_{1:h-1}, b_{1:h-1}) \prod_{k=1}^{h-1} \phi_k(a_k|x_k, b_{1:k \wedge \tau_K})$$

$$\times \max_a \sum_{t=1}^{T} \prod_{k=1}^{\tau_K} \pi_i^t(b_k|x_k) \Big( \big\langle \pi_{i,h}^t(\cdot|x_{i,h}), L_{i,h}^t(x_{i,h}, \cdot) \big\rangle - L_{i,h}^t(x_{i,h}, a) \Big)$$

$$\leq \eta H^2 \max_{\phi \in \Phi_i^K} \sum_{x_h \in \mathcal{X}_{i,h}} \sum_{b_{1:h-1}} \delta^K(a_{1:h-1}, b_{1:h-1}) \prod_{k=1}^{h-1} \phi_k(a_k|x_k, b_{1:k \wedge \tau_K}) \cdot \sum_{t=1}^{T} \prod_{k=1}^{\tau_K} \pi_i^t(b_k|x_k) p_{1:h}^{\pi_{-i}^t}(x_h)$$

$$+ \frac{2H \log A_i}{\eta} \max_{\phi \in \Phi_i^K} \sum_{x_h \in \mathcal{X}_{i,h}} \sum_{b_{1:h-1}} \delta^K(a_{1:h-1}, b_{1:h-1}) \prod_{k=1}^{h-1} \phi_k(a_k|x_k, b_{1:k \wedge \tau_K}).$$

Similar to the proof of Lemma F.1, letting

$$\mathrm{I}_h := \max_{\phi \in \Phi_i^K} \sum_{x_h \in \mathcal{X}_{i,h}} \sum_{b_{1:h-1}} \delta^K(a_{1:h-1}, b_{1:h-1}) \prod_{k=1}^{h-1} \phi_k(a_k|x_k, b_{1:k \wedge h'});$$

$$\mathrm{II}_h := \max_{\phi \in \Phi_i^K} \sum_{x_h \in \mathcal{X}_{i,h}} \sum_{b_{1:h-1}} \delta^K(a_{1:h-1}, b_{1:h-1}) \prod_{k=1}^{h-1} \phi_k(a_k|x_k, b_{1:k \wedge h'}) \cdot \sum_{t=1}^{T} \prod_{k=1}^{h'} \pi_i^t(b_k|x_k) p_{1:h}^{\pi_{-i}^t}(x_h).$$

For fixed $\phi \in \Phi_i^K$ and $x_h \in \mathcal{X}_{i,h}$, by counting the number of $b_{1:h-1}$ such that $\mathsf{D}(a_{1:h-1}, b_{1:h-1}) = K$, we have

$$\sum_{b_{1:h-1}} \delta^K(a_{1:h-1}, b_{1:h-1}) \prod_{k=1}^{h-1} \phi_k(a_k|x_k, b_{1:k \wedge h'}) \leq \sum_{b_{1:h-1}} \delta^K(a_{1:h-1}, b_{1:h-1}) \leq \binom{h-1}{K \wedge h} A_i^{K \wedge (h-1)}.$$

Consequently,

$$\mathrm{I}_h \leq X_{i,h} \binom{h-1}{K \wedge h} A_i^{K \wedge (h-1)}. \tag{29}$$

Note that for fixed $t$, by Corollary D.2,

$$\sum_{x_h \in \mathcal{X}_{i,h}} \sum_{b_{1:h-1}} \delta^K(a_{1:h-1}, b_{1:h-1}) \prod_{k=1}^{h-1} \phi_k(a_k|x_k, b_{1:k \wedge h'}) \prod_{k=1}^{h'} \pi_i^t(b_k|x_k) p_{1:h}^{\pi_{-i}^t}(x_{i,h}) \leq 1.$$

Consequently, we have

$$\max_{\phi \in \Phi_i^K} \sum_{x_h \in \mathcal{X}_{i,h}} \sum_{b_{1:h-1}} \delta^K(a_{1:h-1}, b_{1:h-1}) \prod_{k=1}^{h-1} \phi_k(a_k|x_k, b_{1:k \wedge h'}) \sum_{t=1}^{T} \prod_{k=1}^{h'} \pi_i^t(b_k|x_k) p_{1:h}^{\pi_{-i}^t}(x_{i,h}) \leq T.$$

This yields that

$$\mathrm{II}_h \leq T. \tag{30}$$

Taking summation over $h \in [H]$, we have

$$\sum_{h=1}^{H} \max_{\phi \in \Phi_i^K} G_h^{T, \mathrm{ext}}(x_h; \phi) \leq \sum_{h=1}^{H} \left( \frac{2H \log A_i}{\eta} \mathrm{I}_h + H^2 \eta \mathrm{II}_h \right)$$

$$\leq H^3 \eta T + \sum_{h=1}^{H} \frac{2H \log A_i}{\eta} X_{i,h} \binom{h-1}{K \wedge h} A_i^{K \wedge H}$$

$$\leq H^3 \eta T + \frac{2H \log A_i}{\eta} X_i \binom{H-1}{K \wedge H} A_i^{K \wedge H}$$

$$\leq H^3 \eta T + \frac{2H \log A_i}{\eta} X_i \binom{H}{K \wedge H} A_i^{K \wedge H}.$$

As we choose $\eta = \sqrt{\binom{H}{K \wedge H} A_i^{K \wedge H} X_i \log A_i / (H^2 T)}$ per (8), we have

$$\sum_{h=1}^{H} \max_{\phi \in \Phi_i^K} G_h^{T, \mathrm{ext}}(x_h; \phi) \leq 3 \sqrt{H^4 \binom{H}{K \wedge H} X_i A_i^{K \wedge H} T \log A_i}.$$

$\square$

**Algorithm 6** Sample-based loss estimator for Type-I rechistories ($i^{\text{th}}$ player's version)

---

**Input:** Policy $\pi_i^t, \pi_{-i}^t$. Balanced exploration policies $\{\pi_i^{\star,h}\}_{h\in[H]}$.

1: **for** $1 \le h \le H, \overline{W} \subseteq [h-1]$ with $|\overline{W}| = (K-1) \wedge (h-1)$ **do**

2:     Set policy $\pi_i^{t,(h,\overline{W})} \leftarrow (\pi_{i,k}^{\star,h})_{k\in\overline{W}\cup\{h\}} \cdot (\pi_{i,k}^t)_{k\in[h-1]\setminus\overline{W}} \cdot \pi_{i,(h+1):H}^t$.

3:     Play $\pi_i^{t,(h,\overline{W})} \times \pi_{-i}^t$ for one episode, observe trajectory

$$(x_{i,1}^{t,(h,\overline{W})}, a_{i,1}^{t,(h,\overline{W})}, r_{i,1}^{t,(h,\overline{W})}, \dots, x_{i,H}^{t,(h,\overline{W})}, a_{i,H}^{t,(h,\overline{W})}, r_{i,H}^{t,(h,\overline{W})}).$$

4: **for** all $(x_{i,h}, b_{1:h-1}) \in \Omega_i^{(\mathrm{I}),K}$ **do**

5:     Find $(x_{i,1}, a_1) \prec \cdots \prec (x_{i,h-1}, a_{h-1}) \prec x_{i,h}$.

6:     Set $\overline{W} \leftarrow \mathrm{fill}(\{k \in [h-1] : b_k \ne a_k\}, (K-1) \wedge (h-1))$.

7:     Construct loss estimator for all $a \in \mathcal{A}_i$

$$\widetilde{L}_{(x_{i,h},b_{1:h-1})}^t(a) \leftarrow \frac{\mathbf{1}\left\{(x_{i,h}^{t,(h,\overline{W})}, a_{i,h}^{t,(h,\overline{W})}) = (x_{i,h}, a)\right\}}{\pi_{i,1:h}^{t,(h,\overline{W})}(x_{i,h}, a)} \cdot \sum_{h''=h}^{H} \left(1 - r_{i,h''}^{t,(h,\overline{W})}\right). \tag{32}$$

**Output:** Loss estimators $\left\{\widetilde{L}_{(x_{i,h},b_{1:h-1})}^t(\cdot)\right\}_{(x_{i,h},b_{1:h-1})\in\Omega_i^{(\mathrm{I}),K}}$.

---

# G Proofs for Section 5

This section is devoted to proving Theorem 5. The additional notation presented at the beginning of Section F is also used in this section.

## G.1 Sample-based loss estimator for Type-I rechistories

We first present the sample-based loss estimator for Type-I rechistories in Algorithm 6, complementary to the Type-II case presented in the main text (Algorithm 3). Here, Line 6 uses the "fill operator" defined as follows: For any index set $I \subset \mathbb{Z}_{\ge 1}$ and $n' \ge |I|$, $\mathrm{fill}(I, n')$ is defined as the unique superset of $I$ with size $n'$ and the smallest possible additional elements, i.e.

$$\mathrm{fill}(I, n') := I \cup \{I_{(1)}^c, \dots, I_{(n'-|I|)}^c\}, \tag{31}$$

where $I^c := \mathbb{Z}_{\ge 1} \setminus I$ with sorted elements $I_{(1)}^c < I_{(2)}^c < \cdots$. For example, $\mathrm{fill}(\{1,3,8\}, 5) = \{1,3,8\} \cup \{2,4\} = \{1,2,3,4,8\}$.

## G.2 Algorithm description of Balanced $K$-EFR

Algorithm 7 presents the detailed description of Balanced $K$-EFR.

For each infoset $x_{i,h} \in \mathcal{X}_i$, the algorithm computes time selection functions $\{S_{\mathbf{b}}^t\}_{\mathbf{b}\in\Omega_i^{(\mathrm{I}),K}(x_{i,h})\cup\Omega_i^{(\mathrm{II}),K}(x_{i,h})}$, based on the corresponding $M^t$ (defined in Line 5 & 6), as well as the following additional weighting function (below $W := \{k \in [h-1] : a_k \ne b_k\}$, and $\mathrm{fill}(\cdot, \cdot)$ is defined in (31))

$$w_{b_{1:h-1}}(x_{i,h}) = \prod_{k\in\mathrm{fill}(W,(h-1)\wedge(K-1))\cup\{h\}} \pi_{i,k}^{\star,h}(a_k|x_k), \text{ for all } b_{1:h-1} \in \Omega_i^{(\mathrm{I}),K}(x_{i,h}), \tag{33}$$

$$w_{b_{1:h'}}(x_{i,h}) = \prod_{k\in W\cup\{h'+1,\cdots,h\}} \pi_{i,k}^{\star,h}(a_k|x_k), \text{ for all } b_{1:h'} \in \Omega_i^{(\mathrm{II}),K}(x_{i,h}). \tag{34}$$

The resulting choice of time selection functions, $S_{\mathbf{b}}^t = M_{\mathbf{b}}^t w_{\mathbf{b}}^t(x_{i,h})$, is different from Algorithm 2, and is needed for this sampled case.

**Self-play protocol**   Here we explain the protocol of how we let all players play Algorithm 7 for $T$ rounds via self-play in Theorem 5. Within each round, each player first determines her own policy $\pi_i^t$ by Line 3-8. Then, we let all players compute their sample-based loss estimators in a round-robin fashion: The first player obtains her loss estimators first (Line 9) by playing the sampling policies

within Algorithm 6 & 3, in which case all other players keep playing $\pi_{-1}^t$. Then the same procedure goes on for players $2, \ldots, m$. Note that overall, each round plays $m$ times the number of episodes required for each player (specified by Algorithm 6 & 3). The following Lemma gives a bound on this number of episodes.

**Lemma G.1** (Number of episodes played by sampling algorithms). *One call of Algorithm 6 and Algorithm 3 plays (combinedly)* $\binom{H+1}{K \wedge H+1} + K \wedge H - 1 \leq 3H\binom{H}{K \wedge H}$ *episodes.*

*Proof.* The proof follows by counting the number of $\overline{W}$'s and $W$'s in the sampling algorithms, i.e. the cardinalities of

$$\mathcal{W}_{1,h} := \left\{ \overline{W} : \overline{W} \subseteq [h-1] \text{ with } |\overline{W}| = (K-1) \wedge (h-1) \right\}$$

for $1 \leq h \leq H$, which comes from Line 1 in Algorithm 6, and

$$\mathcal{W}_{2,h',h} := \left\{ W : K \leq h' < h, W \subseteq [h'] \text{ with } |W| = K \text{ and ending in } h' \right\}$$

for $K \leq h \leq H$, which comes from line 1 in Algorithm 3.

For the first kind of sets , we have

$$\sum_{h=1}^{H} |\mathcal{W}_{1,h}| = \sum_{h=1}^{K \wedge H-1} \binom{h-1}{(K-1) \wedge (h-1)} + \sum_{h=K \wedge H}^{H} \binom{h-1}{(K-1) \wedge (h-1)}$$

$$= K \wedge H - 1 + \sum_{h=K}^{H} \binom{h-1}{K \wedge H - 1}$$

$$= K \wedge H - 1 + \binom{H}{K \wedge H}.$$

For the second kind of sets , we have

$$\sum_{K \leq h' < h \leq H} |\mathcal{W}_{2,h',h}| = \sum_{K \leq h' < h \leq H} \binom{h'-1}{K-1}$$

$$= \sum_{h=K}^{H} \sum_{h'=K}^{h-1} \binom{h'-1}{K-1} = \sum_{h=K}^{H} \binom{h-1}{K}$$

$$= \binom{H}{K+1} = \binom{H}{K \wedge H + 1}.$$

Taking summation gives that the number of episodes equals

$$\sum_{h=1}^{H} |\mathcal{W}_{1,h}| + \sum_{K \leq h' < h \leq H} |\mathcal{W}_{2,h',h}| = \binom{H+1}{K \wedge H + 1} + K \wedge H - 1.$$

Finally, we show the above quantity can be upper bounded by $3H\binom{H}{K \wedge H}$. For $K \geq H$, we have $K \wedge H = H$ and the above quantity is $1 + H - 1 = H \leq 3H = 3H\binom{H}{K \wedge H}$. For $K < H$, we have $K \wedge H = K$, and thus

$$\binom{H+1}{K \wedge H + 1} + K \wedge H - 1 = \binom{H+1}{K+1} + K - 1 = \binom{H}{K} \cdot \frac{H+1}{K+1} + K - 1$$

$$\leq 2H \cdot \binom{H}{K} + H \leq 3H \cdot \binom{H}{K},$$

where the last inequality follows from the fact that $\binom{H}{K} \geq 1$. This is the desired bound. $\qquad\square$

**Algorithm 7** Balanced $K$-EFR ($i^{\text{th}}$ player's version)

---

**Input:** Weights $\{w_{b_{1:h-1}}(x_{i,h})\}_{x_{i,h},b_{1:h-1}\in\Omega_i^{(I),K}(x_{i,h})}$ and $\{w_{b_{1:h'}}(x_{i,h})\}_{x_{i,h},b_{1:h'}\in\Omega_i^{(II),K}(x_{i,h})}$ defined
   in (33), (34), learning rates $\{\eta_{x_{i,h}}\}_{x_{i,h}\in\mathcal{X}_i}$, loss upper bound $\overline{L} > 0$.
1: Initialize regret minimizers $\{\mathcal{R}_{x_{i,h}}\}_{x_{i,h}\in\mathcal{X}_i}$ with REGALG, learning rate $\eta_{x_{i,h}}$, and loss upper bound $\overline{L}$.
2: **for** iteration $t = 1, \ldots, T$ **do**
3:   **for** $h = 1, \ldots, H$ **do**
4:     **for** $x_{i,h} \in \mathcal{X}_{i,h}$ **do**
5:       $S_{b_{1:h-1}}^t := M_{b_{1:h-1}}^t w_{b_{1:h-1}}(x_{i,h})$ where $M_{b_{1:h-1}}^t := \prod_{k=1}^{h-1} \pi_{i,k}^t(b_k|x_k)$.
6:       $S_{b_{1:h'}}^t := M_{b_{1:h'}}^t w_{b_{1:h'}}(x_{i,h})$ where $M_{b_{1:h'}}^t := \prod_{k=1}^{h'} \pi_{i,k}^t(b_k|x_k)$.
7:       $\mathcal{R}_{x_{i,h}}.\text{OBSERVE\_TIMESELECTION}(\{S_{b_{1:h-1}}^t\}_{b_{1:h-1}\in\Omega_i^{(I),K}(x_{i,h})} \cup \{S_{b_{1:h'}}^t\}_{b_{1:h'}\in\Omega_i^{(II),K}(x_{i,h})})$.

8:       Set policy $\pi_i^t(\cdot|x_{i,h}) \leftarrow \mathcal{R}_{x_{i,h}}.\text{RECOMMEND}()$.
9:   Obtaining sample-based loss estimators
$$\left\{\widetilde{L}_{(x_{i,h},b_{1:h-1})}^t\right\}_{b_{1:h-1}\in\Omega_i^{(I),K}(x_{i,h})} \quad \text{and} \quad \left\{\widetilde{L}_{(x_{i,h},b_{1:h'})}^t\right\}_{b_{1:h'}\in\Omega_i^{(II),K}(x_{i,h})}$$
   from Algorithm 3 & 6 respectively.
10:  **for** all $x_{i,h} \in \mathcal{X}_i$ **do**
11:    $\mathcal{R}_{x_{i,h}}.\text{OBSERVE\_LOSS}(\{\widetilde{L}_{(x_{i,h},b_{1:h-1})}^t\}_{b_{1:h-1}\in\Omega_i^{(I),K}(x_{i,h})} \cup \{\widetilde{L}_{(x_{i,h},b_{1:h'})}^t\}_{b_{1:h'}\in\Omega_i^{(II),K}(x_{i,h})})$.
**Output:** Policies $\{\pi_i^t\}_{t\in[T]}$.

---

### G.3   Proof of Theorem 5

The proof follows a similar structure as the proof of Theorem 3 (cf. Section F), with different bounds
on the regret terms and bounds on additional concentration terms.

*Proof.* The proof follows by bounding the $K$-EFCE regret (6):

$$R_{i,K}^T = \max_{\phi\in\Phi_i^K} \sum_{t=1}^T \left( V_i^{\phi\diamond\pi_i^t\times\pi_{-i}^t} - V_i^{\pi^t} \right)$$

for all players $i \in [m]$, and then converting to a bound on $K$-EFCEGap$(\overline{\pi})$ by the online-to-batch
conversion (Lemma E.1).

By the regret decomposition for $R_{i,K}^T$ (Lemma E.2), we have $R_{i,K}^T \leq \sum_{h=1}^H R_h^T$, where

$$R_h^T := \max_{\phi\in\Phi_i^K} \sum_{x_h\in\mathcal{X}_{i,h}} G_h^{T,\text{swap}}(x_h;\phi) + \max_{\phi\in\Phi_i^K} \sum_{x_h\in\mathcal{X}_{i,h}} G_h^{T,\text{ext}}(x_h;\phi).$$

We bound the terms

$$\sum_{h=1}^H \max_{\phi\in\Phi_i^K} \sum_{x_h\in\mathcal{X}_{i,h}} G_h^{T,\text{swap}}(x_h;\phi) \quad \text{and} \quad \sum_{h=1}^H \max_{\phi\in\Phi_i^K} \sum_{x_h\in\mathcal{X}_{i,h}} G_h^{T,\text{ext}}(x_h;\phi)$$

when we play Balanced $K$-EFR (Algorithm 7) in the following two lemmas. Their proofs are
presented in Section G.4 & G.5 respectively.

**Lemma G.2** (Bound on summation of $G_h^{T,\text{swap}}(x_{i,h})$ with bandit feedback)**.** *If we choose learning
rates as*

$$\eta_{x_h} = \sqrt{\binom{H}{K\wedge H} X_i A_i^{K\wedge H+1} \iota / (H^3 T)}$$

*for all $x_h \in \mathcal{X}_i$ (same with (10)). With probability at least $1 - p/2$, we have*

$$\sum_{h=1}^H \max_{\phi\in\Phi_i^K} \sum_{x_h\in\mathcal{X}_{i,h}} G_h^{T,\text{swap}}(x_h;\phi) \leq \mathcal{O}\left( \sqrt{H^3 \binom{H}{K\wedge H} A_i^{K\wedge H+1} X_i T \iota} \right) + \mathcal{O}\left( H\binom{H}{K\wedge H} A_i^{K\wedge H+1} X_i \iota \right)$$

$$+\mathcal{O}\left(\binom{H}{K\wedge H}A_i^{K\wedge H+1}X_i\iota\sqrt{\frac{H\binom{H}{K\wedge H}A_i^{K\wedge H+1}X_i\iota}{T}}\right),$$

where $\iota = \log(8X_iA_i/p)$ is a log factor.

**Lemma G.3** (Bound on summation of $G_h^{T,\text{ext}}(x_{i,h})$ with bandit feedback)**.** *If we choose learning rates as*

$$\eta_{x_h} = \sqrt{\binom{H}{K\wedge H}X_iA_i^{K\wedge H+1}\iota/(H^3T)}$$

*for all $x_h \in \mathcal{X}_i$ (same with (10)). With probability at least $1 - p/2$, we have*

$$\sum_{h=1}^{H}\max_{\phi\in\Phi_i^K}\sum_{x_h\in\mathcal{X}_{i,h}}G_h^{T,\text{ext}}(x_h;\phi) \leq \mathcal{O}\left(\sqrt{H^3\binom{H}{K\wedge H}A_i^{K\wedge H+1}X_iT\iota}\right) + \mathcal{O}\left(H\binom{H}{K\wedge H}A_i^{K\wedge H+1}X_i\iota\right)$$

$$+\mathcal{O}\left(\binom{H}{K\wedge H}A_i^{K\wedge H+1}X_i\iota\sqrt{\frac{H\binom{H}{K\wedge H}A_i^{K\wedge H+1}X_i\iota}{T}}\right),$$

where $\iota = \log(8X_iA_i/p)$ is a log factor.

By Lemma G.2, G.3, and a union bound for all $i \in [m]$, we get

$$R_{i,K}^T \leq \sum_{h=1}^{H}R_h^T$$

$$\leq \sum_{h=1}^{H}\left(\max_{\phi\in\Phi_i^K}\sum_{x_h\in\mathcal{X}_{i,h}}G_h^{T,\text{swap}}(x_h;\phi) + \max_{\phi\in\Phi_i^K}\sum_{x_h\in\mathcal{X}_{i,h}}G_h^{T,\text{ext}}(x_h;\phi)\right)$$

$$\leq \mathcal{O}\left(\sqrt{H^3\binom{H}{K\wedge H}A_i^{K\wedge H+1}X_iT\iota} + H\binom{H}{K\wedge H}A_i^{K\wedge H+1}X_i\iota\right.$$

$$\left. + \binom{H}{K\wedge H}A_i^{K\wedge H+1}X_i\iota\sqrt{\frac{H\binom{H}{K\wedge H}A_i^{K\wedge H+1}X_i\iota}{T}}\right).$$

with probability at least $1 - p$ for all $i \in [m]$ simultaneously, where $\iota = \log(8\sum_{j\in[m]}X_jA_j/p)$.

Further using the "trivial" bound $R_{i,K}^T \leq HT$ (by the fact that $V_i^\pi \in [0, H]$ for any joint policy $\pi$) gives

$$R_{i,K}^T \stackrel{(i)}{\leq} HT\cdot\min\left\{1, \mathcal{O}\left(\sqrt{H\binom{H}{K\wedge H}X_iA_i^{K\wedge H+1}\iota/T}\right)\right\}$$

$$\leq \mathcal{O}\left(\sqrt{H^3\binom{H}{K\wedge H}A_i^{K\wedge H+1}X_iT\iota}\right),$$

where (i) follows by noticing that:

- if $T < H\binom{H}{K\wedge H}X_iA_i^{K\wedge H+1}\iota$, $R_{i,K}^T \leq HT = HT\min\left\{1, \mathcal{O}\left(\sqrt{H\binom{H}{K\wedge H}X_iA_i^{K\wedge H+1}\iota/T}\right)\right\}$;

- if $T \geq H\binom{H}{K\wedge H}X_iA_i^{K\wedge H+1}\iota$, $R_{i,K}^T \leq HT\cdot\mathcal{O}\left(\sqrt{H\binom{H}{K\wedge H}X_iA_i^{K\wedge H+1}\iota/T}\right)$.

Therefore, as long as

$$T \geq \mathcal{O}\left(H^3\binom{H}{K\wedge H}\left(\max_{i\in[m]}X_iA_i^{K\wedge H+1}\right)\iota/\varepsilon^2\right),$$

we have by the online-to-batch lemma (Lemma E.1) that the average policy $\overline{\pi} = \mathrm{Unif}(\{\pi^t\}_{t=1}^T)$ satisfies

$$K\text{-EFCEGap}(\overline{\pi}) = \frac{\max_{i\in[m]} R_{i,K}^T}{T} \leq \max_{i\in[m]} \mathcal{O}\sqrt{H^3 \binom{H}{K \wedge H}\left(\max_{i\in[m]} X_i A_i^{K\wedge H+1}\right)\iota/T} \leq \varepsilon.$$

This proves the first part of Theorem 5.

Finally, we count how many episodes are played at each iteration. By our self-play protocol (cf. Section G.2) and Lemma G.1, each iteration involves $m$ rounds of sampling (one for each player), where each round plays at most $3H\binom{H}{K\wedge H}$ episodes. Therefore, each iteration plays at most $3mH\binom{H}{K\wedge H}$ episodes, and so the total number of episodes played by Algorithm 7 is

$$3mH\binom{H}{K \wedge H} \cdot T = \mathcal{O}\left(mH^4 \binom{H}{K \wedge H}^2 \left(\max_{i\in[m]} X_i A_i^{K\wedge H+1}\right)\iota/\varepsilon^2\right).$$

This is the desired result. $\qquad\square$

### G.4   Proof of Lemma G.2

*Proof.* Recall that $G_h^{T,\mathrm{swap}}(x_h;\phi)$ is defined as (eq. (21))

$$G_h^{T,\mathrm{swap}}(x_h;\phi) := \sum_{b_{1:h-1}} \delta^{\leq K-1}(a_{1:h-1},b_{1:h-1}) \prod_{k=1}^{h-1} \phi_k(a_k|x_k,b_{1:k}) \widehat{R}_{(x_h,b_{1:h-1})}^{T,\mathrm{swap}}.$$

For each $h \in [H]$ and $(x_h, b_{1:h-1}) \in \Omega_i^{(\mathrm{I}),K}$, we have

$$\widehat{R}_{(x_h,b_{1:h-1})}^{T,\mathrm{swap}} = \max_{\varphi} \sum_{t=1}^T \prod_{k=1}^{h-1} \pi_i^t(b_k|x_k)\left( \left\langle \pi_{i,h}^t(\cdot|x_{i,h}) - \varphi \diamond \pi_{i,h}^t(\cdot|x_{i,h}), L_{i,h}^t(x_{i,h},\cdot)\right\rangle \right)$$

$$\leq \max_{\varphi} \sum_{t=1}^T \prod_{k=1}^{h-1} \pi_i^t(b_k|x_k)\left\langle \pi_{i,h}^t(\cdot|x_{i,h}) - \varphi \diamond \pi_{i,h}^t(\cdot|x_{i,h}), \widetilde{L}_{(x_h,b_{1:h-1})}^t\right\rangle$$

$$+ \sum_{t=1}^T \prod_{k=1}^{h-1} \pi_i^t(b_k|x_k)\left\langle \pi_{i,h}^t(\cdot|x_{i,h}), L_{i,h}^t(x_{i,h},\cdot) - \widetilde{L}_{(x_h,b_{1:h-1})}^t\right\rangle$$

$$+ \max_{\varphi} \sum_{t=1}^T \prod_{k=1}^{h-1} \pi_i^t(b_k|x_k)\left\langle \varphi \diamond \pi_{i,h}^t(\cdot|x_{i,h}), \widetilde{L}_{(x_h,b_{1:h-1})}^t - L_{i,h}^t(x_{i,h},\cdot)\right\rangle.$$

Substituting this into $\max_{\phi\in\Phi_i^K} \sum_{x_h\in\mathcal{X}_{i,h}} G_h^{T,\mathrm{swap}}(x_h;\phi)$ yields that

$$\max_{\phi\in\Phi_i^K} \sum_{x_h\in\mathcal{X}_{i,h}} G_h^{T,\mathrm{swap}}(x_h;\phi) = \max_{\phi\in\Phi_i^K} \sum_{x_h} \sum_{b_{1:h-1}} \delta^{\leq K-1}(a_{1:h-1},b_{1:h-1}) \prod_{k=1}^{h-1} \phi_k(a_k|x_k,b_{1:k}) \widehat{R}_{(x_h,b_{1:h-1})}^{T,\mathrm{swap}}$$

$$\leq \widetilde{\mathrm{REGRET}}_h^{T,\mathrm{swap}} + \mathrm{BIAS}_{1,h}^{T,\mathrm{swap}} + \mathrm{BIAS}_{2,h}^{T,\mathrm{swap}},$$

where

$$\widetilde{\mathrm{REGRET}}_h^{T,\mathrm{swap}} := \max_{\phi\in\Phi_i^K} \sum_{x_h\in\mathcal{X}_{i,h}} \sum_{b_{1:h-1}} \delta^{\leq K-1}(a_{1:h-1},b_{1:h-1}) \prod_{k=1}^{h-1} \phi_k(a_k|x_k,b_{1:k})$$

$$\times \max_{\varphi} \sum_{t=1}^T \prod_{k=1}^{h-1} \pi_i^t(b_k|x_k)\left\langle \pi_{i,h}^t(\cdot|x_{i,h}) - \varphi \diamond \pi_{i,h}^t(\cdot|x_{i,h}), \widetilde{L}_{(x_h,b_{1:h-1})}^t\right\rangle,$$

$$\mathrm{BIAS}_{1,h}^{T,\mathrm{swap}} := \max_{\phi\in\Phi_i^K} \sum_{x_h\in\mathcal{X}_{i,h}} \sum_{b_{1:h-1}} \delta^{\leq K-1}(a_{1:h-1},b_{1:h-1}) \prod_{k=1}^{h-1} \phi_k(a_k|x_k,b_{1:k})$$

$$\times \sum_{t=1}^T \prod_{k=1}^{h-1} \pi_i^t(b_k|x_k)\left\langle \pi_{i,h}^t(\cdot|x_{i,h}), L_{i,h}^t(x_{i,h},\cdot) - \widetilde{L}_{(x_h,b_{1:h-1})}^t\right\rangle$$

$$\mathrm{BIAS}_{2,h}^{T,\mathrm{swap}} := \max_{\phi \in \Phi_i^K} \sum_{x_h \in \mathcal{X}_{i,h}} \sum_{b_{1:h-1}} \delta^{\leq K-1}(a_{1:h-1}, b_{1:h-1}) \prod_{k=1}^{h-1} \phi_k(a_k|x_k, b_{1:k}),$$

$$\times \max_\varphi \sum_{t=1}^{T} \prod_{k=1}^{h-1} \pi_i^t(b_k|x_k) \left\langle \varphi \diamond \pi_{i,h}^t(\cdot|x_{i,h}), \widetilde{L}_{(x_h, b_{1:h-1})}^t - L_{i,h}^t(x_{i,h}, \cdot) \right\rangle.$$

The bounds for $\sum_{h=1}^{H} \widetilde{\mathrm{REGRET}}_h^{T,\mathrm{swap}}$, $\sum_{h=1}^{H} \mathrm{BIAS}_{1,h}^{T,\mathrm{swap}}$, and $\sum_{h=1}^{H} \mathrm{BIAS}_{2,h}^{T,\mathrm{swap}}$ are given in the following three lemmas (proofs deferred to Appendix G.4.1, G.4.2, and G.4.3) respectively.

**Lemma G.4** (Bound on $\widetilde{\mathrm{REGRET}}_h^{T,\mathrm{swap}}$). *If we choose learning rates as*

$$\eta_{x_h} = \sqrt{\binom{H}{K \wedge H} X_i A_i^{K \wedge H + 1} \log(X_i A_i/p)/(H^3 T)}.$$

*for all $x_h \in \mathcal{X}_i$ (same with (10)). Then with probability at least $1 - p/4$, we have*

$$\sum_{h=1}^{H} \widetilde{\mathrm{REGRET}}_h^{T,\mathrm{swap}} \leq \sqrt{H^3 \binom{H}{K \wedge H} A_i^{K \wedge H + 1} X_i T \iota}$$

$$+ \mathcal{O}\left( \binom{H}{K \wedge H} A_i^{K \wedge H} X_i \iota \sqrt{\frac{H \binom{H}{K \wedge H} A_i^{K \wedge H + 1} X_i \iota}{T}} \right)$$

*where $\iota = \log(8 X_i A_i/p)$ is a log factor.*

**Lemma G.5** (Bound on $\mathrm{BIAS}_{1,h}^{T,\mathrm{swap}}$). *With probability at least $1 - \frac{p}{8}$, we have*

$$\sum_{h=1}^{H} \mathrm{BIAS}_{1,h}^{T,\mathrm{swap}} \leq \mathcal{O}\left( \sqrt{H^3 \binom{H}{K \wedge H} A_i^{K \wedge H} X_i T \iota} + H \binom{H}{K \wedge H - 1} A_i^{K \wedge H} X_i \iota \right),$$

*where $\iota = \log(8 X_i A_i/p)$ is a log factor.*

**Lemma G.6** (Bound on $\mathrm{BIAS}_{2,h}^{T,\mathrm{swap}}$). *With probability at least $1 - \frac{p}{8}$, we have*

$$\sum_{h=1}^{H} \mathrm{BIAS}_{2,h}^{T,\mathrm{swap}} \leq \mathcal{O}\left( \sqrt{H^3 \binom{H}{K \wedge H} A_i^{K \wedge H + 1} X_i T \iota} + H \binom{H}{K \wedge H} A_i^{K \wedge H + 1} X_i \iota \right),$$

*where $\iota = \log(8 X_i A_i/p)$ is a log factor.*

Combining Lemma G.4, G.5, and G.6, we have with probability at least $1 - p/2$ that

$$\sum_{h=1}^{H} \max_{\phi \in \Phi_i^K} \sum_{x_h \in \mathcal{X}_{i,h}} G_h^{T,\mathrm{swap}}(x_h; \phi) \leq \sum_{h=1}^{H} \widetilde{\mathrm{REGRET}}_h^{T,\mathrm{swap}} + \sum_{h=1}^{H} \mathrm{BIAS}_{1,h}^{T,\mathrm{swap}} + \sum_{h=1}^{H} \mathrm{BIAS}_{2,h}^{T,\mathrm{swap}}$$

$$\leq \mathcal{O}\left( \sqrt{H^3 \binom{H}{K \wedge H} A_i^{K \wedge H + 1} X_i T \iota} \right)$$

$$+ \mathcal{O}\left( \binom{H}{K \wedge H} A_i^{K \wedge H} X_i \iota \sqrt{\frac{H \binom{H}{K \wedge H} A_i^{K \wedge H + 1} X_i \iota}{T}} \right)$$

$$+ \mathcal{O}\left( H \binom{H}{K \wedge H} A_i^{K \wedge H + 1} X_i \iota \right).$$

$\square$

### G.4.1 Proof of Lemma G.4: Bound on $\widetilde{\mathrm{REGRET}}_h^{T,\mathrm{swap}}$

*Proof.* Recall that $\widetilde{\mathrm{REGRET}}_h^{T,\mathrm{swap}}$ is defined as

$$\max_{\phi \in \Phi_i^K} \sum_{x_h \in \mathcal{X}_{i,h}} \sum_{b_{1:h-1}} \delta^{\leq K-1}(a_{1:h-1}, b_{1:h-1}) \prod_{k=1}^{h-1} \phi_k(a_k|x_k, b_{1:k})$$

$$\times \max_{\varphi} \sum_{t=1}^{T} \prod_{k=1}^{h-1} \pi_i^t(b_k|x_k) \Big\langle \pi_i^t(\cdot|x_h) - \varphi \diamond \pi_i^t(\cdot|x_h), \widetilde{L}_{(x_h, b_{1:h-1})}^t \Big\rangle.$$

We first apply regret minimization lemma (Lemma A.3) on $\mathcal{R}_{x_h}$ to give an upper bound of

$$\max_{\varphi} \sum_{t=1}^{T} \prod_{k=1}^{h-1} \pi_i^t(b_k|x_k) \Big\langle \pi_i^t(\cdot|x_h) - \varphi \diamond \pi_i^t(\cdot|x_h), \widetilde{L}_{(x_h, b_{1:h-1})}^t \Big\rangle.$$

Recall that in Algorithm 7, each regret minimizer $\mathcal{R}_{x_h}$ is associated with weights $M_{b_{1:h-1}}^t = \prod_{k=1}^{h-1} \pi_i^t(b_k|x_k)$ and $w_{b_{1:h-1}} = \prod_{k \in \mathrm{fill}(W, (h-1) \wedge (K-1)) \cup \{h\}} \pi_{i,k}^{\star,h}(a_k|x_k)$ for any $b_{1:h-1} \in \Omega_i^{(\mathrm{I}),K}(x_h)$, where $W = \{k \in [h-1] : b_k \neq a_k\}$. Let $\overline{W} = \mathrm{fill}(W, (h-1) \wedge (K-1))$, $\eta_{x_h} = \eta$ be the learning rate of $\mathcal{R}_{x_h}$ and $\overline{L} = H$. Since regret minimizers $\mathcal{R}_{x_h}$ observe $\left\{ \widetilde{L}_{(x_{i,h}, b_{1:h'})}^t(\cdot) \right\}_{(x_{i,h}, b_{1:h'}) \in \Omega_i^{(\mathrm{II}),K}}$ and $\left\{ \widetilde{L}_{(x_{i,h}, b_{1:h-1})}^t(\cdot) \right\}_{(x_{i,h}, b_{1:h-1}) \in \Omega_i^{(\mathrm{I}),K}}$ from Algorithm 3 & 6 as its loss vector at round $t$, we have

$$M_{b_{1:h-1}}^t w_{b_{1:h-1}} \widetilde{L}_{(x_{i,h}, b_{1:h-1})}^t(\cdot)$$

$$= M_{b_{1:h-1}}^t w_{b_{1:h-1}} \frac{\mathbf{1}\left\{ (x_{i,h}^{t,(h,\overline{W})}, a_{i,h}^{t,(h,\overline{W})}) = (x_{i,h}, \cdot) \right\}}{\prod_{k \in \overline{W} \cup \{h\}} \pi_{i,k}^{\star,h}(a_k|x_{i,k}) \prod_{k \in [h-1] \setminus \overline{W}} \pi_i^t(b_k|x_{i,k})} \cdot \sum_{h''=h}^{H} \left( 1 - r_{i,h''}^{t,(h,\overline{W})} \right)$$

$$= \prod_{k \in \overline{W}} \pi_i^t(b_k|x_{i,k}) \mathbf{1}\left\{ (x_{i,h}^{t,(h,\overline{W})}, a_{i,h}^{t,(h,\overline{W})}) = (x_{i,h}, \cdot) \right\} \sum_{h''=h}^{H} \left( 1 - r_{i,h''}^{t,(h,\overline{W})} \right)$$

$$\in [0, H] = [0, \overline{L}].$$

Similarly, we have $M_{b_{1:h'}}^t w_{b_{1:h-1}} \widetilde{L}_{(x_{i,h}, b_{1:h'})}(\cdot) \in [0, \overline{L}]$. Moreover, let $\mathcal{F}_{t-1}$ be the $\sigma$-algebra containing all the information until $\pi^t$ is sampled, by the sampling algorithm, we have that the loss estimators are unbiased:

$$\mathbb{E}\left[ \widetilde{L}_{(x_{i,h}, b_{1:h-1})}^t(\cdot)|\mathcal{F}_{t-1} \right] = \mathbb{E}\left[ \widetilde{L}_{(x_{i,h}, b_{1:h'})}^t(\cdot)|\mathcal{F}_{t-1} \right] = L_h^t(x_{i,h}, \cdot),$$

for all $(x_{i,h}, b_{1:h-1}) \in \Omega_i^{(\mathrm{I}),K}$ and all $(x_{i,h}, b_{1:h'}) \in \Omega_i^{(\mathrm{II}),K}$. So the assumptions in Lemma A.3 are satisfied. By Lemma A.3, with probability at least $1 - p/8$, for all $x_h \in \mathcal{X}_i$, we have

$$\max_{\varphi} \sum_{t=1}^{T} \prod_{k=1}^{h-1} \pi_i^t(b_k|x_k) \Big\langle \pi_i^t(\cdot|x_h) - \varphi \diamond \pi_i^t(\cdot|x_h), \widetilde{L}_{(x_h, b_{1:h-1})}^t \Big\rangle.$$

$$\leq \frac{2A_i \log(8X_i A_i/p)}{\eta w_{b_{1:h-1}}} + \eta \sum_{t=1}^{T} M_{b_{1:h-1}}^t \overline{L} \Big\langle \pi_i^t(\cdot|x_k), \widetilde{L}_{(x_h, b_{1:h-1})}^t(\cdot) \Big\rangle$$

$$= \frac{2A_i \log(8X_i A_i/p)}{\eta \prod_{k \in \overline{W} \cup \{h\}} \pi_{i,k}^{\star,h}(a_k|x_k)} + H\eta \prod_{k \in [h-1]} \pi_{i,k}^t(b_k|x_k)$$

$$\times \sum_{t=1}^{T} \Big\langle \pi_i^t(\cdot|x_k), \frac{\mathbf{1}\left\{ (x_h, \cdot) = (x_h^{t,(h,\overline{W})}, a_h^{t,(h,\overline{W})}) \right\} \left( H - h + 1 - \sum_{h'=h}^{H} r_{i,h'}^{t,(h,\overline{W})} \right)}{\prod_{k \in \overline{W} \cup \{h\}} \pi_{i,k}^{\star,h}(a_k|x_k) \prod_{k \in [h-1] \setminus \overline{W}} \pi_{i,k}^t(b_k|x_k)} \Big\rangle$$

$$\leq \frac{2A_i \log(8X_i A_i/p)}{\eta \prod_{k \in \overline{W} \cup \{h\}} \pi_{i,k}^{\star,h}(a_k|x_k)} + H^2\eta \sum_{t=1}^{T} \Big\langle \pi_i^t(\cdot|x_k), \frac{\prod_{k \in \overline{W}} \pi_{i,k}^t(b_k|x_k) \mathbf{1}\left\{ (x_h, \cdot) = (x_h^{t,(h,\overline{W})}, a_h^{t,(h,\overline{W})}) \right\}}{\prod_{k \in \overline{W} \cup \{h\}} \pi_{i,k}^{\star,h}(a_k|x_k)} \Big\rangle.$$

Above, we used $(i)$ our choices of $M^t_{b_{1:h-1}}$ and $w_{b_{1:h-1}}$; $(ii)$ $(|\mathcal{B}^s| + |\mathcal{B}^e|)|\Psi^s| \leq A_i^{H+A_i} \leq X_i A_i^{A_i}$ and $(iii)$ taking union bound over all infosets. Plugging this into $\widetilde{\mathrm{REGRET}}_h^{T,\mathrm{swap}}$, we have

$$\widetilde{\mathrm{REGRET}}_h^{T,\mathrm{swap}}$$

$$= \max_{\phi \in \Phi_i^K} \sum_{x_h \in \mathcal{X}_{i,h}} \sum_{b_{1:h-1}} \delta^{\leq K-1}(a_{1:h-1}, b_{1:h-1}) \prod_{k=1}^{h-1} \phi_k(a_k|x_k, b_{1:k})$$

$$\times \max_\varphi \sum_{t=1}^T \prod_{k=1}^{h-1} \pi_i^t(b_k|x_k) \Big\langle \pi_i^t(\cdot|x_h) - \varphi \diamond \pi_i^t(\cdot|x_h), \widetilde{L}^t_{(x_h, b_{1:h-1})} \Big\rangle.$$

$$\leq \frac{2A_i \log(8X_i A_i/p)}{\eta} \max_{\phi \in \Phi_i^K} \sum_{x_h \in \mathcal{X}_{i,h}} \sum_{b_{1:h-1}} \delta^{\leq K-1}(a_{1:h-1}, b_{1:h-1}) \frac{\prod_{k=1}^{h-1} \phi_k(a_k|x_k, b_{1:k})}{\prod_{k \in \overline{W} \cup \{h\}} \pi_{i,k}^{\star,h}(a_k|x_k)}$$

$$+ H^2 \eta \max_{\phi \in \Phi_i^K} \sum_{x_h \in \mathcal{X}_{i,h}} \sum_{b_{1:h-1}} \delta^{\leq K-1}(a_{1:h-1}, b_{1:h-1}) \frac{\prod_{k=1}^{h-1} \phi_k(a_k|x_k, b_{1:k})}{\prod_{k \in \overline{W} \cup \{h\}} \pi_{i,k}^{\star,h}(a_k|x_k)}$$

$$\times \underbrace{\sum_{t=1}^T \prod_{k \in \overline{W}} \pi_{i,k}^t(b_k|x_k) \Big\langle \pi_i^t(\cdot|x_k), \mathbf{1}\left\{(x_h, \cdot) = (x_h^{t,(h,\overline{W})}, a_h^{t,(h,\overline{W})})\right\} \Big\rangle}_{:=\overline{\Delta}_t^{x_h, b_{1:h-1}}}.$$

Letting

$$\mathrm{I}_h := \frac{2A_i \log(8X_i A_i/p)}{\eta} \max_{\phi \in \Phi_i^K} \sum_{x_h \in \mathcal{X}_{i,h}} \sum_{b_{1:h-1}} \delta^{\leq K-1}(a_{1:h-1}, b_{1:h-1}) \frac{\prod_{k=1}^{h-1} \phi_k(a_k|x_k, b_{1:k})}{\prod_{k \in \overline{W} \cup \{h\}} \pi_{i,k}^{\star,h}(a_k|x_k)};$$

$$\mathrm{II}_h := H^2 \eta \max_{\phi \in \Phi_i^K} \sum_{x_h \in \mathcal{X}_{i,h}} \sum_{b_{1:h-1}} \delta^{\leq K-1}(a_{1:h-1}, b_{1:h-1}) \frac{\prod_{k=1}^{h-1} \phi_k(a_k|x_k, b_{1:k})}{\prod_{k \in \overline{W} \cup \{h\}} \pi_{i,k}^{\star,h}(a_k|x_k)} \sum_{t=1}^T \overline{\Delta}_t^{x_h, b_{1:h-1}}.$$

Using Lemma B.4, we have

$$\sum_{x_h \in \mathcal{X}_{i,h}} \sum_{b_{1:h-1}} \delta^{\leq K-1}(a_{1:h-1}, b_{1:h-1}) \frac{\prod_{k=1}^{h-1} \phi_k(a_k|x_k, b_{1:k})}{\prod_{k \in \overline{W} \cup \{h\}} \pi_{i,k}^{\star,h}(a_k|x_k)}$$

$$= \sum_{x_h \in \mathcal{X}_{i,h}} \sum_{b_{1:h-1}} \delta^{\leq K-1}(a_{1:h-1}, b_{1:h-1}) \frac{\prod_{k=1}^{h-1} \phi_k(a_k|x_k, b_{1:k}) \prod_{k \in [h-1] \setminus \overline{W}} \pi_{i,k}^{\star,h}(a_k|x_k)}{\prod_{k \in [h]} \pi_{i,k}^{\star,h}(a_k|x_k)}$$

$$\overset{(i)}{=} \sum_{x_h \in \mathcal{X}_{i,h}} \sum_{a_h} \frac{\sum_{b_{1:h-1}} \delta^{\leq K-1}(a_{1:h-1}, b_{1:h-1}) \prod_{k=1}^{h-1} \phi_k(a_k|x_k, b_{1:k}) \prod_{k \in [h-1] \setminus \overline{W}} \pi_{i,k}^{\star,h}(b_k|x_k) \cdot \pi_{i,h}^{\star,h}(a_h|x_h)}{\prod_{k \in [h]} \pi_{i,k}^{\star,h}(a_k|x_k)}$$

$$\overset{(ii)}{=} \sum_{\overline{W} \subset [h-1], |\overline{W}|=(h-1)\wedge(K-1)} A_i^{|\overline{W}|} \times \sum_{x_h, a_h} \left(\prod_{k \in [h]} \pi_{i,k}^{\star,h}(a_k|x_k)\right)^{-1}$$

$$\times \sum_{b_{1:h-1}:\mathrm{fill}(\{k \in [h-1]:a_k \neq b_k\}, K \wedge H-1)=\overline{W}} \prod_{k=1}^{h-1} \phi_k(a_k|x_k, b_{1:k}) \prod_{k \in [h-1] \setminus \overline{W}} \pi_{i,k}^{\star,h}(b_k|x_k) \prod_{k \in \overline{W}} \pi_{i,k}^{\mathrm{unif}}(b_k|x_k) \pi_{i,h}^{\star,h}(a_h|x_h)$$

$$\overset{(iii)}{\leq} \sum_{\overline{W} \subset [h-1], |\overline{W}|=(K-1)\wedge(h-1)} A_i^{|\overline{W}|} X_{i,h} A_i$$

$$= X_{i,h} \binom{h-1}{(K-1) \wedge (h-1)} A_i^{K \wedge h}.$$

Here, (i) uses that $\pi_{i,h}^{\star,h}(\cdot|x_h)$ is uniform distribution on $\mathcal{A}_i$ and that for $k \in [h-1] \setminus \overline{W}$, we have $a_k = b_k$; (ii) follows from grouping $x_h$ and $b_{1:h}$ by $|\overline{W}|$, where $\pi_{i,k}^{\mathrm{unif}}$ is the uniform distribution on $\mathcal{A}_i$; (iii) uses Lemma B.4 and the fact that, for each fixed $\overline{W}$ (by relaxing the summation over $b_{1:h-1}$ to the full sum $\sum_{b_{1:h-1}}$) the numerator is no more than the sequence-form of the following policy:

- Sample recommended action $b_k$ from $\pi_{i,k}^{\star,h}(\cdot|x_k)$ if step $k \in \overline{W}$. Otherwise, sample recommended action $b_k$ from $\pi_{i,k}^{\mathrm{unif}}(\cdot|x_k)$.

- "True" actions are sampled from $\phi_k(a_k|x_k, b_{1:k})$ for $k \in [h-1]$. At step $h$, take action $a_h$.

Consequently,

$$\sum_{x_h \in \mathcal{X}_{i,h}} \sum_{b_{1:h-1}} \delta^{\leq K-1}(a_{1:h-1}, b_{1:h-1}) \frac{\prod_{k=1}^{h-1} \phi_k(a_k|x_k, b_{1:k})}{\prod_{k \in \overline{W} \cup \{h\}} \pi_{i,k}^{\star,h}(a_k|x_k)} \leq X_{i,h} \binom{h-1}{K \wedge h - 1} A_i^{K \wedge h}. \tag{35}$$

So we have

$$\mathrm{I}_h \leq \frac{2A_i \log(8X_i A_i/p)}{\eta} X_{i,h} \binom{h-1}{(K-1) \wedge (h-1)} A_i^{K \wedge h}.$$

For $\mathrm{II}_h$, observe that the random variables $\overline{\Delta}_t^{x_h, b_{1:h-1}}$ satisfy the following:

- $\overline{\Delta}_t^{x_h, b_{1:h-1}} = \prod_{k \in \overline{W}} \pi_{i,k}^t(b_k|x_k) \Big\langle \pi_i^t(\cdot|x_k), \mathbf{1}\Big\{(x_h, \cdot) = (x_h^{t,(h,\overline{W})}, a_h^{t,(h,\overline{W})})\Big\} \Big\rangle \in [0,1]$;

- Let $\mathcal{F}_{t-1}$ be the $\sigma$-algebra containing all information until $\pi^t$ is sampled, then

$$\mathbb{E}\Big[\overline{\Delta}_t^{x_h, b_{1:h-1}}|\mathcal{F}_{t-1}\Big]$$

$$= \prod_{k \in \overline{W}} \pi_{i,k}^t(b_k|x_k) \mathbb{E}\Bigg[\sum_{a \in \mathcal{A}_i} \pi_i^t(a|x_k) \mathbf{1}\Big\{(x_h, a) = (x_h^{t,(h,\overline{W})}, a_h^{t,(h,\overline{W})})\Big\} \Big| \mathcal{F}_{t-1}\Bigg]$$

$$= \prod_{k \in \overline{W}} \pi_{i,k}^t(b_k|x_k) \sum_{a \in \mathcal{A}_i} \pi_i^t(a|x_k) \mathbb{P}^{((\pi_{i,k}^{\star,h})_{k \in \overline{W} \cup \{h\}}(\pi_{i,k}^t)_{k \in [h-1] \setminus \overline{W}}) \times \pi_{-i}^t}(x_h^{t,(h,\overline{W})} = x_h, a_h^{t,(h,\overline{W})} = a)$$

$$= \prod_{k \in \overline{W}} \pi_{i,k}^t(b_k|x_k) \sum_{a \in \mathcal{A}_i} \pi_i^t(a|x_k) \Bigg(\prod_{k \in \overline{W}} \pi_{i,k}^{\star,h}(a_k|x_k) \cdot \prod_{k \in [h-1] \setminus \overline{W}} \pi_{i,k}^t(a_k|x_k) \cdot \pi_{i,h}^{\star,h}(a|x_h) p_{1:h}^{\pi_{-i}^t}(x_h)\Bigg)$$

$$= \prod_{k \in [h-1]} \pi_{i,k}^t(b_k|x_k) \cdot \prod_{k \in \overline{W} \cup \{h\}} \pi_{i,k}^{\star,h}(a_k|x_k) \cdot p_{1:h}^{\pi_{-i}^t}(x_h),$$

where the last equation is because $\prod_{k \in \overline{W}} \pi_{i,k}^t(b_k|x_k) \cdot \prod_{k \in [h-1] \setminus \overline{W}} \pi_{i,k}^t(a_k|x_k) = \prod_{k \in [h-1]} \pi_{i,k}^t(b_k|x_k)$;

- The conditional variance $\mathbb{E}[(\overline{\Delta}_t^{x_h, b_{1:h-1}})^2|\mathcal{F}_{t-1}]$ can be bounded as

$$\mathbb{E}\Bigg[\Big(\overline{\Delta}_t^{x_h, b_{1:h-1}}\Big)^2\Big|\mathcal{F}_{t-1}\Bigg] \leq \mathbb{E}\Big[\overline{\Delta}_t^{x_h, b_{1:h-1}}\Big|\mathcal{F}_{t-1}\Big]$$

$$= \prod_{k \in [h-1]} \pi_{i,k}^t(b_k|x_k) \prod_{k \in \overline{W} \cup \{h\}} \pi_{i,k}^{\star,h}(a_k|x_k) p_{1:h}^{\pi_{-i}^t}(x_h),$$

where we used $\overline{\Delta}_t^{x_h, b_{1:h-1}} \in [0,1]$ almost surely.

Therefore, we can apply Freedman's inequality (Lemma A.1) and union bound to get that for any fixed $\lambda \in (0,1]$, with probability at least $1 - p/8$, the following holds simultaneously for all $(h, x_{i,h}, b_{1:h-1})$:

$$\sum_{t=1}^T \overline{\Delta}_t^{x_h, b_{1:h-1}} \leq (\lambda + 1) \sum_{t=1}^T \prod_{k \in [h-1]} \pi_{i,k}^t(b_k|x_k) \prod_{k \in \overline{W} \cup \{h\}} \pi_{i,k}^{\star,h}(a_k|x_k) p_{1:h}^{\pi_{-i}^t}(x_h) + \frac{C \log(X_i A_i/p)}{\lambda},$$

where $C > 0$ is some absolute constant. Plugging this bound into $\mathrm{II}_h$ yields that,

$$\mathrm{II}_h \leq H^2 \eta \max_{\phi \in \Phi_i^K} \sum_{x_h \in \mathcal{X}_{i,h}} \sum_{b_{1:h-1}} \delta^{\leq K-1}(a_{1:h-1}, b_{1:h-1}) \frac{\prod_{k=1}^{h-1} \phi_k(a_k|x_k, b_{1:k})}{\prod_{k \in \overline{W} \cup \{h\}} \pi_{i,k}^{\star,h}(a_k|x_k)}$$

$$\times \left[ (\lambda + 1) \sum_{t=1}^{T} \prod_{k \in [h-1]} \pi_{i,k}^{t}(b_k|x_k) \prod_{k \in \overline{W} \cup \{h\}} \pi_{i,k}^{\star,h}(a_k|x_k) \cdot p_{1:h}^{\pi_{-i}^{t}}(x_h) + \frac{C \log(X_i A_i/p)}{\lambda} \right].$$

Note that by Corollary D.2, for fixed $t$ and any $\phi \in \Phi_i^K$,

$$\sum_{x_h \in \mathcal{X}_{i,h}} \sum_{b_{1:h-1}} \delta^{\leq K-1}(a_{1:h-1}, b_{1:h-1}) \prod_{k=1}^{h-1} \phi_k(a_k|x_k, b_{1:k}) \prod_{k \in [h-1]} \pi_{i,k}^{t}(b_k|x_k) \cdot p_{1:h}^{\pi_{-i}^{t}}(x_{i,h}) \leq 1.$$

so we have

$$\max_{\phi \in \Phi_i^K} \sum_{x_h \in \mathcal{X}_{i,h}} \sum_{b_{1:h-1}} \delta^{\leq K-1}(a_{1:h-1}, b_{1:h-1}) \prod_{k=1}^{h-1} \phi_k(a_k|x_k, b_{1:k}) \sum_{t=1}^{T} \prod_{k \in [h-1]} \pi_{i,k}^{t}(b_k|x_k) \cdot p_{1:h}^{\pi_{-i}^{t}}(x_{i,h}) \leq T.$$

Moreover, by the previous bound (35), for any $\phi \in \Phi_i^K$,

$$\sum_{x_h \in \mathcal{X}_{i,h}} \sum_{b_{1:h-1}} \delta^{\leq K-1}(a_{1:h-1}, b_{1:h-1}) \frac{\prod_{k=1}^{h-1} \phi_k(a_k|x_k, b_{1:k})}{\prod_{k \in \overline{W} \cup \{h\}} \pi_{i,k}^{\star,h}(a_k|x_k)} \leq X_{i,h} \binom{h-1}{K \wedge h - 1} A_i^{K \wedge h}.$$

Using these two inequalities, we can get that

$$\mathrm{II}_h \leq H^2 \eta (\lambda + 1) T + \frac{C H^2 \eta \log(X_i A_i/p)}{\lambda} X_{i,h} \binom{h-1}{K \wedge h - 1} A_i^{K \wedge h}.$$

Taking summation over $h \in [H]$, we have

$$\sum_{h=1}^{H} \widetilde{\mathrm{REGRET}}_h^{T,\mathrm{swap}} = \sum_{h=1}^{H} (\mathrm{I}_h + \mathrm{II}_h)$$

$$\leq H^3 \eta (\lambda + 1) T + \left( \frac{2 A_i \log(8 X_i A_i/p)}{\eta} + \frac{C H^2 \eta \log(X_i A_i/p)}{\lambda} \right) \sum_{h=1}^{H} X_{i,h} \binom{h-1}{(K-1) \wedge (h-1)} A_i^{K \wedge h}$$

$$\leq H^3 \eta (\lambda + 1) T + \left( \frac{2 A_i \log(8 X_i A_i/p)}{\eta} + \frac{C H^2 \eta \log(X_i A_i/p)}{\lambda} \right) X_i \binom{H-1}{K \wedge H - 1} A_i^{K \wedge H},$$

for all $\lambda \in (0, 1]$. Choosing $\lambda = 1$, we have,

$$\sum_{h=1}^{H} \widetilde{\mathrm{REGRET}}_h^{T,\mathrm{swap}} \leq 2 H^3 \eta T + \left( \frac{2 A_i \log(8 X_i A_i/p)}{\eta} + C H^2 \eta \log(X_i A_i/p) \right) X_i \binom{H-1}{K \wedge H - 1} A_i^{K \wedge H}.$$

Then, choosing $\eta = \sqrt{\binom{H}{K \wedge H} A_i^{K \wedge H + 1} X_i \iota / (H^3 T)}$ and using $\binom{H-1}{K \wedge H - 1} \leq \binom{H}{K \wedge H}$ we have

$$\sum_{h=1}^{H} \widetilde{\mathrm{REGRET}}_h^{T,\mathrm{swap}} \leq 2 \sqrt{H^3 \binom{H}{K \wedge H} A_i^{K \wedge H + 1} X_i T \iota}$$

$$+ \mathcal{O}\left( \binom{H}{K \wedge H} A_i^{K \wedge H} X_i \iota \sqrt{\frac{H \binom{H}{K \wedge H} A_i^{K \wedge H + 1} X_i \iota}{T}} \right)$$

with probability at least $1 - p/8$, where $\iota = \log(8 X_i A_i/p)$ is a log factor. $\square$

### G.4.2   Proof of Lemma G.5: Bound on $\mathrm{BIAS}_{1,h}^{T,\mathrm{swap}}$

*Proof.* We can rewrite $\mathrm{BIAS}_{1,h}^{T,\mathrm{swap}}$ as

$$\mathrm{BIAS}_{1,h}^{T,\mathrm{swap}}$$

$$= \max_{\phi \in \Phi_i^K} \sum_{x_h \in \mathcal{X}_{i,h}} \sum_{b_{1:h-1}} \delta^{\leq K-1}(a_{1:h-1}, b_{1:h-1}) \frac{\prod_{k=1}^{h-1} \phi_k(a_k|x_k, b_{1:k})}{\prod_{k \in \overline{W} \cup \{h\}} \pi_{i,k}^{\star,h}(a_k|x_k)}$$

$$\times \sum_{t=1}^{T} \underbrace{\prod_{k \in \overline{W} \cup \{h\}} \pi_{i,k}^{\star,h}(a_k|x_k) \prod_{k=1}^{h-1} \pi_i^t(b_k|x_k) \Big\langle \pi_{i,h}^t(\cdot|x_{i,h}), L_{i,h}^t(x_{i,h}, \cdot) - \widetilde{L}_{(x_h, b_{1:h-1})}^t \Big\rangle}_{:=\widetilde{\Delta}_t^{x_h, b_{1:h-1}}}.$$

Here, for fixed $x_h \in \mathcal{X}_i$ and $b_{1:h-1}$, let $W = \{k \in [h-1] : b_k \neq a_k\}$ and $\overline{W} = \mathrm{fill}(W, (h-1) \wedge (K-1))$. Observe that the random variables $\widetilde{\Delta}_t^{x_h, b_{1:h-1}}$ satisfy the following:

- By the definition of $\widetilde{L}$ in Algorithm 6, we can rewrite $\widetilde{\Delta}_t^{x_h, b_{1:h-1}}$ as

$$\widetilde{\Delta}_t^{x_h, b_{1:h-1}} = \prod_{k \in \overline{W} \cup \{h\}} \pi_{i,k}^{\star,h}(a_k|x_k) \prod_{k=1}^{h-1} \pi_i^t(b_k|x_k) \Big\langle \pi_{i,h}^t(\cdot|x_{i,h}), L_{i,h}^t(x_{i,h}, \cdot) \Big\rangle$$

$$- \prod_{k \in \overline{W}} \pi_{i,k}^t(b_k|x_k) \Big\langle \pi_{i,h}^t(\cdot|x_{i,h}), \mathbf{1}\Big\{(x_h, \cdot) = (x_h^{t,(h,\overline{W})}, a_h^{t,(h,\overline{W})})\Big\} \Big\rangle \cdot \Big(H - h + 1 - \sum_{h'=h}^{H} r_{i,h'}^{t,(h,\overline{W})}\Big)\Big\rangle;$$

- $\widetilde{\Delta}_t^{x_h, b_{1:h-1}} \leq \Big\langle \pi_{i,h}^t(\cdot|x_{i,h}), L_{i,h}^t(x_{i,h}, \cdot) \Big\rangle \leq H;$

- $\mathbb{E}[\widetilde{\Delta}_t^{x_h, b_{1:h-1}}|\mathcal{F}_{t-1}] = 0$, where $\mathcal{F}_{t-1}$ is the $\sigma$-algebra containing all information until $\pi^t$ is sampled. This also can be seen from the unbiasedness of $\widetilde{L};$

- The conditional variance $\mathbb{E}[(\widetilde{\Delta}_t^{x_h, b_{1:h-1}})^2|\mathcal{F}_{t-1}]$ can be bounded as

$$\mathbb{E}\Big[\Big(\widetilde{\Delta}_t^{x_h, b_{1:h-1}}\Big)^2\Big|\mathcal{F}_{t-1}\Big]$$

$$\leq \mathbb{E}\Big[\Big(H - h + 1 - \sum_{h'=h}^{H} r_{i,h'}^{t,(h,\overline{W})}\Big)^2 \Big(\prod_{k \in \overline{W}} \pi_{i,k}^t(b_k|x_k) \Big\langle \pi_{i,h}^t(\cdot|x_{i,h}), \mathbf{1}\Big\{(x_h, \cdot) = (x_h^{t,(h,\overline{W})}, a_h^{t,(h,\overline{W})})\Big\} \Big\rangle\Big)^2 \Big|\mathcal{F}_{t-1}\Big]$$

$$\overset{(i)}{\leq} H^2 \prod_{k \in \overline{W}} \pi_{i,k}^t(b_k|x_k) \cdot \mathbb{E}\Big[\Big\langle \pi_{i,h}^t(\cdot|x_{i,h}), \mathbf{1}\Big\{(x_h, \cdot) = (x_h^{t,(h,\overline{W})}, a_h^{t,(h,\overline{W})})\Big\} \Big\rangle\Big|\mathcal{F}_{t-1}\Big]$$

$$= H^2 \prod_{k \in \overline{W}} \pi_{i,k}^t(b_k|x_k) \cdot \mathbb{E}\Big[\sum_{a \in \mathcal{A}_i} \pi_i^t(a|x_k)\mathbf{1}\Big\{(x_h, a) = (x_h^{t,(h,\overline{W})}, a_h^{t,(h,\overline{W})})\Big\}\Big|\mathcal{F}_{t-1}\Big]$$

$$= H^2 \prod_{k \in \overline{W}} \pi_{i,k}^t(b_k|x_k) \cdot \sum_{a \in \mathcal{A}_i} \pi_i^t(a|x_k)\mathbb{P}^{((\pi_{i,k}^{\star,h})_{k \in \overline{W} \cup \{h\}} (\pi_{i,k}^t)_{k \in [h-1]\setminus \overline{W}}) \times \pi_{-i}^t}(x_h^{t,(h,\overline{W})} = x_h, a_h^{t,(h,\overline{W})} = a)$$

$$= H^2 \prod_{k \in \overline{W}} \pi_{i,k}^t(b_k|x_k) \cdot \sum_{a \in \mathcal{A}_i} \pi_i^t(a|x_k)\Big(\prod_{k \in \overline{W}} \pi_{i,k}^{\star,h}(a_k|x_k) \cdot \prod_{k \in [h-1]\setminus \overline{W}} \pi_{i,k}^t(a_k|x_k) \cdot \pi_{i,h}^{\star,h}(a|x_h)p_{1:h}^{\pi_{-i}^t}(x_h)\Big)$$

$$\overset{(ii)}{\leq} H^2 \prod_{k \in [h-1]} \pi_{i,k}^t(b_k|x_k)p_{1:h}^{\pi_{-i}^t}(x_{i,h}) \prod_{k \in \overline{W} \cup \{h\}} \pi_{i,k}^{\star,h}(a_k|x_k).$$

Here, (i) uses $\Big\langle \pi_{i,h}^t(\cdot|x_{i,h}), \mathbf{1}\Big\{(x_h, \cdot) = (x_h^{t,(h,\overline{W})}, a_h^{t,(h,\overline{W})})\Big\} \Big\rangle \in [0,1]$; (ii) is because $\prod_{k \in \overline{W}} \pi_{i,k}^t(b_k|x_k) \cdot \prod_{k \in [h-1]\setminus \overline{W}} \pi_{i,k}^t(a_k|x_k) = \prod_{k \in [h-1]} \pi_{i,k}^t(b_k|x_k)$.

Therefore, we can apply Freedman's inequality and union bound to get that for any fixed $\lambda \in (0, 1/H]$, with probability at least $1 - p/8$, the following holds simultaneously for all $(h, x_{i,h}, b_{1:h-1})$:

$$\sum_{t=1}^{T} \widetilde{\Delta}_t^{x_h, b_{1:h-1}} \leq \lambda H^2 \prod_{k \in \overline{W} \cup \{h\}} \pi_{i,k}^{\star,h}(a_k|x_k) \sum_{t=1}^{T} \prod_{k \in [h-1]} \pi_{i,k}^t(b_k|x_k)p_{1:h}^{\pi_{-i}^t}(x_{i,h}) + \frac{C\log(X_iA_i/p)}{\lambda},$$

where $C > 0$ is some absolute constant. Plugging this bound into $\mathrm{BIAS}_{1,h}^{T,\mathrm{swap}}$ yields that, for all $h \in [H]$,

$$\mathrm{BIAS}_{1,h}^{T,\mathrm{swap}} \leq \max_{\phi \in \Phi_i^K} \sum_{x_h \in \mathcal{X}_{i,h}} \sum_{b_{1:h-1}} \delta^{\leq K-1}(a_{1:h-1}, b_{1:h-1}) \frac{\prod_{k=1}^{h-1} \phi_k(a_k|x_k, b_{1:k})}{\prod_{k \in \overline{W} \cup \{h\}} \pi_{i,k}^{\star,h}(a_k|x_k)}$$

$$\times \left[ \lambda H^2 \prod_{k \in \overline{W} \cup \{h\}} \pi_{i,k}^{\star,h}(a_k|x_k) \sum_{t=1}^{T} \prod_{k \in [h-1]} \pi_{i,k}^t(b_k|x_k) p_{1:h}^{\pi_{-i}^t}(x_{i,h}) + \frac{C \log(X_i A_i/p)}{\lambda} \right]$$

$$\leq \lambda H^2 \max_{\phi \in \Phi_i^K} \sum_{x_h \in \mathcal{X}_{i,h}} \sum_{b_{1:h-1}} \delta^{\leq K-1}(a_{1:h-1}, b_{1:h-1}) \prod_{k=1}^{h-1} \phi_k(a_k|x_k, b_{1:k}) \sum_{t=1}^{T} \prod_{k \in [h-1]} \pi_{i,k}^t(b_k|x_k) p_{1:h}^{\pi_{-i}^t}(x_{i,h})$$

$$+ \frac{C \log(X_i A_i/p)}{\lambda} \max_{\phi \in \Phi_i^K} \sum_{x_h \in \mathcal{X}_{i,h}} \sum_{b_{1:h-1}} \delta^{\leq K-1}(a_{1:h-1}, b_{1:h-1}) \frac{\prod_{k=1}^{h-1} \phi_k(a_k|x_k, b_{1:k})}{\prod_{k \in \overline{W} \cup \{h\}} \pi_{i,k}^{\star,h}(a_k|x_k)}.$$

By Corollary D.2, for fixed $t$ and any $\phi \in \Phi_i^K$,

$$\sum_{x_h \in \mathcal{X}_{i,h}} \sum_{b_{1:h-1}} \delta^{=K-1}(a_{1:h-1}, b_{1:h-1}) \prod_{k=1}^{h-1} \phi_k(a_k|x_k, b_{1:k}) \prod_{k \in [h-1]} \pi_{i,k}^t(b_k|x_k) p_{1:h}^{\pi_{-i}^t}(x_{i,h}) \leq 1.$$

So we have

$$\lambda H^2 \max_{\phi \in \Phi_i^K} \sum_{x_h \in \mathcal{X}_{i,h}} \sum_{b_{1:h-1}} \delta^{\leq K-1}(a_{1:h-1}, b_{1:h-1}) \prod_{k=1}^{h-1} \phi_k(a_k|x_k, b_{1:k}) \sum_{t=1}^{T} \prod_{k \in [h-1]} \pi_{i,k}^t(b_k|x_k) p_{1:h}^{\pi_{-i}^t}(x_{i,h}) \leq \lambda H^2 T.$$

Moreover, by the inequality (35) in the proof of Lemma G.4, we have for any $\phi \in \Phi_i^K$,

$$\sum_{x_h \in \mathcal{X}_{i,h}} \sum_{b_{1:h-1}} \delta^{\leq K-1}(a_{1:h-1}, b_{1:h-1}) \frac{\prod_{k=1}^{h-1} \phi_k(a_k|x_k, b_{1:k})}{\prod_{k \in \overline{W} \cup \{h\}} \pi_{i,k}^{\star,h}(a_k|x_k)} \leq X_{i,h} \binom{h-1}{(K-1) \wedge (h-1)} A_i^{K \wedge h}.$$

Plugging these bounds into $\mathrm{BIAS}_{1,h}^{T,\mathrm{swap}}$ yields that, with probability at least $1 - p/8$, for all $h \in [H]$,

$$\mathrm{BIAS}_{1,h}^{T,\mathrm{swap}} \leq \lambda H^2 T + \frac{C \log(X_i A_i/p)}{\lambda} X_{i,h} \binom{h-1}{(K-1) \wedge (h-1)} A_i^{K \wedge h}.$$

Taking summation over $h \in [H]$, we have

$$\sum_{h=1}^{H} \mathrm{BIAS}_{1,h}^{T,\mathrm{swap}} \leq \lambda H^3 T + \frac{C \log(X_i A_i/p)}{\lambda} \sum_{h=1}^{H} X_{i,h} \binom{h-1}{(K-1) \wedge (h-1)} A_i^{K \wedge h}$$

$$\leq \lambda H^3 T + \frac{C \log(X_i A_i/p)}{\lambda} X_i \binom{H-1}{K \wedge H - 1} A_i^{K \wedge H}$$

$$\leq \lambda H^3 T + \frac{C \log(X_i A_i/p)}{\lambda} X_i \binom{H}{K \wedge H} A_i^{K \wedge H},$$

for all $\lambda \in (0, 1/H]$. Choose $\lambda = \min \left\{ \frac{1}{H}, \sqrt{\frac{C X_i \binom{H}{K \wedge H} A_i^{K \wedge H} \log(X_i A_i/p)}{H^3 T}} \right\}$, we obtain the bound

$$\sum_{t=1}^{T} \mathrm{BIAS}_{1,h}^{T,\mathrm{swap}} \leq \mathcal{O}\left( \sqrt{H^3 \binom{H}{K \wedge H} A_i^{K \wedge H} X_i T \iota} + H \binom{H}{K \wedge H} A_i^{K \wedge H} X_i \iota \right),$$

where $\iota = \log(8 X_i A_i/p)$ is a log factor. $\qquad\square$

### G.4.3 Proof of Lemma G.6: Bound on $\mathrm{BIAS}_{2,h}^{T,\mathrm{swap}}$

*Proof.* For fixed $x_h$ and $b_{1:h-1}$, let $W = \{k \in [h-1] : b_k \neq a_k\}$ and $\overline{W} = \mathrm{fill}(W, (K-1) \wedge (h-1))$. We can rewrite $\mathrm{BIAS}_{2,h}^{T,\mathrm{swap}}$ as

$$\mathrm{BIAS}_{2,h}^{T,\mathrm{swap}} = \max_{\phi \in \Phi_i^K} \sum_{x_h \in \mathcal{X}_{i,h}} \sum_{b_{1:h-1}} \delta^{\leq K-1}(a_{1:h-1}, b_{1:h-1}) \prod_{k=1}^{h-1} \phi_k(a_k|x_k, b_{1:k})$$

$$\times \max_{\varphi} \sum_{t=1}^{T} \prod_{k=1}^{h-1} \pi_i^t(b_k|x_k) \Big\langle \varphi \diamond \pi_{i,h}^t(\cdot|x_{i,h}), \widetilde{L}_{(x_h,b_{1:h-1})}^t - L_{i,h}^t(x_{i,h},\cdot) \Big\rangle$$

$$= \max_{\phi \in \Phi_i^K} \sum_{x_h \in \mathcal{X}_{i,h}} \sum_{b_{1:h-1}} \delta^{\leq K-1}(a_{1:h-1}, b_{1:h-1}) \prod_{k=1}^{h-1} \phi_k(a_k|x_k, b_{1:k})$$

$$\times \max_{\varphi} \sum_{t=1}^{T} \prod_{k=1}^{h-1} \pi_i^t(b_k|x_k) \sum_{b_h} \pi_{i,h}^t(b_h|x_h) \Big( \widetilde{L}_{(x_h,b_{1:h-1})}^t(\varphi(b_h)) - L_{i,h}^t(x_{i,h}, \varphi(b_h)) \Big)$$

$$\leq \max_{\phi \in \Phi_i^K} \sum_{x_h \in \mathcal{X}_i} \sum_{b_{1:h-1}} \delta^{\leq K-1}(a_{1:h-1}, b_{1:h-1}) \prod_{k=1}^{h-1} \phi_k(a_k|x_k, b_{1:k})$$

$$\times \max_{\phi_h'} \sum_{t=1}^{T} \prod_{k=1}^{h-1} \pi_i^t(b_k|x_k) \sum_{b_h, a_h} \pi_{i,h}^t(b_h|x_h) \phi_h'(a_h|x_h, b_{1:h}) \Big( \widetilde{L}_{(x_h,b_{1:h-1})}^t(a_h) - L_{i,h}^t(x_{i,h}, a_h) \Big)$$

$$\overset{(i)}{=} \max_{\phi \in \Phi_i^K} \sum_{x_{i,h}, a_h} \sum_{b_{1:h-1}} \sum_{b_h} \delta^{\leq K-1}(a_{1:h-1}, b_{1:h-1}) \prod_{k=1}^{h} \phi_k(a_k|x_k, b_{1:k})$$

$$\times \sum_{t=1}^{T} \prod_{k=1}^{h} \pi_i^t(b_k|x_k) \Big( \widetilde{L}_{(x_h,b_{1:h-1})}^t(a_h) - L_{i,h}^t(x_{i,h}, a_h) \Big)$$

$$= \max_{\phi \in \Phi_i^K} \sum_{x_{i,h}, a_h} \sum_{b_{1:h-1}} \sum_{b_h} \delta^{\leq K-1}(a_{1:h-1}, b_{1:h-1}) \frac{\prod_{k=1}^{h} \phi_k(a_k|x_k, b_{1:k})}{\prod_{k \in \overline{W} \cup \{h\}} \pi_{i,k}^{\star,h}(a_k|x_k)}$$

$$\times \underbrace{\sum_{t=1}^{T} \prod_{k=1}^{h} \pi_i^t(b_k|x_k) \Big( \widetilde{L}_{(x_h,b_{1:h-1})}^t(a_h) - L_{i,h}^t(x_{i,h}, a_h) \Big) \prod_{k \in \overline{W} \cup \{h\}} \pi_{i,k}^{\star,h}(a_k|x_k)}_{:=\widetilde{\Delta}_t^{x_h, b_{1:h}, a_h}}.$$

Here, (i) comes from the fact that the inner max over $\phi_h'$ and the outer max over $\phi_{1:h-1}$ are separable and thus can be merged into a single max over $\phi_{1:h}$.

Observe that the random variables $\widetilde{\Delta}_t^{x_{i,h}, b_{1:h}, a_h}$ satisfy the following:

- By the definition of $\widetilde{L}$ in Algorithm 6, we can rewrite $\widetilde{\Delta}_t^{x_h, b_{1:h}, a_h}$ as

$$\widetilde{\Delta}_t^{x_h, b_{1:h}, a_h}$$

$$= \prod_{k \in \overline{W} \cup \{h\}} \pi_{i,k}^t(b_k|x_k) \mathbf{1}\Big\{ (x_h, a_h) = (x_h^{t,(h,\overline{W})}, a_h^{t,(h,\overline{W})}) \Big\} \cdot \left( H - h + 1 - \sum_{h'=h}^{H} r_{i,h'}^{t,(h,\overline{W})} \right)$$

$$- \prod_{k \in [h]} \pi_{i,k}^t(b_k|x_k) \prod_{k \in \overline{W} \cup \{h\}} \pi_{i,k}^{\star,h}(a_k|x_k) L_{i,h}^t(x_{i,h}, a_h).$$

- $\widetilde{\Delta}_t^{x_h, b_{1:h}, a_h} \in [-H, H]$.

- $\mathbb{E}[\widetilde{\Delta}_t^{x_h, b_{1:h}, a_h}|\mathcal{F}_{t-1}] = 0$, where $\mathcal{F}_{t-1}$ is the $\sigma$-algebra containing all information until $\pi^t$ is sampled.

- The conditional variance $\mathbb{E}[(\widetilde{\Delta}_t^{x_h, b_{1:h}, a_h})^2|\mathcal{F}_{t-1}]$ can be bounded as

$$\mathbb{E}\left[ \left( \widetilde{\Delta}_t^{x_h, b_{1:h-1}, a_h} \right)^2 \Big| \mathcal{F}_{t-1} \right]$$

$$\leq \mathbb{E}\left[ \left( H - h + 1 - \sum_{h'=h}^{H} r_{i,h'}^{t,(h,\overline{W})} \right)^2 \left( \prod_{k \in \overline{W} \cup \{h\}} \pi_{i,k}^t(b_k|x_k) \mathbf{1}\Big\{ (x_h, a_h) = (x_h^{t,(h,\overline{W})}, a_h^{t,(h,\overline{W})}) \Big\} \right)^2 \Big| \mathcal{F}_{t-1} \right]$$

$$\leq H^2 \prod_{k \in \overline{W} \cup \{h\}} \pi_{i,k}^t(b_k|x_k) \mathbb{P}^{((\pi_{i,k}^{\star,h})_{k \in \overline{W} \cup \{h\}}(\pi_{i,k}^t)_{k \in [h-1] \setminus \overline{W}}) \times \pi_{-i}^t} \left( (x_h^{t,(h,\overline{W})}, a_h^{t,(h,\overline{W})}) = (x_h, a_h) \right)$$

$$= H^2 \prod_{k \in \overline{W} \cup \{h\}} \pi_{i,k}^t(b_k|x_k) \prod_{k \in \overline{W}} \pi_{i,k}^{\star,h}(a_k|x_k) \cdot \prod_{k \in [h-1] \setminus \overline{W}} \pi_{i,k}^t(a_k|x_k) \cdot \pi_{i,h}^{\star,h}(a_h|x_h) p_{1:h}^{\pi_{-i}^t}(x_{i,h})$$

$$\stackrel{(i)}{=} H^2 \prod_{k \in [h]} \pi_{i,k}^t(b_k|x_k) p_{1:h}^{\pi_{-i}^t}(x_{i,h}) \prod_{k \in \overline{W} \cup \{h\}} \pi_{i,k}^{\star,h}(a_k|x_k).$$

Here, (i) is because $\prod_{k \in \overline{W} \cup \{h\}} \pi_{i,k}^t(b_k|x_k) \cdot \prod_{k \in [h-1] \setminus \overline{W}} \pi_{i,k}^t(a_k|x_k) = \prod_{k \in [h]} \pi_{i,k}^t(b_k|x_k)$.

Therefore, we can apply Freedman's inequality and union bound to get that for any fixed $\lambda \in (0, 1/H]$, with probability at least $1 - p/8$, the following holds simultaneously for all $(h, x_{i,h}, b_{1:h}, a_h)$:

$$\sum_{t=1}^T \widetilde{\Delta}_t^{x_h, b_{1:h}, a_h} \leq \lambda H^2 \prod_{k \in \overline{W} \cup \{h\}} \pi_{i,k}^{\star,h}(a_k|x_k) \sum_{t=1}^T \prod_{k \in [h]} \pi_{i,k}^t(b_k|x_k) p_{1:h}^{\pi_{-i}^t}(x_{i,h}) + \frac{C \log(X_i A_i/p)}{\lambda},$$

where $C > 0$ is some absolute constant. Plugging this bound into $\mathrm{BIAS}_{2,h}^{T,\mathrm{swap}}$ yields that, for all $h \in [H]$ and $\phi \in \Phi_i^K$,

$$\mathrm{BIAS}_{2,h}^{T,\mathrm{swap}} \leq \max_{\phi \in \Phi_i^K} \sum_{x_h, a_h} \sum_{b_{1:h}} \delta^{\leq K-1}(a_{1:h-1}, b_{1:h-1}) \frac{\prod_{k=1}^h \phi_k(a_k|x_k, b_{1:k})}{\prod_{k \in \overline{W} \cup \{h\}} \pi_{i,k}^{\star,h}(a_k|x_k)}$$

$$\times \left[ \lambda H^2 \prod_{k \in \overline{W} \cup \{h\}} \pi_{i,k}^{\star,h}(a_k|x_k) \sum_{t=1}^T \prod_{k \in [h]} \pi_{i,k}^t(b_k|x_k) p_{1:h}^{\pi_{-i}^t}(x_{i,h}) + \frac{C \log(X_i A_i/p)}{\lambda} \right]$$

$$\leq \max_{\phi \in \Phi_i^K} \lambda H^2 \sum_{x_h, a_h} \sum_{b_{1:h}} \delta^{\leq K-1}(a_{1:h-1}, b_{1:h-1}) \prod_{k=1}^h \phi_k(a_k|x_k, b_{1:k}) \sum_{t=1}^T \prod_{k \in [h]} \pi_{i,k}^t(b_k|x_k) p_{1:h}^{\pi_{-i}^t}(x_{i,h})$$

$$+ \max_{\phi \in \Phi_i^K} \frac{C \log(X_i A_i/p)}{\lambda} \sum_{x_h, a_h} \sum_{b_{1:h}} \delta^{\leq K-1}(a_{1:h-1}, b_{1:h-1}) \frac{\prod_{k=1}^h \phi_k(a_k|x_k, b_{1:k})}{\prod_{k \in \overline{W} \cup \{h\}} \pi_{i,k}^{\star,h}(a_k|x_k)}.$$

By Corollary D.2, for fixed $t$ and any $\phi \in \Phi_i^K$,

$$\sum_{x_h \in \mathcal{X}_{i,h}} \sum_{b_{1:h-1}} \delta^{\leq K-1}(a_{1:h-1}, b_{1:h-1}) \prod_{k=1}^{h-1} \phi_k(a_k|x_k, b_{1:k}) \prod_{k \in [h-1]} \pi_{i,k}^t(b_k|x_k) p_{1:h}^{\pi_{-i}^t}(x_{i,h}) \leq 1.$$

so we have

$$\sum_{x_h, a_h} \sum_{b_{1:h}} \delta^{\leq K-1}(a_{1:h-1}, b_{1:h-1}) \prod_{k=1}^h \phi_k(a_k|x_k, b_{1:k}) \sum_{t=1}^T \prod_{k \in [h]} \pi_{i,k}^t(b_k|x_k) p_{1:h}^{\pi_{-i}^t}(x_{i,h})$$

$$= \sum_{x_h} \sum_{b_{1:h-1}} \delta^{\leq K-1}(a_{1:h-1}, b_{1:h-1}) \prod_{k=1}^{h-1} \phi_k(a_k|x_k, b_{1:k}) \sum_{t=1}^T \prod_{k \in [h-1]} \pi_{i,k}^t(b_k|x_k) p_{1:h}^{\pi_{-i}^t}(x_{i,h}) \leq T.$$

Moreover, by the inequality (35) in the proof of Lemma G.4, we have

$$\sum_{x_h, a_h} \sum_{b_{1:h}} \delta^{\leq K-1}(a_{1:h-1}, b_{1:h-1}) \frac{\prod_{k=1}^h \phi_k(a_k|x_k, b_{1:k})}{\prod_{k \in \overline{W} \cup \{h\}} \pi_{i,k}^{\star,h}(a_k|x_k)}$$

$$\stackrel{(i)}{=} A_i \sum_{x_h \in \mathcal{X}_{i,h}} \sum_{b_{1:h-1}} \delta^{\leq K-1}(a_{1:h-1}, b_{1:h-1}) \frac{\prod_{k=1}^{h-1} \phi_k(a_k|x_k, b_{1:k})}{\prod_{k \in \overline{W} \cup \{h\}} \pi_{i,k}^{\star,h}(a_k|x_k)}$$

$$\leq X_{i,h} \binom{h-1}{K \wedge H - 1} A_i^{K \wedge h + 1}.$$

Above, (i) sums over $a_h$ and $b_h$. Plugging these bounds into $\mathrm{BIAS}_{2,h}^{T,\mathrm{swap}}$ yields that, with probability at least $1 - p/8$, for all $h \in [H]$,

$$\mathrm{BIAS}_{2,h}^{T,\mathrm{swap}} \leq \lambda H^2 T + \frac{C \log(X_i A_i/p)}{\lambda} X_{i,h} \binom{h-1}{K \wedge H - 1} A_i^{K \wedge h + 1}.$$

Taking summation over $h \in [H]$, we have

$$\sum_{h=1}^H \mathrm{BIAS}_{2,h}^{T,\mathrm{swap}} \leq \lambda H^3 T + \frac{C \log(X_i A_i/p)}{\lambda} \sum_{h=1}^H X_{i,h} \binom{h-1}{(K-1) \wedge (h-1)} A_i^{K \wedge h + 1}$$

$$\leq \lambda H^3 T + \frac{C \log(X_i A_i/p)}{\lambda} X_i \binom{H-1}{K \wedge H - 1} A_i^{K \wedge H + 1}$$

$$\leq \lambda H^3 T + \frac{C \log(X_i A_i/p)}{\lambda} X_i \binom{H}{K \wedge H} A_i^{K \wedge H + 1},$$

for all $\lambda \in (0, 1/H]$. Choose $\lambda = \min\left\{\frac{1}{H}, \sqrt{\frac{C X_i \binom{H}{K \wedge H} A_i^{K \wedge H + 1} \log(X_i A_i/p)}{H^3 T}}\right\}$, we obtain the bound

$$\sum_{t=1}^{T} \text{BIAS}_{2,h}^{T,\text{swap}} \leq \mathcal{O}\left(\sqrt{H^3 \binom{H}{K \wedge H} A_i^{K \wedge H + 1} X_i T \iota} + H \binom{H}{K \wedge H} A_i^{K \wedge H + 1} X_i \iota\right),$$

where $\iota = \log(8 X_i A_i/p)$ is a log factor. $\qquad\square$

### G.5 Proof of Lemma G.3

*Proof.* Throughout the proof, for $x_h$ and $b_{1:h-1}$, let $W = \{k \in [h-1] : b_k \neq a_k\}$ and $h'$ is the maximal element in $W$, so we have $h' = \tau_K$. Recall that $G_h^{T,\text{ext}}(x_h; \phi)$ (eq. (22)) is defined as

$$G_h^{T,\text{ext}}(x_h; \phi) := \sum_{b_{1:h-1}} \delta^K(a_{1:h-1}, b_{1:h-1}) \prod_{k=1}^{h-1} \phi_k(a_k|x_k, b_{1:k \wedge \tau_K}) \widehat{R}_{(x_h, b_{1:\tau_K}), x_h}^{T,\text{ext}}.$$

For each $h \in [H]$ and $(x_h, b_{1:h'}) \in \Omega_i^{(II),K}$, we have

$$\widehat{R}_{(x_h, b_{1,h'})}^{T,\text{ext}} = \max_a \sum_{t=1}^{T} \prod_{k=1}^{h'} \pi_i^t(b_k|x_k)\left(\left\langle \pi_{i,h}^t(\cdot|x_{i,h}), L_{i,h}^t(x_{i,h}, \cdot)\right\rangle - L_{i,h}^t(x_{i,h}, a)\right)$$

$$\leq \max_a \sum_{t=1}^{T} \prod_{k=1}^{h'} \pi_i^t(b_k|x_k)\left(\left\langle \pi_{i,h}^t(\cdot|x_{i,h}), \widetilde{L}_{(x_h, b_{1:h'})}^t\right\rangle - \widetilde{L}_{(x_h, b_{1:h'})}^t(a)\right)$$

$$+ \sum_{t=1}^{T} \prod_{k=1}^{h'} \pi_i^t(b_k|x_k)\left\langle \pi_{i,h}^t(\cdot|x_{i,h}), L_{i,h}^t(x_{i,h}, \cdot) - \widetilde{L}_{(x_h, b_{1:h'})}^t\right\rangle$$

$$+ \max_a \sum_{t=1}^{T} \prod_{k=1}^{h'} \pi_i^t(b_k|x_k)\left(\widetilde{L}_{(x_h, b_{1:h'})}^t(a) - L_{i,h}^t(x_{i,h}, a)\right).$$

Substituting this into $\max_{\phi \in \Phi_i^K} \sum_{x_h \in \mathcal{X}_{i,h}} G_h^{T,\text{ext}}(x_h; \phi)$ yields that

$$\max_{\phi \in \Phi_i^K} \sum_{x_h \in \mathcal{X}_{i,h}} G_h^{T,\text{ext}}(x_h; \phi) = \max_{\phi \in \Phi_i^K} \sum_{x_h \in \mathcal{X}_{i,h}} \sum_{b_{1:h-1}} \delta^K(a_{1:h-1}, b_{1:h-1}) \prod_{k=1}^{h-1} \phi_k(a_k|x_k, b_{1:k \wedge \tau_K}) \widehat{R}_{(x_h, b_{1:\tau_K}), x_h}^{T,\text{ext}}$$

$$\leq \widetilde{\text{REGRET}}_h^{T,\text{ext}} + \text{BIAS}_{1,h}^{T,\text{ext}} + \text{BIAS}_{2,h}^{T,\text{ext}},$$

where

$$\widetilde{\text{REGRET}}_h^{T,\text{ext}} := \max_{\phi \in \Phi_i^K} \sum_{x_h \in \mathcal{X}_{i,h}} \sum_{b_{1:h-1}} \delta^K(a_{1:h-1}, b_{1:h-1}) \prod_{k=1}^{h-1} \phi_k(a_k|x_k, b_{1:k \wedge \tau_K})$$

$$\times \max_a \sum_{t=1}^{T} \prod_{k=1}^{\tau_K} \pi_i^t(b_k|x_k)\left(\left\langle \pi_{i,h}^t(\cdot|x_{i,h}), \widetilde{L}_{(x_h, b_{1:\tau_K})}^t\right\rangle - \widetilde{L}_{(x_h, b_{1:\tau_K})}^t(a)\right),$$

$$\text{BIAS}_{1,h}^{T,\text{ext}} := \max_{\phi \in \Phi_i^K} \sum_{x_h \in \mathcal{X}_{i,h}} \sum_{b_{1:h-1}} \delta^K(a_{1:h-1}, b_{1:h-1}) \prod_{k=1}^{h-1} \phi_k(a_k|x_k, b_{1:k \wedge \tau_K})$$

$$\times \sum_{t=1}^{T} \prod_{k=1}^{\tau_K} \pi_i^t(b_k|x_k)\left\langle \pi_{i,h}^t(\cdot|x_{i,h}), L_{i,h}^t(x_{i,h}, \cdot) - \widetilde{L}_{(x_h, b_{1:\tau_K})}^t\right\rangle,$$

$$\text{BIAS}_{2,h}^{T,\text{ext}} := \max_{\phi \in \Phi_i^K} \sum_{x_h \in \mathcal{X}_{i,h}} \sum_{b_{1:h-1}} \delta^K(a_{1:h-1}, b_{1:h-1}) \prod_{k=1}^{h-1} \phi_k(a_k|x_k, b_{1:k \wedge \tau_K})$$

$$\times \max_a \sum_{t=1}^{T} \prod_{k=1}^{\tau_K} \pi_i^t(b_k|x_k) \left( \widetilde{L}^t_{(x_h, b_{1:\tau_K})}(a) - L^t_{i,h}(x_{i,h}, a) \right).$$

The bounds for $\sum_{h=1}^{H} \widetilde{\mathrm{REGRET}}_h^{T,\mathrm{ext}}$, $\sum_{h=1}^{H} \mathrm{BIAS}_{1,h}^{T,\mathrm{ext}}$, and $\sum_{h=1}^{H} \mathrm{BIAS}_{2,h}^{T,\mathrm{ext}}$ are given in the following three Lemmas (proofs deferred to Appendix G.5.1, G.5.2, and G.5.3) respectively.

**Lemma G.7** (Bound on $\widetilde{\mathrm{REGRET}}_h^{T,\mathrm{ext}}$). *If we choose learning rates as*

$$\eta_{x_h} = \sqrt{\binom{H}{K \wedge H} X_i A_i^{K \wedge H+1} \log(8X_i A_i/p)/(H^3 T)}$$

*for all $x_h \in \mathcal{X}_i$ (same with (10)). Then with probability at least $1 - p/4$, we have*

$$\sum_{h=1}^{H} \widetilde{\mathrm{REGRET}}_h^{T,\mathrm{ext}} \leq \sqrt{H^3 \binom{H}{K \wedge H} A_i^{K \wedge H+1} X_i T \iota}$$

$$+ \mathcal{O}\left( \binom{H}{K \wedge H} A_i^{K \wedge H+1} X_i \iota \sqrt{\frac{H \binom{H}{K \wedge H} A_i^{K \wedge H+1} X_i \iota}{T}} \right),$$

*where $\iota = \log(8X_i A_i/p)$ is a log factor.*

**Lemma G.8** (Bound on $\mathrm{BIAS}_{1,h}^{T,\mathrm{ext}}$). *With probability at least $1 - p/8$, we have*

$$\sum_{h=1}^{H} \mathrm{BIAS}_{1,h}^{T,\mathrm{ext}} \leq \mathcal{O}\left( \sqrt{H^3 \binom{H}{K \wedge H} A_i^{K \wedge H+1} X_i T \iota} + H \binom{H}{K \wedge H} A_i^{K \wedge H+1} X_i \iota \right),$$

*where $\iota = \log(8X_i A_i/p)$ is a log factor.*

**Lemma G.9** (Bound on $\mathrm{BIAS}_{2,h}^{T,\mathrm{ext}}$). *With probability at least $1 - p/8$, we have*

$$\sum_{h=1}^{H} \mathrm{BIAS}_{2,h}^{T,\mathrm{ext}} \leq \mathcal{O}\left( \sqrt{H^3 \binom{H}{K \wedge H} A_i^{K \wedge H} X_i T \iota} + H \binom{H}{K \wedge H} A_i^{K \wedge H} X_i \iota \right),$$

*where $\iota = \log(8X_i A_i/p)$ is a log factor.*

Combining Lemma G.7, G.8, and G.9, we have with probability at least $1 - p/2$ that

$$\sum_{h=1}^{H} \max_{\phi \in \Phi_i^K} \sum_{x_h \in \mathcal{X}_{i,h}} G_h^{T,\mathrm{ext}}(x_h; \phi) \leq \sum_{h=1}^{H} \widetilde{\mathrm{REGRET}}_h^{T,\mathrm{ext}} + \sum_{h=1}^{H} \mathrm{BIAS}_{1,h}^{T,\mathrm{ext}} + \sum_{h=1}^{H} \mathrm{BIAS}_{2,h}^{T,\mathrm{ext}}$$

$$\leq \sqrt{H^3 \binom{H}{K \wedge H} A_i^{K \wedge H+1} X_i T \iota}$$

$$+ \mathcal{O}\left( \binom{H}{K \wedge H} A_i^{K \wedge H+1} X_i \iota \sqrt{\frac{H \binom{H}{K \wedge H} A_i^{K \wedge H+1} X_i \iota}{T}} \right)$$

$$+ \mathcal{O}\left( H \binom{H}{K \wedge H} A_i^{K \wedge H+1} X_i \iota \right).$$

$\square$

### G.5.1  Proof of Lemma G.7: Bound on $\widetilde{\mathrm{REGRET}}_h^{T,\mathrm{ext}}$

*Proof.* Recall that $\widetilde{\mathrm{REGRET}}_h^{T,\mathrm{ext}}$ is defined as

$$\max_{\phi \in \Phi_i^K} \sum_{x_h \in \mathcal{X}_{i,h}} \sum_{b_{1:h-1}} \delta^K(a_{1:h-1}, b_{1:h-1}) \prod_{k=1}^{h-1} \phi_k(a_k|x_k, b_{1:k \wedge \tau_K})$$

$$\times \max_a \sum_{t=1}^{T} \prod_{k=1}^{\tau_K} \pi_i^t(b_k|x_k)\left(\left\langle \pi_{i,h}^t(\cdot|x_{i,h}), \widetilde{L}_{(x_h,b_{1:\tau_K})}^t\right\rangle - \widetilde{L}_{(x_h,b_{1:\tau_K})}^t(a)\right).$$

We first apply regret minimization lemma (Lemma A.3) on $\mathcal{R}_{x_h}$ to give an upper bound of

$$\max_a \sum_{t=1}^{T} \prod_{k=1}^{\tau_K} \pi_i^t(b_k|x_k)\left(\left\langle \pi_{i,h}^t(\cdot|x_{i,h}), \widetilde{L}_{(x_h,b_{1:\tau_K})}^t\right\rangle - \widetilde{L}_{(x_h,b_{1:\tau_K})}^t(a)\right).$$

For $\mathcal{R}_{x_h}$, Algorithm 7 gives that $M_{b_{1:h'}}^t = \prod_{k=1}^{h'} \pi_i^t(b_k|x_k)$ and $w_{b_{1:h'}} = \prod_{k\in W\cup\{h'+1,\ldots,h\}} \pi_{i,k}^{\star,h}(a_k|x_k)$ for any $b_{1:h'} \in \Omega_i^{(\mathrm{II}),K}(x_h)$, where $W = \{k \in [h-1] : b_k \neq a_k\}$. Letting $W(h) = W \cup \{h'+1,\ldots,h\}$, $\eta$ be the learning rate of $\mathcal{R}_{x_h}$ and $\overline{L} = H$. The assumptions in Lemma A.3 are verified in Section G.4.1. So by Lemma A.3, with probability at least $1 - p/8$, we have for all $x_h \in cX_i$,

$$\max_a \sum_{t=1}^{T} \prod_{k=1}^{\tau_K} \pi_i^t(b_k|x_k)\left(\left\langle \pi_{i,h}^t(\cdot|x_{i,h}), \widetilde{L}_{(x_h,b_{1:\tau_K})}^t\right\rangle - \widetilde{L}_{(x_h,b_{1:\tau_K})}^t(a)\right)$$

$$\leq \frac{2\log(8X_iA_i/p)}{\eta w_{b_{1:h'}}} + \eta \sum_{t=1}^{T} M_{b_{1:h'}}^t \overline{L}\left\langle \pi_i^t(\cdot|x_k), \widetilde{L}_{(x_h,b_{1:'})}^t(\cdot)\right\rangle$$

$$= \frac{2\log(8X_iA_i/p)}{\eta \prod_{k\in W(h)} \pi_{i,k}^{\star,h}(a_k|x_k)} + H\eta \prod_{k=1}^{h'} \pi_i^t(b_k|x_k)$$

$$\times \sum_{t=1}^{T}\left\langle \pi_{i,h}^t(\cdot|x_{i,h}), \frac{\mathbf{1}\left\{(x_h,\cdot) = (x_h^{t,(h,h',W)}, a_h^{t,(h,h',W)})\right\} \sum_{h''=h}^{H}\left(1 - r_{i,h''}^{t,(h,h',W)}\right)}{\prod_{k\in W(h)} \pi_{i,k}^{\star,h}(a_k|x_k) \prod_{k\in[h']\backslash W} \pi_{i,k}^t(b_k|x_k)}\right\rangle$$

$$\leq \frac{2\log(8X_iA_i/p)}{\eta \prod_{k\in W(h)} \pi_{i,k}^{\star,h}(a_k|x_k)} + H^2\eta \sum_{t=1}^{T}\left\langle \pi_i^t(\cdot|x_k), \frac{\prod_{k\in W} \pi_{i,k}^t(b_k|x_k)\mathbf{1}\left\{(x_h,a) = (x_h^{t,(h,h',W)}, a_h^{t,(h,h',W)})\right\}}{\prod_{k\in W(h)} \pi_{i,k}^{\star,h}(a_k|x_k)}\right\rangle.$$

Here, we use $(i)$ our choices of $M_{b_{1:h'}}^t$ and $w_{b_{1:h'}}$; $(ii)$ $(|\mathcal{B}^s| + |\mathcal{B}^e|)|\Psi^e| \leq A_i^{H+1} \leq X_iA_i$ and $(iii)$ taking union bound over all infosets. Plugging this into $\widetilde{\mathrm{REGRET}}_h^{T,\mathrm{ext}}$, we have

$$\widetilde{\mathrm{REGRET}}_h^{T,\mathrm{ext}}$$

$$= \max_{\phi\in\Phi_i^K} \sum_{x_h\in\mathcal{X}_{i,h}} \sum_{b_{1:h-1}} \delta^K(a_{1:h-1}, b_{1:h-1}) \prod_{k=1}^{h-1} \phi_k(a_k|x_k, b_{1:k\wedge h'})$$

$$\times \max_a \sum_{t=1}^{T} \prod_{k=1}^{\tau_K} \pi_i^t(b_k|x_k)\left(\left\langle \pi_{i,h}^t(\cdot|x_{i,h}), \widetilde{L}_{(x_h,b_{1:\tau_K})}^t\right\rangle - \widetilde{L}_{(x_h,b_{1:\tau_K})}^t(a)\right)$$

$$\leq \frac{2\log(8X_iA_i/p)}{\eta} \max_{\phi\in\Phi_i^K} \sum_{x_h\in\mathcal{X}_{i,h}} \sum_{b_{1:h-1}} \delta^K(a_{1:h-1}, b_{1:h-1}) \frac{\prod_{k=1}^{h-1} \phi_k(a_k|x_k, b_{1:k\wedge h'})}{\prod_{k\in W(h)} \pi_{i,k}^{\star,h}(a_k|x_k)}$$

$$+ H^2\eta \max_{\phi\in\Phi_i^K} \sum_{x_h\in\mathcal{X}_{i,h}} \sum_{b_{1:h-1}} \delta^K(a_{1:h-1}, b_{1:h-1}) \frac{\prod_{k=1}^{h-1} \phi_k(a_k|x_k, b_{1:k\wedge h'})}{\prod_{k\in W(h)} \pi_{i,k}^{\star,h}(a_k|x_k)}$$

$$\times \underbrace{\sum_{t=1}^{T} \prod_{k\in W} \pi_{i,k}^t(b_k|x_k)\left\langle \pi_{i,h}^t(\cdot|x_{i,h}), \mathbf{1}\left\{(x_h,\cdot) = (x_h^{t,(h,h',W)}, a_h^{t,(h,h',W)})\right\}\right\rangle}_{:=\overline{\Delta}_t^{(x_h,b_{1:h'})}}.$$

Letting

$$\mathrm{I}_h := \frac{2\log(8X_iA_i/p)}{\eta} \max_{\phi\in\Phi_i^K} \sum_{x_h\in\mathcal{X}_{i,h}} \sum_{b_{1:h-1}} \delta^K(a_{1:h-1}, b_{1:h-1}) \frac{\prod_{k=1}^{h-1} \phi_k(a_k|x_k, b_{1:k\wedge h'})}{\prod_{k\in W(h)} \pi_{i,k}^{\star,h}(a_k|x_k)};$$

$$\mathrm{II}_h := H^2\eta \max_{\phi\in\Phi_i^K} \sum_{x_h\in\mathcal{X}_{i,h}} \sum_{b_{1:h-1}} \delta^K(a_{1:h-1}, b_{1:h-1}) \frac{\prod_{k=1}^{h-1} \phi_k(a_k|x_k, b_{1:k\wedge h'})}{\prod_{k\in W(h)} \pi_{i,k}^{\star,h}(a_k|x_k)} \sum_{t=1}^{T} \overline{\Delta}_t^{(x_h,b_{1:h'})}.$$

Using Lemma B.4, we have

$$\sum_{x_h \in \mathcal{X}_{i,h}} \sum_{b_{1:h-1}} \delta^K(a_{1:h-1}, b_{1:h-1}) \frac{\prod_{k=1}^{h-1} \phi_k(a_k|x_k, b_{1:k \wedge h'})}{\prod_{k \in W(h)} \pi_{i,k}^{\star,h}(a_k|x_k)}$$

$$= \sum_{x_h \in \mathcal{X}_{i,h}} \sum_{b_{1:h-1}} \delta^K(a_{1:h-1}, b_{1:h-1}) \frac{\prod_{k=1}^{h-1} \phi_k(a_k|x_k, b_{1:k \wedge h'}) \prod_{k \in [h-1] \setminus W} \pi_{i,k}^{\star,h}(a_k|x_k)}{\prod_{k \in [h]} \pi_{i,k}^{\star,h}(a_k|x_k)}$$

$$\overset{(i)}{=} \sum_{x_h \in \mathcal{X}_{i,h}} \sum_{a_h} \frac{\sum_{b_{1:h-1}} \delta^K(a_{1:h-1}, b_{1:h-1}) \prod_{k=1}^{h-1} \phi_k(a_k|x_k, b_{1:k \wedge h'}) \prod_{k \in [h-1] \setminus W} \pi_{i,k}^{\star,h}(b_k|x_k) \cdot \pi_{i,h}^{\star,h}(a_h|x_h)}{\prod_{k \in [h]} \pi_{i,k}^{\star,h}(a_k|x_k)}$$

$$\overset{(ii)}{=} \sum_{W \subset [h-1], |W| = K} A_i^{|W|} \sum_{x_h, a_h} \left( \prod_{k \in [h]} \pi_{i,k}^{\star,h}(a_k|x_k) \right)^{-1}$$

$$\times \sum_{b_{1:h-1}: \{k \in [h-1]: a_k \neq b_k\} = W} \prod_{k=1}^{h-1} \phi_k(a_k|x_k, b_{1:k \wedge h'}) \prod_{k \in [h-1] \setminus W} \pi_{i,k}^{\star,h}(b_k|x_k) \prod_{k \in W} \pi_{i,k}^{\mathrm{unif}}(b_k|x_k) \pi_{i,h}^{\star,h}(a_h|x_h)$$

$$\overset{(iii)}{\le} \sum_{W \subset [h-1], |W| = K} A_i^{|W|} X_{i,h} A_i$$

$$= X_{i,h} \binom{h-1}{K} A_i^{K \wedge H + 1}.$$

Here, (i) uses that $\pi_{i,h}^{\star,h}(\cdot|x_h)$ is uniform distribution on $\mathcal{A}_i$ and that for $k \in [h-1] \setminus W$, we have $a_k = b_k$; (ii) follows from grouping $x_h$ and $b_{1:h}$ by $|W|$, where $\pi_{i,k}^{\mathrm{unif}}$ is the uniform distribution on $\mathcal{A}_i$; (iii) uses Lemma B.4 and the fact that for any fixed $W$, (by relaxing the summation over $b_{1:h-1}$ to the full sum $\sum_{b_{1:h-1}}$) the numerator is no more than the sequence-form of the following policy:

- Sample recommended action $b_k$ from $\pi_{i,k}^{\star,h}(\cdot|x_k)$ if step $k \in W$. Otherwise, sample recommended action $b_k$ from $\pi_{i,k}^{\mathrm{unif}}(\cdot|x_k)$.

- "True" actions are sampled from $\phi_k(a_k|x_k, b_{1:k})$ for $k \in [h-1]$. At step $h$, "True" action is sampled from $a_h$.

Consequently,

$$\sum_{x_h \in \mathcal{X}_{i,h}} \sum_{b_{1:h-1}} \delta^K(a_{1:h-1}, b_{1:h-1}) \frac{\prod_{k=1}^{h-1} \phi_k(a_k|x_k, b_{1:k \wedge h'})}{\prod_{k \in W(h)} \pi_{i,k}^{\star,h}(a_k|x_k)} \le X_{i,h} \binom{h-1}{K} A_i^{K \wedge H + 1}. \tag{36}$$

So we have

$$\mathrm{I}_h \le \frac{2 \log(8 X_i A_i / p)}{\eta} X_{i,h} \binom{h-1}{K} A_i^{K \wedge H + 1}.$$

To give an upper bound of $\mathrm{I}_h$, obvserve that the random variables $\overline{\Delta}_t^{(x_h, b_{1:h'})}$ satisfy the following:

- $\overline{\Delta}_t^{(x_h, b_{1:h'})} = \prod_{k \in W} \pi_{i,k}^t(b_k|x_k) \left\langle \pi_{i,h}^t(\cdot|x_{i,h}), \mathbf{1}\left\{ (x_h, \cdot) = (x_h^{t,(h,h',W)}, a_h^{t,(h,h',W)}) \right\} \right\rangle \in [0, 1]$.

- Let $\mathcal{F}_{t-1}$ be the $\sigma$-algebra containing all information until $\pi^t$ is sampled, then

$$\mathbb{E}\left[ \overline{\Delta}_t^{(x_h, b_{1:h'})} \middle| \mathcal{F}_{t-1} \right]$$

$$= \prod_{k \in W} \pi_{i,k}^t(b_k|x_k) \mathbb{E}\left[ \sum_a \pi_{i,h}^t(a|x_{i,h}) \mathbf{1}\left\{ (x_h, a) = (x_h^{t,(h,h',W)}, a_h^{t,(h,h',W)}) \right\} \middle| \mathcal{F}_{t-1} \right]$$

$$= \prod_{k \in W} \pi_{i,k}^t(b_k|x_k) \sum_a \pi_{i,h}^t(a|x_{i,h}) \mathbb{P}^{((\pi_{i,k}^{\star,h})_{k \in W(h)} (\pi_{i,k}^t)_{k \in [h'] \setminus W}) \times \pi_{-i}^t}(x_h^{t,(h,h',W)} = x_h, a_h^{t,(h,h',W)} = a)$$

$$= \prod_{k \in W} \pi_{i,k}^t(b_k|x_k) \sum_a \pi_{i,h}^t(a|x_{i,h}) \left( \prod_{k \in W(h)} \pi_{i,k}^{\star,h}(a_k|x_k) \cdot \prod_{k \in [h'] \setminus W} \pi_{i,k}^t(a_k|x_k) \cdot \pi_{i,h}^{\star,h}(a|x_h) p_{1:h}^{\pi_{-i}^t}(x_h) \right)$$

$$= \prod_{k \in [h']} \pi_{i,k}^t(b_k|x_k) \prod_{k \in W(h)} \pi_{i,k}^{\star,h}(a_k|x_k) p_{1:h}^{\pi_{-i}^t}(x_h),$$

where the last equation is because $\prod_{k \in W} \pi_{i,k}^t(b_k|x_k) \cdot \prod_{k \in [h'] \setminus W} \pi_{i,k}^t(a_k|x_k) = \prod_{k \in [h']} \pi_{i,k}^t(b_k|x_k)$;

- The conditional variance $\mathbb{E}[(\overline{\Delta}_t^{(x_h, b_{1:h'})})^2 | \mathcal{F}_{t-1}]$ can be bounded as

$$\mathbb{E}\left[ \left( \overline{\Delta}_t^{(x_h, b_{1:h'})} \right)^2 \middle| \mathcal{F}_{t-1} \right] \leq \mathbb{E}\left[ \overline{\Delta}_t^{(x_h, b_{1:h'})} \middle| \mathcal{F}_{t-1} \right]$$

$$= \prod_{k \in [h']} \pi_{i,k}^t(b_k|x_k) \prod_{k \in W(h)} \pi_{i,k}^{\star,h}(a_k|x_k) p_{1:h}^{\pi_{-i}^t}(x_h).$$

Here, the inequality comes from that $\overline{\Delta}_t^{(x_h, b_{1:h'})} \in [0, 1]$.

Therefore, we can apply Freedman's inequality and union bound to get that for any fixed $\lambda \in (0, 1]$, with probability at least $1 - p/8$, the following holds simultaneously for all $(h, x_{i,h}, b_{1:h-1})$:

$$\sum_{t=1}^T \overline{\Delta}_t^{(x_h, b_{1:h'})} \leq (\lambda + 1) \sum_{t=1}^T \prod_{k \in [h']} \pi_{i,k}^t(b_k|x_k) \prod_{k \in W(h)} \pi_{i,k}^{\star,h}(a_k|x_k) p_{1:h}^{\pi_{-i}^t}(x_h) + \frac{C \log(X_i A_i/p)}{\lambda},$$

where $C > 0$ is some absolute constant. Plugging this bound into $\text{II}_h$ yields that,

$$\text{II}_h \leq H^2 \eta \max_{\phi \in \Phi_i^K} \sum_{x_h \in \mathcal{X}_{i,h}} \sum_{b_{1:h-1}} \delta^K(a_{1:h-1}, b_{1:h-1}) \frac{\prod_{k=1}^{h-1} \phi_k(a_k|x_k, b_{1:k \wedge h'})}{\prod_{k \in W(h)} \pi_{i,k}^{\star,h}(a_k|x_k)}$$

$$\times \left[ (\lambda + 1) \sum_{t=1}^T \prod_{k \in [h']} \pi_{i,k}^t(b_k|x_k) \prod_{k \in W(h)} \pi_{i,k}^{\star,h}(a_k|x_k) p_{1:h}^{\pi_{-i}^t}(x_h) + \frac{C \log(X_i A_i/p)}{\lambda} \right].$$

Note that for fixed $t$ and any $\phi \in \Phi_i^K$, by Corollary D.2,

$$\sum_{x_h \in \mathcal{X}_{i,h}} \sum_{b_{1:h-1}} \delta^K(a_{1:h-1}, b_{1:h-1}) \prod_{k=1}^{h-1} \phi_k(a_k|x_k, b_{1:k \wedge h'}) \prod_{k \in [h']} \pi_{i,k}^t(b_k|x_k) p_{1:h}^{\pi_{-i}^t}(x_{i,h}) \leq 1.$$

So we have

$$\max_{\phi \in \Phi_i^K} \sum_{x_h \in \mathcal{X}_{i,h}} \sum_{b_{1:h-1}} \delta^K(a_{1:h-1}, b_{1:h-1}) \prod_{k=1}^{h-1} \phi_k(a_k|x_k, b_{1:k \wedge h'}) \sum_{t=1}^T \prod_{k \in [h']} \pi_{i,k}^t(b_k|x_k) p_{1:h}^{\pi_{-i}^t}(x_{i,h}) \leq T.$$

Moreover, by the previous bound (36), for any $\phi \in \Phi_i^K$,

$$\sum_{x_h \in \mathcal{X}_{i,h}} \sum_{b_{1:h-1}} \delta^K(a_{1:h-1}, b_{1:h-1}) \frac{\prod_{k=1}^{h-1} \phi_k(a_k|x_k, b_{1:k \wedge h'})}{\prod_{k \in W(h)} \pi_{i,k}^{\star,h}(a_k|x_k)} \leq X_{i,h} \binom{h-1}{K} A_i^{K \wedge H + 1}.$$

Using these two inequalities, we can get that

$$\text{II}_h \leq H^2 \eta (\lambda + 1) T + \frac{C H^2 \eta \log(X_i A_i/p)}{\lambda} X_{i,h} \binom{h-1}{K} A_i^{K \wedge H + 1}.$$

Taking summation over $h \in [H]$, we have

$$\sum_{h=1}^H \widetilde{\text{REGRET}}_h^{T,\text{ext}} = \sum_{h=1}^H (\text{I}_h + \text{II}_h)$$

$$\leq H^3 \eta (\lambda + 1) T + \left( \frac{2 \log(8 X_i A_i/p)}{\eta} + \frac{C H^2 \eta \log(X_i A_i/p)}{\lambda} \right) \sum_{h=1}^H X_{i,h} \binom{h-1}{K} A_i^{K \wedge H + 1}$$

$$\leq H^3\eta(\lambda+1)T + \left(\frac{2\log(8X_iA_i/p)}{\eta} + \frac{CH^2\eta\log(X_iA_i/p)}{\lambda}\right)X_i\binom{H-1}{K\wedge H}A_i^{K\wedge H+1}$$

$$\leq H^3\eta(\lambda+1)T + \left(\frac{2\log(8X_iA_i/p)}{\eta} + \frac{CH^2\eta\log(X_iA_i/p)}{\lambda}\right)X_i\binom{H}{K\wedge H}A_i^{K\wedge H+1},$$

for all $\lambda \in (0,1]$. Choosing $\lambda = 1$, we have,

$$\sum_{h=1}^{H}\widetilde{\mathrm{REGRET}}_h^{T,\mathrm{ext}} \leq 2H^3\eta T + \left(\frac{2\log(8X_iA_i/p)}{\eta} + CH^2\eta\log(X_iA_i/p)\right)X_i\binom{H-1}{K\wedge H}A_i^{K\wedge H+1}.$$

Then, choosing $\eta = \sqrt{\binom{H}{K\wedge H}A_i^{K\wedge H+1}X_i\iota/(H^3T)}$, we have

$$\sum_{h=1}^{H}\widetilde{\mathrm{REGRET}}_h^{T,\mathrm{ext}} \leq \sqrt{H^3\binom{H}{K\wedge H}A_i^{K\wedge H+1}X_iT\iota}$$

$$+\mathcal{O}\left(\binom{H}{K\wedge H}A_i^{K\wedge H+1}X_i\iota\sqrt{\frac{H\binom{H}{K\wedge H}A_i^{K\wedge H+1}X_i\iota}{T}}\right),$$

where $\iota = \log(8X_iA_i/p)$ is a log factor. $\qquad\square$

### G.5.2 Proof of Lemma G.8: Bound on $\mathrm{BIAS}_{1,h}^{T,\mathrm{ext}}$

*Proof.* For fixed $x_h$ and $b_{1:h-1}$, let $W = \{k \in [h-1] : b_k \neq a_k\}$ and $h'$ is the maximal element in $W$, so we have $h' = \tau_K$. We define $W(h)$ as the set $W \cup \{h'+1, \ldots, h\}$. We can rewrite $\mathrm{BIAS}_{1,h}^{T,\mathrm{ext}}$ as

$$\mathrm{BIAS}_{1,h}^{T,\mathrm{ext}}$$
$$= \max_{\phi \in \Phi_i^K}\sum_{x_h \in \mathcal{X}_{i,h}}\sum_{b_{1:h-1}}\delta^K(a_{1:h-1}, b_{1:h-1})\frac{\prod_{k=1}^{h-1}\phi_k(a_k|x_k, b_{1:k\wedge h'})}{\prod_{k \in W(h)}\pi_{i,k}^{\star,h}(a_k|x_k)}$$

$$\times \sum_{t=1}^{T}\underbrace{\prod_{k \in W(h)}\pi_{i,k}^{\star,h}(a_k|x_k)\prod_{k=1}^{h'}\pi_{i,k}^t(b_k|x_k)\left\langle\pi_{i,h}^t(\cdot|x_{i,h}), L_{i,h}^t(x_{i,h},\cdot) - \widetilde{L}_{(x_h,b_{1:h'})}^t\right\rangle}_{:=\widetilde{\Delta}_t^{(x_h,b_{1:h'})}}.$$

Observe that the random variable $\widetilde{\Delta}_t^{(x_h,b_{1:h'})}$ satisfy the following:

- By the definition of $\widetilde{L}$, we can rewrite $\widetilde{\Delta}_t^{(x_h,b_{1:h'})}$ as

$$\widetilde{\Delta}_t^{(x_h,b_{1:h'})} = \prod_{k=1}^{h'}\pi_{i,k}^t(b_k|x_k)\left\langle\pi_{i,h}^t(\cdot|x_h), L_{(x_h,b_{1:h'})}^t\right\rangle\prod_{k \in W(h)}\pi_{i,k}^{\star,h}(a_k|x_k)$$

$$- \prod_{k \in W}\pi_{i,k}^t(b_k|x_k)\left\langle\pi_{i,h}^t(\cdot|x_h), \mathbf{1}\left\{(x_h,\cdot) = (x_h^{t,(h,h',W)}, a_h^{t,(h,h',W)})\right\}\cdot\left(H-h+1-\sum_{h''=h}^{H}r_{i,h''}^{t,(h,h',W)}\right)\right\rangle;$$

- $\widetilde{\Delta}_t^{(x_h,b_{1:h'})} \leq \left\langle\pi_i^t(\cdot|x_h), L_{(x_h,b_{1:h'})}^t\right\rangle \leq H$ ;

- $\mathbb{E}[\widetilde{\Delta}_t^{(x_h,b_{1:h'})}|\mathcal{F}_{t-1}] = 0$, where $\mathcal{F}_{t-1}$ is the $\sigma$-algebra containing all information until $\pi^t$ is sampled. This also can be seen from the unbiasedness of $\widetilde{L}$;

- The conditional variance $\mathbb{E}[(\widetilde{\Delta}_t^{(x_h,b_{1:h'})})^2|\mathcal{F}_{t-1}]$ can be bounded as

$$\mathbb{E}\left[\left(\widetilde{\Delta}_t^{(x_h,b_{1:h'})}\right)^2\bigg|\mathcal{F}_{t-1}\right]$$

$$\leq \mathbb{E}\left[\left(H - h + 1 - \sum_{h''=h}^{H} r_{i,h''}^{t,(h,h',W)}\right)^2 \left(\left\langle \pi_{i,h}^t(\cdot|x_h), \prod_{k\in W} \pi_{i,k}^t(b_k|x_k)\mathbf{1}\left\{(x_h,\cdot) = (x_h^{t,(h,h',W)}, a_h^{t,(h,h',W)})\right\}\right\rangle\right)^2 \Big| \mathcal{F}_{t-1}\right]$$

$$\overset{(i)}{\leq} H^2 \prod_{k\in W} \pi_{i,k}^t(b_k|x_k)\mathbb{E}\left[\left\langle \pi_{i,h}^t(\cdot|x_h), \mathbf{1}\left\{(x_h,\cdot) = (x_h^{t,(h,h',W)}, a_h^{t,(h,h',W)})\right\}\right\rangle \Big| \mathcal{F}_{t-1}\right]$$

$$= H^2 \prod_{k\in W} \pi_{i,k}^t(b_k|x_k)\mathbb{E}\left[\sum_{a\in\mathcal{A}_i} \pi_{i,h}^t(a|x_h), \mathbf{1}\left\{(x_h,a) = (x_h^{t,(h,h',W)}, a_h^{t,(h,h',W)})\right\}\right\rangle \Big| \mathcal{F}_{t-1}\right]$$

$$= H^2 \prod_{k\in W} \pi_{i,k}^t(b_k|x_k) \cdot \sum_{a\in\mathcal{A}_i} \pi_{i,h}^t(a|x_h)\mathbb{P}^{((\pi_{i,k}^{\star,h})_{k\in W(h)}(\pi_{i,k}^t)_{k\in[h']\setminus W})\times\pi_{-i}^t}\left((x_h^{t,(h,h',W)}, a_h^{t,(h,h',W)}) = (x_h,a)\right)$$

$$= H^2 \prod_{k\in W} \pi_{i,k}^t(b_k|x_k) \cdot \sum_{a\in\mathcal{A}_i} \pi_{i,h}^t(a|x_h) \prod_{k\in W(h)} \pi_{i,k}^{\star,h}(a_k|x_k) \cdot \prod_{k\in[h']\setminus W} \pi_{i,k}^t(a_k|x_k) \cdot p_{1:h}^{\pi_{-i}^t}(x_{i,h})$$

$$\overset{(ii)}{\leq} H^2 \prod_{k\in[h']} \pi_{i,k}^t(b_k|x_k) \cdot \prod_{k\in W(h)} \pi_{i,k}^{\star,h}(a_k|x_k) \cdot p_{1:h}^{\pi_{-i}^t}(x_{i,h}).$$

Here, (i) uses $\left\langle \pi_{i,h}^t(\cdot|x_h), \mathbf{1}\left\{(x_h,\cdot) = (x_h^{t,(h,h',W)}, a_h^{t,(h,h',W)})\right\}\right\rangle \in [0,1]$; (ii) is because

$$\prod_{k\in W} \pi_{i,k}^t(b_k|x_k) \prod_{k\in[h']\setminus W} \pi_{i,k}^t(a_k|x_k) = \prod_{k\in[h']} \pi_{i,k}^t(b_k|x_k).$$

Therefore, we can apply Freedman's inequality and union bound to get that for any fixed $\lambda \in (0, 1/H]$, with probability at least $1 - p/8$, the following holds simultaneously for all $(h, x_{i,h}, b_{1:h-1})$:

$$\sum_{t=1}^{T} \widetilde{\Delta}_t^{(x_h, b_{1:h'})} \leq \lambda H^2 \prod_{k\in W(h)} \pi_{i,k}^{\star,h}(a_k|x_k) \sum_{t=1}^{T} \prod_{k\in[h']} \pi_{i,k}^t(b_k|x_k)p_{1:h}^{\pi_{-i}^t}(x_{i,h}) + \frac{C\log(X_i A_i/p)}{\lambda},$$

where $C > 0$ is some absolute constant. Plugging this bound into $\text{BIAS}_{1,h}^{T,\text{ext}}$ yields that, for all $h \in [H]$,

$$\text{BIAS}_{1,h}^{T,\text{ext}} \leq \max_{\phi\in\Phi_i^K} \sum_{x_h\in\mathcal{X}_{i,h}} \sum_{b_{1:h-1}} \delta^K(a_{1:h-1}, b_{1:h-1})\frac{\prod_{k=1}^{h-1} \phi_k(a_k|x_k, b_{1:k\wedge h'})}{\prod_{k\in W(h)} \pi_{i,k}^{\star,h}(a_k|x_k)}$$

$$\times \left[\lambda H^2 \prod_{k\in W(h)} \pi_{i,k}^{\star,h}(a_k|x_k) \sum_{t=1}^{T} \prod_{k\in[h']} \pi_{i,k}^t(b_k|x_k)p_{1:h}^{\pi_{-i}^t}(x_{i,h}) + \frac{C\log(X_i A_i/p)}{\lambda}\right]$$

$$\leq \lambda H^2 \max_{\phi\in\Phi_i^K} \sum_{x_h\in\mathcal{X}_{i,h}} \sum_{b_{1:h-1}} \delta^K(a_{1:h-1}, b_{1:h-1})\prod_{k=1}^{h-1} \phi_k(a_k|x_k, b_{1:k\wedge h'}) \sum_{t=1}^{T} \prod_{k\in[h']} \pi_{i,k}^t(b_k|x_k)p_{1:h}^{\pi_{-i}^t}(x_{i,h})$$

$$+ \frac{C\log(X_i A_i/p)}{\lambda} \max_{\phi\in\Phi_i^K} \sum_{x_h\in\mathcal{X}_{i,h}} \sum_{b_{1:h-1}} \delta^K(a_{1:h-1}, b_{1:h-1})\frac{\prod_{k=1}^{h-1} \phi_k(a_k|x_k, b_{1:k\wedge h'})}{\prod_{k\in W(h)} \pi_{i,k}^{\star,h}(a_k|x_k)}.$$

Note that for fixed $t$ and any $\phi \in \Phi_i^K$, by Corollary D.2,

$$\sum_{x_h\in\mathcal{X}_{i,h}} \sum_{b_{1:h-1}} \delta^K(a_{1:h-1}, b_{1:h-1})\prod_{k=1}^{h-1} \phi_k(a_k|x_k, b_{1:k\wedge h'}) \prod_{k\in[h']} \pi_{i,k}^t(b_k|x_k)p_{1:h}^{\pi_{-i}^t}(x_{i,h}) \leq 1.$$

So we have

$$\lambda H^2 \max_{\phi\in\Phi_i^K} \sum_{x_h\in\mathcal{X}_{i,h}} \sum_{b_{1:h-1}} \delta^K(a_{1:h-1}, b_{1:h-1})\prod_{k=1}^{h-1} \phi_k(a_k|x_k, b_{1:k\wedge h'}) \sum_{t=1}^{T} \prod_{k\in[h']} \pi_{i,k}^t(b_k|x_k)p_{1:h}^{\pi_{-i}^t}(x_{i,h}) \leq \lambda H^2 T.$$

Moreover, by the inequality (36) in the proof of Lemma G.7, we have for any $\phi \in \Phi_i^K$,

$$\sum_{x_h\in\mathcal{X}_{i,h}} \sum_{b_{1:h-1}} \delta^K(a_{1:h-1}, b_{1:h-1})\frac{\prod_{k=1}^{h-1} \phi_k(a_k|x_k, b_{1:k\wedge h'})}{\prod_{k\in W(h)} \pi_{i,k}^{\star,h}(a_k|x_k)} \leq X_{i,h}\binom{h-1}{K}A_i^{K\wedge H+1}.$$

Plugging these bounds into $\text{BIAS}_{1,h}^{T,\text{ext}}$ yields that, with probability at least $1 - p/8$, for all $h \in [H]$,

$$\text{BIAS}_{1,h}^{T,\text{ext}} \leq \lambda H^2 T + \frac{C \log(X_i A_i/p)}{\lambda} X_{i,h} \binom{h-1}{K} A_i^{K \wedge H + 1}.$$

Taking summation over $h \in [H]$, we have

$$\sum_{h=1}^{H} \text{BIAS}_{1,h}^{T,\text{ext}} \leq \lambda H^3 T + \frac{C \log(X_i A_i/p)}{\lambda} \sum_{h=1}^{H} X_{i,h} \binom{h-1}{K} A_i^{K \wedge H + 1}$$

$$\leq \lambda H^3 T + \frac{C \log(X_i A_i/p)}{\lambda} X_i \binom{H}{K \wedge H} A_i^{K \wedge H + 1},$$

for all $\lambda \in (0, 1/H]$. Choose $\lambda = \min \left\{ \frac{1}{H}, \sqrt{\frac{C X_{i,h} \binom{H}{K \wedge H} A_i^{K \wedge H + 1} \log(X_i A_i/p)}{H^3 T}} \right\}$, we obtain

$$\sum_{t=1}^{T} \text{BIAS}_{1,h}^{T,\text{ext}} \leq \mathcal{O} \left( \sqrt{H^3 \binom{H}{K \wedge H} A_i^{K \wedge H + 1} X_i T \iota} + H \binom{H}{K \wedge H} A_i^{K \wedge H + 1} X_i \iota \right),$$

where $\iota = \log(8 X_i A_i/p)$ is a log factor. $\qquad \square$

### G.5.3 Proof of Lemma G.9: Bound on $\text{BIAS}_{2,h}^{T,\text{ext}}$

*Proof.* For fixed $x_h$ and $b_{1:h-1}$, let $W = \{k \in [h-1] : b_k \neq a_k\}$ and $h'$ is the maximal element in $W$, so we have $h' = \tau_K$. We define $W(h)$ as the set $W \cup \{h'+1, \ldots, h\}$. We can rewrite $\text{BIAS}_{2,h}^{T,\text{ext}}$ as

$$\text{BIAS}_{2,h}^{T,\text{ext}} = \max_{\phi \in \Phi_i^K} \sum_{x_h \in \mathcal{X}_{i,h}} \sum_{b_{1:h-1}} \delta^K(a_{1:h-1}, b_{1:h-1}) \prod_{k=1}^{h-1} \phi_k(a_k | x_k, b_{1:k \wedge h'})$$

$$\times \max_a \sum_{t=1}^{T} \prod_{k=1}^{h'} \pi_i^t(b_k | x_k) \left( \widetilde{L}_{(x_h, b_{1:\tau_K})}^t(a) - L_{i,h}^t(x_{i,h}, a) \right)$$

$$\leq \max_{\phi \in \Phi_i^K} \sum_{x_h \in \mathcal{X}_{i,h}} \sum_{b_{1:h-1}} \delta^K(a_{1:h-1}, b_{1:h-1}) \prod_{k=1}^{h-1} \phi_k(a_k | x_k, b_{1:k \wedge h'})$$

$$\times \max_{\phi_h'} \sum_{a_h} \phi_h'(a_h | x_h, b_{1:k \wedge h'}) \sum_{t=1}^{T} \prod_{k=1}^{h'} \pi_i^t(b_k | x_k) \left( \widetilde{L}_{(x_h, b_{1:h'})}^t(a_h) - L_{(x_h, b_{1:h'})}^t(a_h) \right)$$

$$= \max_{\phi \in \Phi_i^K} \sum_{x_{i,h}, a_h} \sum_{b_{1:h-1}} \delta^K(a_{1:h-1}, b_{1:h-1}) \prod_{k=1}^{h} \phi_k(a_k | x_k, b_{1:k \wedge h'})$$

$$\times \sum_{t=1}^{T} \prod_{k=1}^{h'} \pi_i^t(b_k | x_k) \left( \widetilde{L}_{(x_h, b_{1:h'})}^t(a_h) - L_{(x_h, b_{1:h'})}^t(a_h) \right)$$

$$= \max_{\phi \in \Phi_i^K} \sum_{x_{i,h}, a_h} \sum_{b_{1:h-1}} \delta^K(a_{1:h-1}, b_{1:h-1}) \frac{\prod_{k=1}^{h} \phi_k(a_k | x_k, b_{1:k \wedge h'})}{\prod_{k \in W(h)} \pi_{i,k}^{\star,h}(a_k | x_k)}$$

$$\times \underbrace{\sum_{t=1}^{T} \prod_{k=1}^{h'} \pi_i^t(b_k | x_k) \left( \widetilde{L}_{(x_h, b_{1:h'})}^t(a_h) - L_{(x_h, b_{1:h'})}^t(a_h) \right) \prod_{k \in W(h)} \pi_{i,k}^{\star,h}(a_k | x_k)}_{:= \widetilde{\Delta}_t^{x_h, b_{1:h'}, a_h}}$$

Here, (i) comes from the fact that the inner max over $\phi_h'$ and the outer max over $\phi_{1:h-1}$ are separable and thus can be merged into a single max over $\phi_{1:h}$.

Observe that the random variable $\widetilde{\Delta}_t^{x_h, b_{1:h'}, a_h}$ satisfy the following:

- By the definition of $\widetilde{L}$, we can rewrite $\widetilde{\Delta}_t^{x_h, b_{1:h'}, a_h}$ as

$$
\widetilde{\Delta}_t^{x_h, b_{1:h'}, a_h}
$$

$$
= \prod_{k \in W} \pi_{i,k}^t(b_k|x_k) \mathbf{1}\left\{(x_h, a_h) = (x_h^{t,(h,h',W)}, a_h^{t,(h,h',W)})\right\} \cdot \left(H - h + 1 - \sum_{h''=h}^{H} r_{i,h''}^{t,(h,h',W)}\right)
$$
$$
- \prod_{k \in [h']} \pi_{i,k}^t(b_k|x_k) L_{(x_h, b_{1:h'})}^t(a_h) \prod_{k \in W(h)} \pi_{i,k}^{\star,h}(a_k|x_k).
$$

- $\widetilde{\Delta}_t^{x_h, b_{1:h'}, a_h} \leq H.$

- $\mathbb{E}[\widetilde{\Delta}_t^{x_h, b_{1:h'}, a_h}|\mathcal{F}_{t-1}] = 0$, where $\mathcal{F}_{t-1}$ is the $\sigma$-algebra containing all information until $\pi^t$ is sampled.

- The conditional variance $\mathbb{E}[(\widetilde{\Delta}_t^{x_h, b_{1:h'}, a_h})^2|\mathcal{F}_{t-1}]$ can be bounded as

$$
\mathbb{E}\left[\left(\widetilde{\Delta}_t^{x_h, b_{1:h'}, a_h}\right)^2\bigg|\mathcal{F}_{t-1}\right]
$$
$$
\leq \mathbb{E}\left[\left(H - h + 1 - \sum_{h''=h}^{H} r_{i,h''}^{t,(h,h',W)}\right)^2 \left(\prod_{k \in W} \pi_{i,k}^t(b_k|x_k) \mathbf{1}\left\{(x_h, a_h) = (x_h^{t,(h,h',W)}, a_h^{t,(h,h',W)})\right\}\right)^2\bigg|\mathcal{F}_{t-1}\right]
$$
$$
\leq H^2 \prod_{k \in W} \pi_{i,k}^t(b_k|x_k) \mathbb{P}^{((\pi_{-i}^{\star,h})_{k \in W(h)}(\pi_{i,k}^t)_{k \in [h']\setminus W}) \times \pi_{-i}^t}\left((x_h^{t,(h,h',W)}, a_h^{t,(h,h',W)}) = (x_h, a_h)\right)
$$
$$
= H^2 \prod_{k \in W} \pi_{i,k}^t(b_k|x_k) \prod_{k \in W} \pi_{i,k}^{\star,h}(a_k|x_k) \cdot \prod_{k \in [h']\setminus W} \pi_{i,k}^t(a_k|x_k) \cdot \pi_{i,h}^{\star,h}(a_h|x_h) p_{1:h}^{\pi_{-i}^t}(x_{i,h})
$$
$$
\overset{(i)}{=} H^2 \prod_{k \in [h']} \pi_{i,k}^t(b_k|x_k) p_{1:h}^{\pi_{-i}^t}(x_{i,h}) \prod_{k \in W(h)} \pi_{i,k}^{\star,h}(a_k|x_k).
$$

Here, (i) is because $\prod_{k \in [h']\setminus W} \pi_{i,k}^t(a_k|x_k) = \prod_{k \in [h']\setminus W} \pi_{i,k}^t(b_k|x_k)$.

Therefore, we can apply Freedman's inequality and union bound to get that for any fixed $\lambda \in (0, 1/H]$, with probability at least $1 - p/8$, the following holds simultaneously for all $(h, x_{i,h}, b_{1:h}, a_h)$:

$$
\sum_{t=1}^{T} \widetilde{\Delta}_t^{x_h, b_{1:h'}, a_h} \leq \lambda H^2 \prod_{k \in W(h)} \pi_{i,k}^{\star,h}(a_k|x_k) \sum_{t=1}^{T} \prod_{k \in [h']} \pi_{i,k}^t(b_k|x_k) p_{1:h}^{\pi_{-i}^t}(x_{i,h}) + \frac{C \log(X_i A_i/p)}{\lambda},
$$

where $C > 0$ is some absolute constant. Plugging this bound into $\mathrm{BIAS}_{2,h}^{T,\mathrm{ext}}$ yields that, for all $h \in [H]$ and $\phi \in \Phi_i^K$,

$$
\mathrm{BIAS}_{2,h}^{T,\mathrm{ext}} \leq \max_{\phi \in \Phi_i^K} \sum_{x_h, a_h} \sum_{b_{1:h-1}} \delta^K(a_{1:h-1}, b_{1:h-1}) \frac{\prod_{k=1}^{h} \phi_k(a_k|x_k, b_{1:k\wedge h'})}{\prod_{k \in W(h)} \pi_{i,k}^{\star,h}(a_k|x_k)} \cdot
$$
$$
\times \left[\lambda H^2 \prod_{k \in W(h)} \pi_{i,k}^{\star,h}(a_k|x_k) \sum_{t=1}^{T} \prod_{k \in [h']} \pi_{i,k}^t(b_k|x_k) p_{1:h}^{\pi_{-i}^t}(x_{i,h}) + \frac{C \log(X_i A_i/p)}{\lambda}\right]
$$
$$
\leq \max_{\phi \in \Phi_i^K} \lambda H^2 \sum_{x_h, a_h} \sum_{b_{1:h-1}} \delta^K(a_{1:h-1}, b_{1:h-1}) \prod_{k=1}^{h} \phi_k(a_k|x_k, b_{1:k\wedge h'}) \sum_{t=1}^{T} \prod_{k \in [h']} \pi_{i,k}^t(b_k|x_k) p_{1:h}^{\pi_{-i}^t}(x_{i,h})
$$
$$
+ \max_{\phi \in \Phi_i^K} \frac{C \log(X_i A_i/p)}{\lambda} \sum_{x_h, a_h} \sum_{b_{1:h-1}} \delta^K(a_{1:h-1}, b_{1:h-1}) \frac{\prod_{k=1}^{h} \phi_k(a_k|x_k, b_{1:k\wedge h'})}{\prod_{k \in W(h)} \pi_{i,k}^{\star,h}(a_k|x_k)}.
$$

Note that for fixed $t$ and any $\phi \in \Phi_i^K$, by Corollary D.2,

$$
\sum_{x_h \in \mathcal{X}_{i,h}} \sum_{b_{1:h-1}} \delta^K(a_{1:h-1}, b_{1:h-1}) \prod_{k=1}^{h-1} \phi_k(a_k|x_k, b_{1:k\wedge h'}) \prod_{k \in [h']} \pi_{i,k}^t(b_k|x_k) p_{1:h}^{\pi_{-i}^t}(x_{i,h}) \leq 1.
$$

So we have

$$\sum_{x_h, a_h} \sum_{b_{1:h-1}} \delta^K(a_{1:h-1}, b_{1:h-1}) \prod_{k=1}^{h} \phi_k(a_k|x_k, b_{1:k \wedge h'}) \sum_{t=1}^{T} \prod_{k \in [h']} \pi_{i,k}^t(b_k|x_k) p_{1:h}^{\pi_{-i}^t}(x_{i,h}) \leq T.$$

Moreover, by the inequality (36) in the proof of Lemma G.7, we have for any $\phi \in \Phi_i^K$,

$$\sum_{x_h, a_h} \sum_{b_{1:h-1}} \delta^K(a_{1:h-1}, b_{1:h-1}) \frac{\prod_{k=1}^{h} \phi_k(a_k|x_k, b_{1:k \wedge h'})}{\prod_{k \in W(h)} \pi_{i,k}^{\star,h}(a_k|x_k)}$$

$$= \sum_{x_h} \sum_{b_{1:h-1}} \delta^K(a_{1:h-1}, b_{1:h-1}) \frac{\prod_{k=1}^{h-1} \phi_k(a_k|x_k, b_{1:k \wedge h'})}{\prod_{k \in W(h)} \pi_{i,k}^{\star,h}(a_k|x_k)}$$

$$\leq X_{i,h} \binom{h-1}{K} A_i^{K \wedge H}$$

Plugging these bounds into $\mathrm{BIAS}_{2,h}^{T,\mathrm{ext}}$ yields that, with probability at least $1 - p/8$, for all $h \in [H]$,

$$\mathrm{BIAS}_{2,h}^{T,\mathrm{ext}} \leq \lambda H^2 T + \frac{C \log(X_i A_i / p)}{\lambda} X_{i,h} \binom{h-1}{K} A_i^{K \wedge H}.$$

Taking summation over $h \in [H]$, we have

$$\sum_{h=1}^{H} \mathrm{BIAS}_{2,h}^{T,\mathrm{ext}} \leq \lambda H^3 T + \frac{C \log(X_i A_i / p)}{\lambda} \sum_{h=1}^{H} X_{i,h} \binom{h-1}{K} A_i^{K \wedge H}$$

$$\leq \lambda H^3 T + \frac{C \log(X_i A_i / p)}{\lambda} X_i \binom{H}{K} A_i^{K \wedge H},$$

for all $\lambda \in (0, 1/H]$. Choose $\lambda = \min \left\{ \frac{1}{H}, \sqrt{\frac{C X_i \binom{H}{K \wedge H} A_i^{K \wedge H} \log(X_i A_i / p)}{H^3 T}} \right\}$, we obtain the bound

$$\sum_{t=1}^{T} \mathrm{BIAS}_{2,h}^{T,\mathrm{ext}} \leq \mathcal{O} \left( \sqrt{H^3 \binom{H}{K \wedge H} A_i^{K \wedge H} X_i T \iota} + H \binom{H}{K \wedge H} A_i^{K \wedge H} X_i \iota \right),$$

where $\iota = \log(8 X_i A_i / p)$ is a log factor. $\qquad \square$

# H  Additional discussions

## H.1  Implementation of $K$-EFCE by the mediator

Given a $K$-EFCE, the mediator can implement it as follows. Before the game starts, the mediator samples a product policy from the $K$-EFCE (which is a correlated policy), and initializes a "deviation counter" for each player at 0. Then, at each round, the mediator by default recommends the sampled actions to all players. After players take their actual action, the mediator increments each player's "deviation counter" by 1 if their action is different from the recommendation. The mediator stops recommending to any player as soon as their counter reaches $K$. We remark that such an implementation (viewed from the mediator's side) corresponds exactly to the definition of a $K$-EFCE strategy modification (for the player's side) in Definition 1 and Algorithm 1.

## H.2  Requirement on knowing the tree structure in bandit-feedback setting

Under bandit-feedback, in Algorithm 3 and 6, the input balanced exploration policies $\{\pi_i^{\star,h}\}_{h \in [H]}$ depend on the number of children $\mathcal{C}_h(x_{i,h}, a_{i,h})$ for all infoset $x_{i,h}$ and action $a_{i,h}$. This requires knowing the structure of each player's game tree (treeplex). A similar requirement is also needed in the Balanced OMD and Balanced CFR algorithm of Bai et al. [5]. We remark that this requirement is relatively mild as the tree structure can be extracted efficiently from just one tree traversal for each player.

### H.3 Timeability condition

Our formulation of tree-structure, perfect-recall POMGs is able to express any IIEFG with perfect recall and the additional *timeability* condition [25], a mild condition which roughly speaking requires that infosets for all players combinedly could be partitioned into ordered "layers". Therefore, our results hold for all timeable IIEFGs with perfect recall.

Furthermore, our algorithm $K$-EFR and Balanced $K$-EFR does not depend on the joint game tree for all players. Instead, the $i^{\text{th}}$ player's version of our algorithms only depends on the $i^{\text{th}}$ player's own game tree (which is timeable for any perfect-recall IIEFG). Therefore, our algorithms and theoretical guarantees (when formulated in general IIEFGs with perfect recall) can be generalized directly to any perfect-recall IIEFGs that is not necessarily timeable, similar as existing CFR/OMD type algorithms for external regret minimization [45, 19, 5].

### H.4 Comparison between $K$-EFR and [11, 20] for learning EFCE under full feedback

Celli et al. [11] and its extended version [20] design the first uncoupled no-regret algorithm for computing EFCEs under full feedback. Their algorithms are based on a two-level regret decomposition, which first decomposes the EFCE regret into trigger regrets [22], one for each subtree policy at each infoset, and minimizing each trigger regret via Counterfactual Regret Minimization (CFR). By contrast, our $K$-EFR utilizes a slightly different decomposition, which decomposes the $K$-EFCE regret directly into wide-range regrets (7) at each infoset, and uses wide-range regret minimization with WRHEDGE to learn the $K$-EFCE. We remark that, for learning the EFCE, our approach works by learning the 1-EFCE, which is equivalent to the (trigger definition of the) EFCE in terms of the exact equilibria they define, but a slightly stricter version in terms of $\varepsilon$-approximate equilibria (cf. Proposition C.1).

Morrill et al. [34] considers various forms of correlated equilibria and also uses wide-range regret minimization to learn these equilibria. When specializing in EFCE, we improve its result by a more refined analysis and our new wide-range regret minimization algorithm (WRHEDGE).