# OpenReview forum: "Sample-Efficient Learning of Correlated Equilibria in Extensive-Form Games"
_NeurIPS.cc/2022/Conference — NeurIPS 2022 Accept_

### Official Review · Reviewer_RRjP · 2022-07-12

**Rating:** 7
**Confidence:** 4
**Soundness:** 4 excellent
**Presentation:** 4 excellent
**Contribution:** 3 good

**Summary:**

The paper's main contributions are threefold:

1. The notion of K-EFCE is defined. It is a generalization of EFCE, whereby the mediator (correlation device) can condition their recommendation based on the number of deviations of each player, which can be any number between 0 and K. After K deviations have been reached, the mediator will stop issuing recommendations. K-EFCE is a natural generalization of EFCE: in fact, EFCE is simply a 1-EFCE.

2. The paper shows that, within the *full-information* setting, an $\epsilon$-approximated K-EFCE can be computed by means of uncoupled no-regret dynamics after running $\tilde O(A^K / \epsilon^2)$ iterations where $A$ is the maximum number of player actions (equivalently, each player is guaranteed to cumulate $\tilde O(A^K \sqrt T)$ regret after $T$ iterations).

3. The paper shows that, withing the *trajectory-bandit setting*, an $\epsilon$-approximated K-EFCE can be computed by means of uncoupled no-regret dynamics after running $\tilde O(A^{K+1} / \epsilon^2)$ iterations where $A$ is the maximum number of player actions (equivalently, each player is guaranteed to cumulate $\tilde O(A^{K+1} \sqrt T)$ regret after $T$ iterations).


**Questions:**

1. Can you please respond to my questions in the "Clarity and quality" assessment above?

2. Currently, the main factor driving my (positive!) score is the originality of the work. While I also gave a good look in the appendix, it is possible---especially given the volume of a 60-page paper---that I might have missed some particularly innovative analysis tool that was used in this paper. If that's the case, please push back on my assessment above.

3. Can you please confirm that your algorithm cannot be used in adversarial bandit-feedback settings (as it requires exploring the opponent for several episodes assuming the opponent's strategy does not change)?

**Limitations:**

The authors adequately addressed the limitations and potential negative societal impact of their work.

**Strengths And Weaknesses:**

I want to start by saying that I think the paper represents interesting, well-executed, high-quality work. Despite the very tight page limits, the body is well-organized and easy to follow. The appendix is also well-organized, and contains a trove of additional interesting results---I especially enjoyed browsing through appendix C.

Significance: I believe the paper contributes interesting results that fit well within the current research trend around extensive-form games. For example, learning EFCE with bandit feedback is a natural and interesting question.

Originality: The extension from EFCE to K-EFCE is interesting conceptually, and like I said I believe the paper is well executed and addresses relevant questions. In terms of significance, my impression is that the paper does not introduce any groundbreaking tool in the space. Rather, it fundamentally combines several expected existing techniques, such as balanced exploration policies and standard decompositions of Phi-regret for EFCE, obtaining results that were perhaps easy to guess for an expert in the field. I want to be clear that I am not saying that the paper is trivial---the authors put together and master several specialized tools, creating a paper that I expect will be very well received in the community.

Clarity and quality: the paper is very well executed and polished, both in the body and in the appendix. There is only one point that I found was slightly confusing: on lines 40-41, the bandit feedback setting is introduced as the opposite of the case in which "the full game is known". Just to be clear: your bandit algorithm still requires that at least the *decision space* (treeplex) of each player be known so that the balanced exploration strategy can be instantiated, correct? So, this bandit algorithm is not akin to, say, that of the cited works by Farina and Sandholm ("Model-Free Online Learning...") or Kozuno et al. ("Model-free learning"), which instead address both an unknown structure *and* bandit feedback on the utilities?

Minor nits:
- Abstract, line 9: I would avoid "more generalized" in favor of either "generalized" or "more general".

---

> ### Author Response · Authors · 2022-08-02
> **Response to Reviewer RRjP**
>
> Thank you for your thoughtful reviews and positive feedback on our paper! We respond to your comments as follows.
> > Your bandit algorithm still requires that at least the decision space (treeplex) of each player be known so that the balanced exploration strategy can be instantiated, correct? So, this bandit algorithm is not akin to, say, that of the cited works by Farina and Sandholm ("Model-Free Online Learning...") or Kozuno et al. ("Model-free learning"), which instead address both an unknown structure and bandit feedback on the utilities?
>
> Right, in our bandit algorithm, the balanced exploration policy requires knowing the structure of each player’s decision space (treeplex), and thus the algorithm does not yet handle the unknown tree structure setting as (Farina and Sandholm, 2021) and (Kozuno et al. 2021). We have added a short discussion about this in Appendix H.2.
>
> Currently, the balanced exploration policy is needed in order to obtain the sharp (linear in $X$) sample complexity under bandit feedback. We believe that achieving the same rate under unknown tree structures, e.g. by “learning” an exploration policy that works as well as the balanced one, would be an interesting open question.
>
> > Can you please confirm that your algorithm cannot be used in adversarial bandit-feedback settings (as it requires exploring the opponent for several episodes assuming the opponent's strategy does not change)?
>
> Right, in the bandit-feedback setting, our algorithm requires the opponents to fix their strategy for a constant number of episodes (Algorithm 3 & 6), and thus it cannot be used to achieve no-regret in the adversarial setting (even though it can be used to learn the equilibrium in the self-play setting). Under full feedback, our algorithm can handle adversarial opponents and have a regret bound (Corollary F.1).
>
> > Minor nits: Abstract, line 9: I would avoid "more generalized" in favor of either "generalized" or "more general".
>
> We have changed this to “generalized” in our revision.

---

> > ### Comment · Reviewer_RRjP · 2022-08-09
> > **Thank you!**
> >
> > Thanks for your response. I don't have any additional questions. It might make sense to explicitly mention the two weaknesses with the current approach (difficulties in achieving model-freeness and adversarial bandit feedback).

---

### Official Review · Reviewer_hT1T · 2022-07-12

**Rating:** 7
**Confidence:** 2
**Soundness:** 3 good
**Presentation:** 3 good
**Contribution:** 3 good

**Summary:**

The paper has 3 main contributions. First, the authors propose a generalization of extensive-form correlated equilibrium (EFCE), known as K-EFCE. In this generalization, players continue to receive recommendations even after they have deviated from their recommendations (received at each infoset, just like regular EFCE) for up to K deviations. K-EFCE encompass a number of common equilibrium concepts. A 1-efce is just a regular EFCE, while a 0-EFCE is a normal-form CCE.  The second key contribution is a self-play algorithm to compute K-EFCE, which the authors call K-EFR. The third contribution is the a variant of K-EFR that can handle bandit feedback. The authors claim that this is the first algorithm for learning EFCE from bandit feedback.

**Questions:**

1. I would have liked a discussion on how K-EFCE could be implemented. In the original EFCE paper by von Stengel, EFCEs were implemented by means of sealed recommendations, even though the more popular formulation today is based on [13, 16], which use an equivalent formulation based on trigger sequences. How can a mediator (practically) achieve a K-EFCE where K > 1?

2. The authors use POMG as their choice of framework, rather than extensive form games. Is there a difference between the 2, given that the POMG is restricted to be tree structured?



**Limitations:**

-

**Strengths And Weaknesses:**

Generally, I felt there was too much content to follow and feels more suitable for a journal. In particular, the part on bandit feedback feels rather cramped (though it is an important contribution in its own right). Given the authors’ claim that K-EFR outperforms [11], [20], I felt it would be good to give a more detailed sketch on the key differences between these methods—-this is currently done in lines 233-240, which (at least to me) was rather opaque. This should help to make the paper more accessible.

Other than that, the paper seems good, except for the aforementioned points about it not being as accessible to readers not already familiar with EFR.

Small quips:
In the definition of Type-I rechistories, $\mathcal{A}_i^{h-1}$ is not defined.

Note: The paper is entirely theoretical with many pages of proof. I did not go through all the proofs, especially those in the appendix. I am not familiar with the state of the art EFCE solvers, and am unable to definitively comment on the authors’ claims with regard to comparison with other works.

---

> ### Author Response · Authors · 2022-08-02
> **Response to Reviewer hT1T**
>
> Thank you for your thoughtful reviews and positive feedback on our paper! We respond to your comments as follows.
> > Given the authors’ claim that K-EFR outperforms [11], [20], I felt it would be good to give a more detailed sketch on the key differences between these methods…This should help to make the paper more accessible.
>
> We have added a more detailed comparison between KEFR and the algorithm of [11,20] in Appendix H.4 in our revision, and will make sure to include it in the main text in our final version.
> > How can a mediator (practically) achieve a K-EFCE where K > 1?
>
> Given a $K$-EFCE, the mediator can implement it as follows. Before the game starts, the mediator samples a product policy from the $K$-EFCE (which is a correlated policy), and initializes a “deviation counter” for each player at 0. Then, at each round, the mediator by default recommends the sampled actions to all players. After players take their actual action, the mediator increments each player’s “deviation counter” by 1 if their action is different from the recommendation. The mediator stops recommending to any player as soon as their counter reaches $K$. We have added this in Appendix H.1 in our revision.
> > The authors use POMG as their choice of framework, rather than extensive form games. Is there a difference between the 2, given that the POMG is restricted to be tree structured?
>
> Our POMG formulation with tree-structure and perfect-recall assumptions is almost equivalent to the usual formulation of imperfect-information extensive-form games (IIEFGs). Concretely, our formulation can express any IIEFG with perfect recall and the additional timeability condition, a mild condition which roughly speaking requires that infosets for all players combinedly could be partitioned into ordered “layers”. Further, we believe our algorithms and theoretical guarantees can hold for any general IIEFG that is not necessarily timeable, and we have added a short discussion about this in Appendix H.3 (see also our response to reviewer tyaC for details).
> > Small quips: In the definition of Type-I rechistories $\mathcal{A}_i^{h-1}$ is not defined.
>
> We have added the definition in our revision (Line 185-186).

---

> > ### Comment · Reviewer_hT1T · 2022-08-08
> > **Response**
> >
> > Apologies for the late reply.
> >
> > In the original EFCE paper by Von Stengel, deviations were not "detected" by the mediator per se, and EFCEs were implemented by means of sealed recommendations. That is, reduced normal form strategies were sampled from the beginning and recommendations revealed only after reaching an information state. Because a reduced normal form is sampled, if a player deviates, the reduced normal form no longer contains any recommendations for following information sets. This implementation means that the mediator doesn't play an active role while the game is being played, and more importantly, does not require the mediator to know when a player has deviated (a player's action could be private). The authors' proposed mechanism *does* require the mediator to know each player's actions exactly (to detect deviations). Nonetheless, this is a relatively minor point and does not take away from the authors' key contribution.
> >
> > I have no additional questions, and am leaving my score as is.

---

### Official Review · Reviewer_tyaC · 2022-07-12

**Rating:** 7
**Confidence:** 5
**Soundness:** 2 fair
**Presentation:** 3 good
**Contribution:** 3 good

**Summary:**

This paper studies the problem of learning extensive-form correlated equilibria (EFCE) in imperfect-information extensive-form games. In particular, the paper introduces a new notion of EFCE, called K-EFCE, in which a player does not stop receiving recommendation right after the first time they deviate, but they continue to receive them up to their K-th deviation. The paper studies the problem of learning K-EFCE, and it provides sample-efficient algorithms for both the full feedback and the bandit feedback settings.

**Questions:**

1) Can you please address my concern in the second point of the quality assessment?
---------- AFTER AUTHORS' RESPONSE ----------
The authors clarified my concerns about the applicability of their algorithms to non-timeable perfect-recall games. As a result, I raised my score to 7 (from an original score of 6).

**Limitations:**

Yes.

**Strengths And Weaknesses:**

ORIGINALITY

(+) To the best of my knowledge, this is the first paper that proposes sample-efficient no-regret learning dynamics that converge to EFCEs under the bandit feedback model.

QUALITY

(+) As far as I am concerned, the results in the paper are correct, though I did not check all the proofs in the details.

(-) I only have one major concern about the formalization of imperfect-information extensive-form games with perfect recall. In particular, the authors claim that in order to have perfect recall it should be possible to partition the information sets of one player according to the tilmestep they belong to. However, the fact that the game is perfect recall does not necessarily imply that the states in the same infoset all refer to the same timestep of the game. Indeed, this is not the case whenever the game is not timeable. As far as I am concerned, the results in the paper rely on the assumption that the game is timeable, which is a more restrictive assumption than perfect recall only. If this is correct, this would somehow limit the impact of the results. See the following paper for a reference on timeability of extensive-form games:

“Jakobsen, S. K., Sørensen, T. B., & Conitzer, V. (2016, January). Timeability of extensive-form games. In Proceedings of the 2016 ACM Conference on Innovations in Theoretical Computer Science (pp. 191-199).”

CLARITY

(+) The paper is well written and all the concepts are well presented.

(-) My only concern regarding the presentation is that too many results are in the appendix, which renders the main paper (perhaps too much) high level in terms of technical details.

SIGNIFICANCE

(+) The problem of learning EFCEs is receiving considerable attention over the last few years from the algorithmic game theory community and beyond. Thus, this paper could be of interest for a considerable portion of the NeurIPS community (as also testified by the Best Paper Award at NeurIPS 2020).

---

> ### Author Response · Authors · 2022-08-02
> **Response to Reviewer tyaC**
>
> Thank you for your support of our paper and the valuable feedback! We respond to your question as follows.
> > The fact that the game is perfect recall does not necessarily imply that the states in the same infoset all refer to the same timestep of the game. Indeed, this is not the case whenever the game is not timeable. As far as I am concerned, the results in the paper rely on the assumption that the game is timeable, which is a more restrictive assumption than perfect recall only.
>
> Thank you for bringing up this important point. We agree that our formulation of IIEFGs through POMGs (with tree structure and perfect recall) implicitly made the timeability assumption.
>
> However, upon an initial look at our analysis, we believe our algorithms and theoretical guarantees (when formulated in general IIEFGs with perfect recall) can indeed generalize to any general IIEFG as well that is not necessarily timeable. This is because our algorithms K-EFR and Balanced K-EFR only depend on each player’s own game tree (infosets) instead of the joint game tree for all players, similar to existing CFR/OMD type algorithms for external-regret minimization (Zinkevich et al. 2007,  Farina al. 2020, Bai et al. 2022).
>
> We have added a discussion about timeability in Appendix H.3 in our revision, and will make sure to include it in the main text in the final version.

---

> > ### Comment · Reviewer_tyaC · 2022-08-08
> > **Thanks for your response**
> >
> > Thank you very much for your detailed response to my concerns. As I was expecting, you made me clear the fact that your algorithms can indeed work under the general model of perfect-recall games that are not timeable. Thus, I will increase my score from 6 to 7. However, I strongly encourage the authors to provide additional explanations on this point in the final version of the paper. In particular, I suggest the authors to add some comments in the technical part of the paper, in those points in which you used the timeability assumption in order to ease notation/presentation. In this way, it would be easier to track in which points your results can be easily generalized to the non-timeable case.

---

### Official Review · Reviewer_cdkB · 2022-07-13

**Rating:** 6
**Confidence:** 3
**Soundness:** 3 good
**Presentation:** 3 good
**Contribution:** 2 fair

**Summary:**

The authors first introduce a new equilibrium concept: K-EFCE in extensive-form games. Then the authors propose sample-efficient regret-minimization based algorithms to learn the equilibrium in both full-information and bandit feedback. When K=1 under the bandit feedback, this is the first sample-efficient algorithm to learn EFCE.

**Questions:**

1. While this is not specific to this paper, I think POMGs are much more general than the authors introduce. Specifically, a POMG, like a POMDP, should have an observation model $O(\cdot|s,a)$, which is a conditional probability given state s and action a. In the paper, the authors simplify this to a partition of the state space, which is sufficient for IIEFGs but I think POMGs are more than that and there is no need to bring up this term.

2. Why is K-EFCE is an interesting extension to EFCE? Without this extension, can the reducing algorithms be further simplified?


**Strengths And Weaknesses:**

To me it seems like the most important contribution of this paper is the first sample-efficient algorithm to learn EFCE under the bandit feedback. On the other hand, I don't really get why the authors introduce K-EFCE, a generalization of EFCE. Intuitively, we can definitely consider this generalization, but I don't know if the generalization induces interesting concepts or properties and the contribution I mentioned can completely hold without this generalization if I understand correctly.

In terms of algorithms, it directly adapts WRHedge for the online learning literature, which is quite reasonable but I think this may limit the novelty of this paper. In the bandit setting, it looks like the balanced sampling idea also comes directly from another paper. Another weakness of this paper is probably the lack of experimental evaluations of the algorithms.

---

> ### Author Response · Authors · 2022-08-02
> **Response to Reviewer cdkB**
>
> Thank you for your support of our paper and the valuable feedback! We respond to your comments as follows.
> > In terms of algorithms, it directly adapts WRHedge for the online learning literature…it looks like the balanced sampling idea also comes directly from another paper.
>
> We agree that both WRHedge and the balanced sampling ideas are not new. However, we made non-trivial adaptations of both ideas in order to facilitate our proofs.
> 1. For WRHedge, we generalized the algorithm to the stochastic setting (SWRHedge) with nontrivial adaptations, so that it can admit *multiple* sample-based loss estimators in each round, one for each time selection function, with the same mean (cf. Line 310-313).
> 2. Our balanced exploration policy is the same as in (Bai et al. 2022), but the way we use it is new. The sampling policies in Algorithm 3 are *interlaced concatenations* of the current policy and the balanced exploration policy along time steps $h$ (cf. Line 298-301).
>
> > While this is not specific to this paper, I think POMGs are much more general than the authors introduce…I think POMGs are more than that and there is no need to bring up this term.
>
> We agree that in general POMGs are much broader than IIEFGs. We formulate IIEFGs as POMG with tree-structure and perfect-recall mostly due to its clearer notation system (in our opinion), which helped simplify the presentation in various algorithms and proofs.
> We have added a footnote (Footnoe 2, Page 3) in our revision to highlight that we only consider a specific subclass of POMGs instead of general POMGs.
>
> > Why is K-EFCE an interesting extension to EFCE?
>
> We believe K-EFCEs are a fairly natural generalization of the (original) EFCE. The EFCE can be viewed as a mediator "penalizing" the player (does not reveal recommendations) after any deviation. In this sense, the mediator of a K-EFCE is "more relaxed" in its penalization, only penalizing when a player has deviated for $K$ times. Therefore, it is a natural way to make deviating players stronger (hence equilibrium defined w.r.t. such deviations stricter). We also remark that, considering general definitions of correlated equilibria beyond EFCE is of interest in the community, e.g. (Morrill et al. 2021). Indeed, the two extreme cases of K-EFCE with $K=0$ and $K=\infty$ are equivalent to (Normal-Form) Coarse Correlated Equilibrium and the "Behavioral Correlated Equilibria'' of (Morrill et al. 2021; cf. our Proposition C.3).
>
> > Without this extension, can the reducing algorithms be further simplified?
>
> Plugging $K=1$ into our algorithms gives algorithms for learning the 1-EFCE (i.e. the standard EFCE). In that case, the algorithm may be written in mathematically equivalent but slightly simplified forms (for example, definitions of type-I and type-II rechistories, and Balanced Sampling strategies in Algorithm 3, Line 1-3, may be presented in simplified notation when $K=1$).

---

### Author Response · Authors · 2022-08-02
**Revision uploaded**

We thank all reviewers for their valuable feedback on our paper. We have revised our paper to incorporate the reviewers’ suggestions. For clarity, all changes are marked in red.

---

### Meta-Review · Area_Chair_41WV · 2022-08-20

**Recommendation:** Accept
**Confidence:** Certain

**Metareview:**

Reviews on this paper are uniformly positive, and all reviewers feel that it is
an interesting set of results. One minor criticism is that  some reviewers felt the presentation
could be improved, and the authors should try to address this for the camera ready.


**Award:**

No

---

### Decision · Program_Chairs · 2022-09-14

Accept